# Policy Gradients for Cumulative Prospect Theory in Reinforcement Learning

## Abstract

We derive a policy gradient theorem for Cumulative Prospect Theory (CPT) objectives in finite-horizon Reinforcement Learning (RL), generalizing the standard policy gradient theorem and encompassing distortion-based risk objectives as special cases. Motivated by behavioral economics, CPT combines an asymmetric utility transformation around a reference point with probability distortion. Building on our theorem, we design a first-order policy gradient algorithm for CPT-RL using a Monte Carlo gradient estimator based on order statistics. We establish statistical guarantees for the estimator and prove asymptotic convergence of the resulting algorithm to first-order stationary points of the (generally nonconvex) CPT objective. We complement our asymptotic analysis with a non-asymptotic total sample complexity analysis to reach an approximate first-order stationary policy. Simulations illustrate qualitative behaviors induced by CPT and compare our first-order approach to existing zeroth-order methods.

## 1 Introduction

In classical reinforcement learning (RL), rational agents make decisions to maximize their expected cumulative rewards through interaction with their environment. This paradigm is primarily guided by expected utility theory, which has long been the dominant framework for decision-making under risk. Within this framework, agents are generally considered to be risk-neutral; however, risk-seeking and risk-averse behaviors can also be modeled by modifying the utility function, thereby adjusting the policy optimization objective (see e.g. Prashanth et al. (2022); Biswas & Borkar (2023); Bäuerle & Jaśkiewicz (2024) for recent surveys).

While risk-sensitive RL extends beyond risk-neutral settings to capture individual risk preferences (using e.g. variance or conditional value at risk metrics), many commonly used risk-sensitive objectives capture particular aspects of risk preferences (e.g., tail risk) but do not simultaneously model gain–loss asymmetry around a reference point together with probability distortion. In particular, many standard risk-sensitive criteria do not model the asymmetric perception of gains and losses, as well as the probability distortions inherent in human cognition, such as the tendency to overestimate rare events and underestimate frequent ones (see App. F for an illustration in a simulation). These behavioral phenomena, relevant in human-facing decision problems under uncertainty, are beyond the scope of traditional risk-sensitive RL frameworks, necessitating a more comprehensive approach.

Cumulative Prospect Theory (CPT), introduced by Kahneman and Tversky in their seminal works combining cognitive psychology and economics (Kahneman & Tversky, 1979; Tversky & Kahneman, 1992), provides a descriptive behavioral model of decision-making under risk to explain several empirical observations that challenge the standard expected utility theory. CPT models how individuals perceive outcomes asymmetrically, being risk-averse in the domain of gains and risk-seeking in the domain of losses, and distort probabilities to reflect cognitive biases. These insights have led to widespread applications of CPT in stateless static settings in domains such as healthcare in psychiatry (Sip et al., 2018; George et al., 2019; Mkrtchian et al., 2023), chronic diseases treatment (Zhao et al., 2023) and emergency decision making (Sun et al., 2022) where modeling behavioral factors can be important, as well as other application domains such as energy (Ebrahimigharehbaghi et al., 2022; Dorahaki et al., 2022) and finance (Ladrón de Guevara Cortés

et al., 2023; Luxenberg et al., 2024) to name a few. However, these applications often overlook the sequential decision-making nature of many real-world problems, where the outcome of one decision can affect future choices, a critical aspect of RL.

The integration of CPT into RL provides a promising avenue for behaviorally-aligned sequential decision-making, as it allows RL agents to consider both risk preferences and probability distortions in dynamic environments. While a few isolated recent works have explored this integration (L.A. et al., 2016; Borkar & Chandak, 2021; Ramasubramanian et al., 2021; Danis et al., 2023; Foo et al., 2023), the understanding and practical impact of CPT in RL remains limited. Specifically, the computational challenges and theoretical underpinnings of CPT-based RL models are still underexplored.

**Contributions.** In this work, we focus on the policy optimization problem where the objective is the CPT value of the cumulative sum of rewards, induced by a parametrized policy in a Markov Decision Process. Our main contributions are summarized as follows:

**(i) Policy gradient theorem for CPT-RL.** We establish a policy gradient theorem providing a closed form expectation expression for the gradient of our CPT-value objective w.r.t. the policy parameter under suitable regularity conditions on the utility and probability distortion functions. This result generalizes the standard policy gradient theorem in RL.

**(ii) Policy gradient algorithm for CPT-RL.** Building on our policy gradient theorem, we design a policy gradient (PG) algorithm to solve the CPT policy optimization problem. Our PG algorithm for CPT-RL uses first-order information and does not rely on zeroth-order policy gradient estimation.

**(iii) Convergence and sample complexity.** We show that our PG estimator is consistent and we analyze its sample complexity. Notably, our sample complexity scales logarithmically in the policy-parameter dimension (under our estimator and regularity assumptions), in contrast to existing zeroth-order PG estimators. We further show that the iterates of our PG algorithm converge asymptotically to the set of stationary points of the CPT-RL objective which is typically nonconvex in the policy parameters. We conclude our analysis with a total sample complexity analysis for reaching an approximate first-order stationary policy, by showing smoothness of the CPT objective under adequate assumptions on the policy parameterization, utility and weighting functions.

**(iv) Simulations.** We test our PG algorithm on several applications to illustrate the flexibility of CPT-RL compared to standard and risk-sensitive RL in leading to more nuanced behavior. We also compare the performance of our PG algorithm to the previously proposed zeroth order algorithm to compare against a zeroth-order baseline as the problem size increases in our simulation settings.

**Closest related work.** Closest to our work are policy gradient methods for distortion risk measures (DRMs) (Vijayan & L.A, 2024), which optimize distorted-expectation objectives via first-order gradients. CPT shares the probability distortion component but additionally incorporates a reference point and an asymmetric utility transformation (and potentially separate gain/loss distortions), enabling objectives beyond distortion-only criteria. Our policy gradient expression reduces to DRM policy gradients under the corresponding restrictions (e.g., identity utility and no reference point), while providing a first-order method for the more general CPT objective. We provide additional discussion after Theorem 5. A more comprehensive related work discussion can be found in section 6 and App. B.

## 2 From Classical RL To CPT-RL

### 2.1 MDPs, CPT and Notation

**Markov Decision Process.** A discrete-time Markov Decision Process (MDP) (Puterman, 2014) is a tuple $\mathcal{M} = (\mathcal{S}, \mathcal{A}, \mathcal{P}, r, H, \rho)$, where $\mathcal{S}, \mathcal{A}$ are respectively the state and action spaces, supposed to be finite for simplicity, $\mathcal{P} : \mathcal{S} \times \mathcal{A} \times \mathcal{S} \to [0, 1]$ is the state transition probability kernel, $r : \mathcal{S} \times \mathcal{A} \to [-r_{\max}, r_{\max}]$ is the reward function bounded by $r_{\max} > 0$, $\rho$ is the initial state probability distribution and $H \geq 1$ is a finite horizon. A randomized stationary Markovian policy, which we will simply call a policy, is a mapping $\pi : \mathcal{S} \to \Delta(\mathcal{A})$ which specifies for each $s \in \mathcal{S}$ a probability measure over the set of actions $\mathcal{A}$ by $\pi(\cdot|s) \in \Delta(\mathcal{A})$

where $\Delta(\mathcal{A})$ is the simplex over the finite action space $\mathcal{A}$. Each policy $\pi$ induces a discrete-time Markov reward process $\{(s_t, r_t := r(s_t, a_t))\}_{t \in \mathbb{N}}$ where $s_t \in \mathcal{S}$ represents the state of the system at time $t$ and $r_t$ corresponds to the reward received when executing action $a_t \in \mathcal{A}$ in state $s_t \in \mathcal{S}$. We denote by $\mathbb{P}_{\rho,\pi}$ the probability distribution of the Markov chain $(s_t, a_t)_{t \in \mathbb{N}}$ generated by the MDP controlled by policy $\pi$ with initial state distribution $\rho$. We use $\mathbb{E}_{\rho,\pi}$ (or often simply $\mathbb{E}$) to denote the expectation. In classical RL, the goal of the agent is to find a policy $\pi$ maximizing the expected return $J(\pi) := \mathbb{E}_{\rho,\pi}[\sum_{t=0}^{H-1} r_t]$ where $s_0 \sim \rho$ and $H \geq 1$ is a finite horizon.

**Policy classes.** We now introduce different sets of policies which will be important for stating our results. Each policy class is defined according to the information history the policies have access to for selecting actions. Here, a history $h_t \in \mathcal{H} := \bigcup_{h \in [H]} \mathcal{H}_h$ (where $\mathcal{H}_h := \mathcal{S}^h \times \mathcal{A}^h \times [-r_{\max}, r_{\max}]^h$) is a finite sequence of successive states, actions and rewards: $(s_0, a_0, r_0, ..., s_{t-1}, a_{t-1}, r_{t-1})$.[1] More specifically, throughout this work, we will consider the following sets of policies: $\Pi_{NM} := \{\mathcal{H} \to \Delta(\mathcal{A})\}$ is the set of policies that are not necessarily Markovian, $\Pi_{\Sigma,NS} := \{\mathcal{S} \times \mathbb{R} \times \mathbb{N} \to \Delta(\mathcal{A})\}$ is the set of non-stationary policies that only depend on the current state, the sum of rewards accumulated so far and the timestep (i.e. $\pi(s, \sum_{k=0}^{t-1} r_k, t)$), $\Pi_{\Sigma,S} := \{\mathcal{S} \times \mathbb{R} \to \Delta(\mathcal{A})\}$ is the set of policies that only depend on the state and the sum of rewards (i.e. $\pi(s, \sum_{k=0}^{t-1} r_k)$), $\Pi_{M,NS} := \{\mathcal{S} \times \mathbb{N} \to \Delta(\mathcal{A})\}$ is the set of (non-stationary) Markovian policies (i.e. $\pi(s, t)$) and $\Pi_{M,S} := \{\mathcal{S} \to \Delta(\mathcal{A})\}$ is the set of stationary Markovian policies, i.e. Markovian policies which are time-independent. Deterministic policies assign a single action to each state. For each set of policies defined above, we define their corresponding subset of deterministic policies with a superscript $D$, e.g. $\Pi_{NM}^D$.

**Remark 1.** $\Pi_{M,S} \subseteq \Pi_{M,NS} \subseteq \Pi_{\Sigma,NS} \subseteq \Pi_{NM}$ and $\Pi_{M,S} \subseteq \Pi_{\Sigma,S} \subseteq \Pi_{\Sigma,NS} \subseteq \Pi_{NM}$, see Fig. 4 in App. A.

**Cumulative Prospect Theory Value.** Instead of the expected return, CPT prescribes to consider the CPT value which relies on three distinct elements:

**(a) Reference point $x_0$.** Rewards larger than the reference are perceived as gains whereas lower values are viewed as losses.

**(b) Gain and loss utility functions.** The agent evaluates outcomes relative to a reference point $x_0$. We use two nonnegative utility functions $U^+, U^- : \mathbb{R}_+ \to \mathbb{R}_+$, where $U^+$ measures the subjective value of gains and $U^-$ measures the subjective magnitude of losses. We define, for every $x \in \mathbb{R}$,

$$u^+(x) := U^+(x - x_0)\mathbf{1}\{x \geq x_0\}, \qquad u^-(x) := U^-(x_0 - x)\mathbf{1}\{x \leq x_0\}.$$

Thus, both $u^+$ and $u^-$ are nonnegative, with $u^+$ active only on gains and $u^-$ active only on losses. Typically, $U^+$ is concave, capturing risk aversion over gains, while the signed loss value $-U^-(x_0 - x)$ is convex as a function of outcomes below the reference point, capturing risk seeking over losses. For concreteness, we will use Kahneman & Tversky (1979)'s utility function as a running example: $U^+(y) = y^\alpha, U^-(y) = \lambda y^\alpha, y \geq 0$ where $\lambda = 2.25$ and $\alpha = 0.88$ are commonly used parameters. Equivalently, $u^+(x) = (x - x_0)_+^\alpha, u^-(x) = \lambda(x_0 - x)_+^\alpha$ where $y_+ = \max\{0, y\}$ for any $y \in \mathbb{R}$. See Fig. 11 for an illustration with $x_0 = 0$.

**(c) Probability distortion function $w : [0, 1] \to [0, 1]$.** This is a continuous non-decreasing weight function that distorts the probability distributions of the gain and loss variables. Monotonicity here preserves ordering: it is natural to suppose that events with (truly) higher probability remain perceived as more probable than less truly probable ones after distortion. This distortion function $w$ typically captures the human tendency to overestimate the probability of rare events and underestimate the probability of more certain ones. Similarly to the utility function, we denote by $w_+$ (resp. $w_-$) the function that warps the cumulative distribution function for gains (resp. for losses). Both functions are required to be defined on $[0, 1]$, with values in $[0, 1]$ and to be non-decreasing, continuous, with $w_+(0) = w_-(0) = 0$ and $w_+(1) = w_-(1) = 1$. Examples of such weights functions in the literature include $w : p \mapsto p^\eta(p^\eta + (1-p)^\eta)^{-\frac{1}{\eta}}$ (Kahneman & Tversky, 1979) and $w : p \mapsto \exp(-(-\ln p)^\eta)$ (Prelec, 1998) where $\eta \in (0, 1)$ is a hyperparameter. We refer the reader to App. I.3 for examples and plots of utility and probability weight functions.

The CPT value of a real-valued random variable $X$ is

$$\mathbb{C}(X) = \int_0^{+\infty} w_+(\mathbb{P}(u^+(X) > z))dz - \int_0^{+\infty} w_-(\mathbb{P}(u^-(X) > z))dz, \tag{1}$$

---

[1]Rewards can be discarded from the history when they are deterministic functions of state-action pairs.

where appropriate integrability assumptions are assumed. While the CPT value $\mathbb{C}(X)$ accounts for the human agent's distortions in perception, it also recovers the expectation $\mathbb{E}(X)$ whenever $X$ is integrable when $w_+$ and $w_-$ are the identity functions and $u^+(x) = x_+, u^-(x) = (-x)_+$. In addition, several risk measures such as variance, Conditional Value at Risk (CVaR) and distortion risk measures are also particular cases of CPT values with *discontinuous* probability weighting functions (see App. I and Table 1 therein).

## 2.2 CPT-RL Problem Formulation

In this work, we will focus on the CPT Policy Optimization (CPT-PO) problem where the objective is the CPT value of the random variable $X = \sum_{t=0}^{H-1} r_t$ recording the cumulative rewards induced by the MDP and the policy $\pi$ for the finite horizon $H \geq 1$:

$$\max_{\pi \in \Pi_{NM}} \mathbb{C}\left[\sum_{t=0}^{H-1} r_t\right]. \tag{CPT-PO}$$

For an illustration of the CPT-RL problem formulation and its elements in a personal treatment for pain management application, see App. G.5. In the particular case of CPT-PO in which $w_+, w_-$ are set to the identity, namely the Expected Utility Policy Optimization (EUT-PO) objective, only returns are distorted by the utility function:

$$\max_{\pi \in \Pi_{NM}} \mathbb{E}\left[u^+\left(\sum_{t=0}^{H-1} r_t\right) - u^-\left(\sum_{t=0}^{H-1} r_t\right)\right]. \tag{EUT-PO}$$

**Challenges.** We outline the main difficulties in solving CPT-PO. First, the CPT value does not satisfy a Bellman equation: the nonlinear utility and weight functions violate the additivity and linearity of the standard expected return. While the special case EUT-PO has been studied (see e.g. Bäuerle & Rieder (2014); Fei et al. (2020); Wu & Xu (2023)), the CPT setting introduces fundamental differences. In particular, probability distortion breaks the dynamic programming structure, and optimal policies may be stochastic even under identity utility. Crucially, this aspect, central to CPT-RL and enabling richer behavioral modeling, has not been addressed in prior work on EUT-PO. Second, CPT-PO is highly nonconvex. The utility function is nonconvex in general (convex over gains, concave over losses), and the probabilities are also distorted by a nonconvex weight function. While the standard policy optimization problem is already nonconvex, CPT-PO compounds this difficulty with additional sources of nonconvexity.

**Scope of this work.** We assume that the utility and weighting functions are known, e.g., from domain experts, surveys, prior empirical studies on target user groups, or behavioral studies (Mkrtchian et al., 2023; George et al., 2019; Sip et al., 2018), see App. I.7 for more details on the choice of the utility function. Our goal is to align the agent's behavior with the given preferences (encoded in utility and weighting functions) by optimizing for the CPT value of returns. Typically, utility and weight functions are chosen as the KT model and hyperparameters of this model are estimated from data. We leave the question of discovering or inferring human preferences (e.g. RL from Human Feedback (RLHF)) to future work.[2] Note that the transition model $\mathcal{P}$ and the reward function $r$ are still supposed to be unknown in our setting. In particular, we do not estimate them as in a model-based approach and our algorithm only uses sampled state-action-reward trajectories.

## 2.3 Peculiarities of Optimal Policies in CPT-RL

In this section, we highlight the properties of optimal policies to CPT-PO when they exist and contrast them with classical RL and EUT-PO (details in App. I.1):

**The need for stochastic policies.** In MDPs, optimal *deterministic* stationary policies always exist; in CPT-RL, optimal policies may need to be stochastic. In general, stochasticity of the policy is essential in solving CPT-PO. See App. C.1 for a proof.

---

[2]See section H for an extended discussion comparing CPT-RL to RLHF as preference learning paradigms, their pros and cons and opportunities for future work in combining them as they are not mutually exclusive. See also App. G for applications.

**Importance of probability weighting.** Under EUT-PO, deterministic non-Markovian policies suffice (see e.g. Theorem 1 in Bäuerle & Rieder (2014)), but this is not the case for CPT-PO with probability weighting in general. Therefore, the need for stochasticity in the optimal policy is clearly due to the probability distortions in the CPT value.

**The need for non-Markovian policies.** Under EUT-PO, optimal *Markovian* policies exist for special cases of utility functions. Bäuerle & Rieder (2014) establish a characterization of such utility functions which turn out to be either affine or exponential (when $\mathcal{U}$ is continuous and increasing). This highlights the role of the (nonlinear) utility functions on the nature of optimal policies. However, this result cannot be extended to CPT-PO in general as we show next.

**Proposition 2.** *There exist instances of CPT-PO whose gain/loss value functions are induced by an exponential value function of the form $x \mapsto A + B\exp(Cx)$, for constants $A, B, C \in \mathbb{R}$, and for which CPT-PO does not admit an optimal policy in $\Pi_{M,NS}$.*

Even exponential utilities, which guarantee the existence of optimal Markovian policies in EUT-PO, may fail to admit Markovian optimal policies in CPT-PO.

**Sufficiency of Markovian policies over a reward-augmented state space.** We conclude this section by showing that, for terminal-return objectives such as (CPT-PO), it is sufficient to optimize over Markov policies on a reward-augmented state space. The augmentation adds an auxiliary variable tracking the accumulated reward along the trajectory.

**Proposition 3** (Reward-augmented Markov policies are sufficient)**.** *Consider a finite-horizon MDP $(\mathcal{S}, \mathcal{A}, \mathcal{P}, r, H, \rho)$ with deterministic reward function $r : \mathcal{S} \times \mathcal{A} \to \mathbb{R}$. Let $Z_t := \sum_{k=0}^{t-1} r(s_k, a_k)$ (with $Z_0 = 0$) denote the accumulated reward before time $t \geq 1$. For any history-dependent randomized policy $\pi_t(\cdot|h_t)$ where $h_t = (s_k, a_k, r_k)_{0 \leq k \leq t-1} \cup \{s_t\}$, there exists a randomized non-stationary Markov policy $\bar{\pi}$ on the augmented state space $\bar{\mathcal{S}} := \mathcal{S} \times \mathbb{R}$ such that $\bar{\pi}_t(a|s, z)$ only depends on $(s, z, t)$ and the terminal accumulated reward $Z_H$ has the same distribution under $\pi$ and $\bar{\pi}$. Therefore, for any functional $C$ that depends only on the distribution of $Z_H$, we have $C(Z_H^\pi) = C(Z_H^{\bar{\pi}})$ where where $Z_H^\pi$ and $Z_H^{\bar{\pi}}$ denote the terminal accumulated rewards induced by $\pi$ and $\bar{\pi}$, respectively. As a consequence, $\sup_{\pi \in \Pi_{NM}} C(Z_H^\pi) = \sup_{\pi \in \Pi_{\Sigma,NS}} C(Z_H^\pi)$.*

**Remark 4.** *Proposition 3 applies to any objective depending only on the distribution of the terminal accumulated reward, including the CPT objective in (CPT-PO). Thus, while Markov policies over the original state space $\mathcal{S}$ (of the form $\pi_t(a|s)$) may be suboptimal, arbitrary history dependence is not required: the accumulated reward $Z_t$, together with the current state and time, is a sufficient statistic. Related reward-augmented reductions have appeared in risk-sensitive RL, including for average-value-at-risk (see e.g., Theorem 3.2 in Bäuerle & Ott (2011)) and more recently for optimized certainty equivalents (see, e.g., Theorem 2.1 in Wang et al. (2025)).*

## 3  Policy Gradient for CPT-value maximization

In this section, we design a policy gradient algorithm for solving CPT-PO. From this section on, we parametrize policies $\pi \in \Pi_{NM}$ by a vector $\theta \in \mathbb{R}^d$ and we denote by $\pi_\theta$ the parametrized policy. As a consequence, the CPT objective in CPT-PO becomes a function of the policy parameter $\theta$ and we use the notation $J(\theta)$ for the corresponding CPT objective value.

### 3.1  CPT Policy Gradient Theorem

Our key result enabling our algorithm design is a PG theorem for CPT value maximization.

**Theorem 5.** *(Policy Gradient for CPT-RL).* *Suppose that the utility functions $u^-, u^+$ are continuous and that the weight functions $w_-, w_+$ are differentiable. Assume in addition that the policy parametrization $\theta \mapsto \pi_\theta(a|h)$ is differentiable and satisfies $\pi_\theta(a|h) > 0$ for any $h, a \in \mathcal{H} \times \mathcal{A}$. Then, for every $\theta \in \mathbb{R}^d$,*

*the gradient of the CPT-PO objective $J$ w.r.t. the policy parameter $\theta$ is given by:*

$$\nabla J(\theta) = \mathbb{E}\left[\phi\left(R(\tau)\right) \sum_{t=0}^{H-1} \nabla_\theta \log \pi_\theta(a_t|h_t)\right],$$

*where $R(\tau) := \sum_{t=0}^{H-1} r_t$ with $\tau := (s_t, a_t, r_t)_{0 \le t \le H-1}$ is a trajectory of length $H$ generated from the MDP by following policy $\pi_\theta$ [3] and for all $v \in \mathbb{R}$,*

$$\phi(v) := \int_0^{u^+(v)} w'_+(\mathbb{P}(u^+(R(\tau')) > z))dz - \int_0^{u^-(v)} w'_-(\mathbb{P}(u^-(R(\tau')) > z))dz, \tag{2}$$

*where $\tau'$ is a random trajectory generated by policy $\pi_\theta$ and $w'_+, w'_-$ denote the derivatives of the functions $w_+, w_-$ respectively.*

We provide a few comments regarding this result:

- Theorem 5 recovers the celebrated policy gradient theorem for standard RL (Sutton et al., 1999) by taking $w_+$ and $w_-$ to be the identity functions (in which case $w'_+$ is the constant function equal to 1) and $u^+(x) = x_+, u^-(x) = (-x)_+$ (where the notation $x_+ = \max\{0, x\}$) which implies that $\phi(R(\tau)) = R(\tau)$.

- In the special case where (i) the CPT utilities are identity functions ($u^+(x) = x_+$ and $u^-(x) = (-x)_+$) and (ii) the gain/loss weighting functions derive from a single distortion function $g$ (i.e. $w_+(t) = 1 - g(1 - t), w_-(t) = g(t)$), the CPT value reduces to a DRM objective (see App. I.4) and Theorem 5 recovers the DRM policy gradient of Vijayan & L.A (2024). CPT objectives are strictly more expressive than distortion risk metrics as they do not impose additional constraints on $u^\pm, w_\pm$. The duality relation between $w_+$ and $w_-$ means that overweighting low probabilities of large losses implies a specific fixed way of transforming probabilities of large gains. As a consequence, behaviors like 'extreme pessimism about rare losses and mild optimism about rare gains' cannot be captured by DRMs. CPT is more expressive as it allows arbitrary $u_+, u_-, w_+, w_-$ with no duality constraint. In particular, it captures reference dependence, loss aversion, unequal tail distortions and asymmetric risk attitudes. CPT uses different utility and probability weight functions for gains and loss regions (possibly asymmetric), allowing for more nuanced behaviors, e.g. risk seeking for gains and risk averse for losses at the same time.

- We stated the theorem in the general setting where the policy is non-Markovian. As shown in Proposition 3, reward-augmented Markovian policies are sufficient to find an optimal policy of the CPT-RL objective over all possible (history-dependent) policies.

**Remark 6** (Infinite-horizon extensions)**.** *The finite-horizon assumption is mainly used to work with a bounded terminal return and to justify the differentiation of the CPT functional under the stated regularity assumptions. An analogous policy-gradient theorem for discounted infinite-horizon returns $X_\gamma = \sum_{t=0}^\infty \gamma^t r_t, \gamma \in (0, 1)$, should hold under bounded rewards and suitable regularity and domination assumptions allowing the exchange of differentiation and integration over the infinite trajectory distribution. The average-reward case is more involved. In that setting, one must first define the CPT objective through a long-run limit, for example via $\lim_{T\to\infty} C\left(\frac{1}{T}\sum_{t=0}^{T-1} r_t\right)$, or through the CPT value of an appropriately defined limiting random variable. Since the CPT functional is nonlinear in the return distribution, it is not immediate that this limit behaves analogously to the standard average-reward criterion, nor that the long-run limit can be interchanged with the CPT integral and policy differentiation. A rigorous treatment would likely require additional ergodicity, mixing, and uniform integrability or domination assumptions. We leave such infinite-horizon extensions to future work.*

---

[3]The integral $\phi(R(\tau))$ is finite under our continuity assumptions since the return $R(\tau)$ is bounded.

---

**Algorithm 1** CPT-Policy Gradient (CPT-PG)

---

1: **Input:** $\theta_0 \in \mathbb{R}^d$, utility functions $u^+, u^-$, weight functions $w_+, w_-$, step sizes $(\alpha_k)$.
2: **for** $k = 0, \cdots, K$ **do**
   / Policy gradient estimation
3:     Sample $m$ trajectories $\tau^\ell := (s_t^\ell, a_t^\ell, r_t^\ell)_{0 \le t \le H-1}$, $1 \le \ell \le m$, with $s_0^\ell \sim \rho$ following $\pi_{\theta_k}$.
    // Quantile estimation
4:     Sample $n$ independent trajectories $\tilde{\tau}^j := (\tilde{s}_t^j, \tilde{a}_t^j, \tilde{r}_t^j)_{0 \le t \le H-1}$, $1 \le j \le n$, with $\tilde{s}_0^j \sim \rho$ following $\pi_{\theta_k}$.
5:     Compute the order statistics of $\{u^\pm(R(\tilde{\tau}^j))\}_{j=1}^n$ and denote them by $Y_{(1)}^\pm \le Y_{(2)}^\pm \le \cdots \le Y_{(n)}^\pm$.
    // Approximation of $\phi(R(\tau^\ell))$
6:     **for** $\ell = 1, \ldots, m$ **do**
7:       For each $\sigma \in \{+, -\}$, set $Y_{(0)}^\sigma := 0$, $v_\ell^\sigma := u^\sigma(R(\tau^\ell))$ and $k_\ell^\sigma := \max\left\{k \in \{0, \ldots, n\} : Y_{(k)}^\sigma \le v_\ell^\sigma\right\}$.
8:       $\widehat{\phi}_n^{\sigma, \ell} = \sum_{k=0}^{k_\ell^\sigma - 1} w_\sigma'\left(\frac{n-k}{n}\right)\left(Y_{(k+1)}^\sigma - Y_{(k)}^\sigma\right) + w_\sigma'\left(\frac{n-k_\ell^\sigma}{n}\right)\left(v_\ell^\sigma - Y_{(k_\ell^\sigma)}^\sigma\right)$.
9:     **end for**
10:    $\widehat{\nabla}_{n,m} J(\theta_k) = \frac{1}{m} \sum_{\ell=1}^m \left(\widehat{\phi}_n^{+,\ell} - \widehat{\phi}_n^{-,\ell}\right) \sum_{t=0}^{H-1} \nabla_\theta \log \pi_{\theta_k}(a_t^\ell | h_t^\ell)$.
   / Policy gradient update
11:    $\theta_{k+1} = \theta_k + \alpha_k \widehat{\nabla}_{n,m} J(\theta_k)$.
12: **end for**

---

## 3.2 Stochastic PG Algorithm for CPT-RL

In the light of Theorem 5, we will perform a policy gradient ascent on the objective $J$ to solve CPT-PO. Our general PG algorithm is presented in Algorithm 1. As usual, since we only have access to sampled trajectories from the MDP, we need a stochastic policy gradient to estimate the true unknown gradient given by the theorem. In particular, we need an approximation of $\phi(R(\tau))$ for any sampled trajectory $\tau$ from the MDP following policy $\pi_\theta$. In the case of EUT-PO in which $w_+$ and $w_-$ are the identity functions, the unknown quantity $\phi(R(\tau))$ reduces to $u^+(R(\tau)) - u^-(R(\tau))$, which can be computed since $u^+$, $u^-$, and $R(\tau)$ are known.

In the more general setting, the approximation task requires to compute the term $\int w_+'(\mathbb{P}(u^+(R(\tau)) > z)dz$ (and likewise for the second integral term). We address this challenge using the following result.

**Proposition 7** (Empirical approximation of the CPT-gradient integral). *Let $X$ be a real-valued random variable and define $Y^\sigma := u^\sigma(X)$ for each sign $\sigma \in \{+, -\}$. Assume that $0 \le Y^\sigma \le M_u$ almost surely and that $w_\sigma'$ is $L_w$-Lipschitz on $[0, 1]$. Let $Y_1^\sigma, \ldots, Y_n^\sigma$ be i.i.d. copies of $Y^\sigma$, and define the empirical function:*

$$\widehat{S}_n^\sigma(z) := \frac{1}{n} \sum_{j=1}^n \mathbf{1}\{Y_j^\sigma > z\}, \qquad z \in \mathbb{R}.$$

*Then,*

$$\sup_{v \in [0, M_u]} \left| \int_0^v w_\sigma'\left(\widehat{S}_n^\sigma(z)\right) dz - \int_0^v w_\sigma'(\mathbb{P}(Y^\sigma > z)) dz \right| \longrightarrow 0 \qquad \textit{almost surely as } n \to \infty.$$

*In particular, for every fixed $v \in [0, M_u]$,*

$$\int_0^v w_\sigma'\left(\widehat{S}_n^\sigma(z)\right) dz \longrightarrow \int_0^v w_\sigma'(\mathbb{P}(Y^\sigma > z)) dz \qquad \textit{almost surely as } n \to \infty.$$

*Moreover, let $Y_{(1)}^\sigma \le \cdots \le Y_{(n)}^\sigma$ denote the order statistics of $Y_1^\sigma, \ldots, Y_n^\sigma$, and set $Y_{(0)}^\sigma := 0$. For any $v \ge 0$, define*

$$k_v^\sigma := \max\left\{k \in \{0, \ldots, n\} : Y_{(k)}^\sigma \le v\right\}.$$

*Then the empirical integral admits the explicit order-statistic form:*

$$\int_0^v w_\sigma'\left(\widehat{S}_n^\sigma(z)\right) dz = \sum_{k=0}^{k_v^\sigma - 1} w_\sigma'\left(\frac{n-k}{n}\right)\left(Y_{(k+1)}^\sigma - Y_{(k)}^\sigma\right) + w_\sigma'\left(\frac{n-k_v^\sigma}{n}\right)\left(v - Y_{(k_v^\sigma)}^\sigma\right).$$

*The same statement holds for both $\sigma = +$ and $\sigma = -$.*

A similar result to Proposition 7 appeared in Proposition 6 in L.A. et al. (2016). We note here though several differences: (a) the integrand is the derivative $w'_+$ (instead of $w_+$) and the integral is taken over a bounded interval; (b) while L.A. et al. (2016) use this result to approximate the CPT value, we use it for approximating our special integral terms involving the derivatives of the weight functions as they appear in the policy gradient. We obtain a different approximation formula which is tailored to our setting. The approximation is essentially a Riemann sum using simple staircase functions. We provide a complete proof of Proposition 7 in Appendix D.2. The proof of the almost sure convergence relies on the Glivenko-Cantelli theorem which shows the convergence of the empirical survival function to the true survival function whereas the finite sum integral formula follows from observing that the empirical survival function is a staircase function which is constant between two successive values of the order-statistics.

Using Proposition 7, we approximate the integral using a finite sum with a given number of samples $n$. Overall, compared to a vanilla PG algorithm, our additional required estimation procedure requires a mild sorting step which can be executed in $\mathcal{O}(n \ln n)$ running time (without even invoking parallel implementations) where $n$ is the length of the rewards to be sorted (see Algorithm 1).

**Comparison to the CPT-SPSA-G algorithm.** Our algorithm is designed for maximizing the CPT value of a sum of rewards generated by an MDP while the CPT Simultaneous Perturbation Stochastic Approximation Gradient (CPT-SPSA-G) algorithm in L.A. et al. (2016) can be used to maximize the CPT value of any real-valued random variable. However, we highlight that (a) this cumulative reward return structure is natural and ubiquitous in RL and economics applications and foremost (b) thanks to this problem structure, our PG algorithm leverages first-order information whereas CPT-SPSA-G only uses zeroth-order information, i.e. CPT value estimations. This difference is crucial as zeroth-order optimization algorithms are known to suffer from the curse of dimensionality. Our algorithm can scale better to higher dimensional problems as it is notoriously known for PG algorithms in classic RL. We provide empirical evidence of this fact in section 5 to further support the benefits of our algorithm.

**Further comparison to Vijayan & L.A (2024).** Our algorithm is not obtained by simply applying utilities to the return samples in the DRM estimator of Vijayan & L.A (2024). The two approaches use different policy gradient representations. Vijayan & L.A (2024) derive a CDF-gradient formula (see Theorem 1 therein) and estimate it by plugging in both an empirical CDF and an empirical CDF-gradient; their Lemma 3 is an order-statistic evaluation of this plug-in integral. In contrast, Theorem 5 gives a different expectation form using a trajectory-level CPT marginal valuation and the score function. Our estimator therefore first estimates the scalar weight $\phi(R(\tau^\ell))$ using order statistics of $u^+(R)$ and $u^-(R)$, and then inserts it into a standard REINFORCE-style Monte Carlo average. Thus Algorithm 1 replaces the raw return in REINFORCE by $\hat{\phi}^+(R(\tau^\ell)) - \hat{\phi}^-(R(\tau^\ell))$. This distinction is visible even in the risk-neutral setting: with identity weights and $u^+(x) = x_+$, $u^-(x) = (-x)_+$, our estimator reduces exactly to vanilla REINFORCE, whereas the Lemma 3 plug-in formula of Vijayan & L.A (2024) telescopes to a score estimator with an endpoint baseline. Thus, even in the DRM setting, the finite-sample estimators are different.

## 4 Convergence and Sample Complexity

In this section, we establish convergence and sample complexity guarantees for Algorithm 1. First, we show that our PG estimator is consistent.

**Proposition 8** (Consistency). *Suppose that the utility functions $u^+$ and $u^-$ are continuous and uniformly bounded by a positive constant $M_u$ on the return range. Assume in addition that the functions $w'_+, w'_-$ are $L_w$-Lipschitz, and that the score function is bounded, i.e. there exists $M_\psi > 0$ s.t. $\|\nabla \ln \pi_\theta(a|s)\|_2 \leq M_\psi$ for all $\theta \in \mathbb{R}^d, (s,a) \in \mathcal{S} \times \mathcal{A}$. Then for any $\theta \in \mathbb{R}^d, \hat{\nabla}_{n,m} J(\theta) \to \nabla J(\theta)$ almost surely as the batch size parameters $n \to \infty$ and $m \to \infty$.*

The proof relies on writing the order-statistic estimator in its empirical-integral form. The Lipschitz continuity of $w'_\pm$ reduces the approximation error to the uniform deviation of the empirical distribution functions, which vanishes almost surely by the Glivenko–Cantelli theorem. The key step is to control the order-statistic

approximation error uniformly over the evaluation trajectories. For each sign $\sigma \in \{+, -\}$, define

$$\phi^\sigma(v) := \int_0^{u^\sigma(v)} w'_\sigma(\mathbb{P}(u^\sigma(R) > z)) \, dz,$$

and let $\widehat{\phi}_n^{\sigma,\ell}$ be the corresponding empirical approximation used in Algorithm 1 with $v_\ell^\sigma := u^\sigma(R(\tau^\ell))$. Since $0 \leq v_\ell^\sigma \leq M_u$, Proposition 8 yields

$$\max_{1 \leq \ell \leq m} \left| \widehat{\phi}_n^{\sigma,\ell} - \phi^\sigma(R(\tau^\ell)) \right| \leq \sup_{v \in [0, M_u]} \left| \int_0^v w'_\sigma\left( \widehat{S}_n^\sigma(z) \right) \, dz - \int_0^v w'_\sigma(\mathbb{P}(u^\sigma(R) > z)) \, dz \right| \longrightarrow 0 \quad \text{a.s. as } n \to \infty.$$

The right-hand side does not depend on $m$, and therefore the approximation error vanishes uniformly over the $m$ evaluation trajectories. Consequently,

$$\max_{1 \leq \ell \leq m} \left| \left( \widehat{\phi}_n^{+,\ell} - \widehat{\phi}_n^{-,\ell} \right) - \phi(R(\tau^\ell)) \right| \longrightarrow 0 \quad \text{a.s. as } n \to \infty.$$

The remaining Monte Carlo term $\frac{1}{m} \sum_{\ell=1}^m \phi(R(\tau^\ell)) \sum_{t=0}^{H-1} \nabla_\theta \log \pi_{\theta_k}(a_t^\ell | h_t^\ell)$ then converges almost surely to the true gradient by the strong law of large numbers as $m \to \infty$. The full proof can be found in Appendix E.1.

Beyond consistency, we now quantify the number of trajectories $(n+m)$ required to obtain an $\varepsilon$-approximate policy gradient for any policy parameter. Under the same assumptions as for the consistency result, we make an additional score boundedness assumption which is standard in the analysis of PG methods (see e.g. Papini et al. (2018); Yuan et al. (2022); Fatkhullin et al. (2023)).

**Proposition 9** (Gradient estimation sample complexity). *Under the assumptions of Prop. 8, there exists a numerical constant $c > 0$ s.t. for any desired precision $\varepsilon > 0$, confidence $\delta \in (0, 1)$ and for all $\theta \in \mathbb{R}^d$, we have $\|\hat{\nabla}_{n,m} J(\theta) - \nabla J(\theta)\|_2 \leq \varepsilon$ with probability at least $1 - \delta$, if $m \geq \frac{16(cHM_\psi M_u \overline{w})^2}{\varepsilon^2} \log\left(\frac{4d}{\delta}\right)$ and $n \geq \frac{8(HM_\psi M_u L_w)^2}{\varepsilon^2} \log\left(\frac{8m}{\delta}\right)$ where $\overline{w} := \max\{\|w'_+\|_\infty, \|w'_-\|_\infty\} < \infty$ and $M_u$ is a uniform bound on the utility functions $u^\pm$ (which exists by continuity of $u^\pm$ and boundedness of the instantaneous rewards).*

The sample complexity result coincides with the classical $n + m = \tilde{\mathcal{O}}(\varepsilon^{-2})$ statistical rate of Monte Carlo estimation. The sample complexity increases with the curvature of the probability weight function, the magnitude of the utility function and the horizon length. The proof relies on using concentration inequality results, namely (i) a Hoeffding's style inequality for bounded vector-valued random variables (Jin et al., 2019) and (ii) Dvoretzky-Kiefer-Wolfowitz inequality to quantify concentration of the empirical distribution of a random variable. Part of the proof is inspired from the proof of Prop. 3 in L.A. et al. (2016) where it is used for CPT value estimation rather than our distorted reward (see Thm. 5 and Prop. 7). Note though that our first-order policy gradient estimator is different and the direct dependence on the dimension $d$ of the policy parameter is only logarithmic. In contrast, the sample complexity of zeroth-order PG estimation in L.A. et al. (2016) scales with the dimension $d$ due to the need to estimate each policy gradient coordinate.

**Remark 10.** *Under only $\alpha$-Hölder continuity of $w'_\pm$ for $\alpha \in (0, 1)$, the sample complexity degrades to $\tilde{\mathcal{O}}(\varepsilon^{-2/\alpha})$. See end of Appendix E.2 for a proof with explicit dependence on the problem constants.*

**Remark 11** (Examples and regularization of probability weighting functions). *The estimator and sample-complexity results above assume that $w'_+$ and $w'_-$ are Lipschitz. This condition is used to control the order-statistic/Riemann-sum approximation of the integral terms appearing in the CPT policy-gradient estimator. It is satisfied by twice continuously differentiable probability distortions with bounded first and second derivatives on $[0, 1]$, such as the identity $w(p) = p$, quadratic distortions $w(p) = p + \lambda p(1-p), \lambda \in [-1, 1]$, and more generally smooth polynomial distortions satisfying $w(0) = 0$, $w(1) = 1$, and $w' \geq 0$. The assumption also holds for endpoint-regularized versions of standard CPT weighting functions. Let $w$ be nondecreasing and strictly increasing on $[\varepsilon, 1 - \varepsilon]$ for some $\varepsilon \in (0, 1/2)$, and define $w_\varepsilon(p) = \frac{w(\varepsilon + (1-2\varepsilon)p) - w(\varepsilon)}{w(1-\varepsilon) - w(\varepsilon)}, p \in [0, 1]$. Then $w_\varepsilon(0) = 0$, $w_\varepsilon(1) = 1$, $w_\varepsilon$ maps $[0, 1]$ into $[0, 1]$, and $w_\varepsilon$ is nondecreasing. If $w$ is twice continuously differentiable on $[\varepsilon, 1 - \varepsilon]$, then $w'_\varepsilon$ is Lipschitz on $[0, 1]$, since $w''_\varepsilon$ is bounded. This applies to endpoint-regularized versions of common CPT weights that are smooth on $(0, 1)$, including the Tversky–Kahneman weighting function $w(p) = \frac{p^\gamma}{(p^\gamma + (1-p)^\gamma)^{1/\gamma}}, \gamma > 0$, the Prelec weighting function $w(p) = \exp(-\beta(-\log p)^\alpha), \alpha, \beta > 0,$*

*and Goldstein–Einhorn-type functions $w(p) = \frac{\delta p^\gamma}{\delta p^\gamma + (1-p)^\gamma}, \delta, \gamma > 0$. The inverse-S parameter regimes often used in CPT correspond to $0 < \gamma < 1$ for the Tversky–Kahneman/Goldstein–Einhorn forms and $0 < \alpha < 1$ for the Prelec form. While the raw versions of these functions may fail to have globally Lipschitz derivatives on $[0, 1]$ because of endpoint singularities, their endpoint-regularized versions satisfy the assumption.*

We now discuss the convergence of the policy parameter sequence produced by Algorithm 1. The CPT-RL objective is nonconvex in the policy parameter due to non-convexity of both utility and probability weighting functions. While this lack of structure makes global optimality out of reach, we can still obtain a standard asymptotic convergence result towards the set of stationary points using the machinery of stochastic approximation (see e.g. Borkar (2008); Benaïm (2006)). We first establish smoothness of the CPT objective $J$ w.r.t. policy parameters. We introduce a few additional assumptions to prove this result.

**Assumption 12** (Smoothness of the policy parametrization and weighting functions)**.** *Assume that for every $h \in \mathcal{H}$, $a \in \mathcal{A}$, and $\theta \in \mathbb{R}^d$, $\pi_\theta(a|h) > 0$, and the map $\theta \mapsto \log \pi_\theta(a|h)$ is twice continuously differentiable. Moreover, assume that there exist $M_\psi, M_{\psi,2} > 0$ s.t. $\|\nabla_\theta \log \pi_\theta(a|h)\|_2 \leq M_\psi, \|\nabla_\theta^2 \log \pi_\theta(a|h)\|_{op} \leq M_{\psi,2}$ for all $\theta, h, a$ where $\|\cdot\|_{op}$ stands for the matrix operator norm. Finally, assume that $w_+$ and $w_-$ are twice continuously differentiable with $\|w'_\pm\|_\infty \leq B_w, \|w''_\pm\|_\infty \leq L_w$.*

**Lemma 13** (Smoothness of the CPT objective)**.** *Suppose that Assumption 12 holds and that the utility functions $u^+$ and $u^-$ are uniformly bounded on the return range: $|u^+(R(\tau))| \leq M_u, |u^-(R(\tau))| \leq M_u$ for every trajectory $\tau$.[4] Then the CPT objective $J$ is $L_J$-smooth for $L_J = 2M_u \left[ L_w H^2 M_\psi^2 + B_w \left( H M_{\psi,2} + H^2 M_\psi^2 \right) \right]$, i.e., for all $\theta, \theta' \in \mathbb{R}^d$, $\|\nabla J(\theta) - \nabla J(\theta')\|_2 \leq L_J \|\theta - \theta'\|_2$.*

Using smoothness of the objective, we prove asymptotic convergence of the iterates of the algorithm to the set of stationary points of the CPT objective $J$ under appropriate step size and batch size conditions.

**Theorem 14** (Asymptotic convergence)**.** *Let Assumption 12 hold. In addition, suppose that the assumptions of Proposition 8 hold and that the iterates $(\theta_k)$ remain bounded. Let the positive step sizes $(\alpha_k)$ satisfy the Robbins–Monro conditions $\sum_{k=0}^\infty \alpha_k = \infty, \sum_{k=0}^\infty \alpha_k^2 < \infty$, and assume that the batch sizes $n_k, m_k$ satisfy $n_k \to +\infty$ and $m_k \to +\infty$. Then the sequence of iterates $(\theta_k)$ generated by Algorithm 1 converges to the set of stationary points of $J$ almost surely, i.e., $\mathrm{dist}(\theta_k, \mathcal{Z}) \to 0$ almost surely as $k \to \infty$, where $\mathcal{Z} := \{\theta \in \mathbb{R}^d : \nabla J(\theta) = 0\}$, and, for any $x \in \mathbb{R}^d$ and nonempty set $\mathcal{A} \subseteq \mathbb{R}^d$, $\mathrm{dist}(x, \mathcal{A}) := \inf_{y \in \mathcal{A}} \|x - y\|_2$.*

The finite-sample CPT-PG estimator is generally biased because of the order-statistic/Riemann-sum approximation of the CPT-gradient integral (see Propositions 7 and 9). For asymptotic convergence to stationary points of the exact CPT objective $J$, we therefore let the batch sizes $n_k, m_k$ (instead of $n, m$ in Algorithm 1) increase to guarantee a vanishing bias. This is analogous to standard stochastic approximation conditions for biased gradient estimators. The key distinction from CPT-SPSA-G is that our method uses first-order likelihood-ratio information rather than zeroth-order finite-difference perturbations. A useful consequence of the first-order CPT-PG estimator is that its finite-sample bias is not divided by a vanishing finite-difference radius. Hence, in our first-order analysis, asymptotic convergence is obtained under the simpler condition $n_k, m_k \to \infty$, rather than a rate condition coupling the batch sizes to a vanishing finite-difference perturbation parameter as in zeroth-order CPT-SPSA analyses. As for our boundedness assumption, it can be simply relaxed by adding a projection in the gradient ascent step in Algorithm 1 and modifying the limit set to account for the projection similarly to the statement of Thm. 1 in L.A. et al. (2016). We prefer our simpler statement. We close this section by discussing the total sample complexity to obtain an approximate first-order stationary policy for the CPT objective, using the gradient estimation sample complexity established in Proposition 9.

**Theorem 15** (Finite-time sample complexity for approximate stationarity)**.** *Let Assumption 12 hold and assume that $J$ is bounded above by $J_{\max}$. Consider Algorithm 1 with constant step size $\alpha \leq \frac{1}{4L_J}$ where $L_J$ is the smoothness constant of $J$ defined in Lemma 13, and suppose that at each iteration the gradient estimator is computed with batch sizes $n, m$. Then, for any $\varepsilon > 0$, $\delta \in (0, 1)$, if $T \geq \frac{4\Delta_0}{\alpha \varepsilon^2}$, and $m \geq \frac{80(cHM_\psi M_u \overline{w})^2}{\varepsilon^2} \log\left(\frac{4dT}{\delta}\right), n \geq \frac{40(HM_\psi M_u L_w)^2}{\varepsilon^2} \log\left(\frac{8mT}{\delta}\right)$, where $\Delta_0 := J_{\max} - J(\theta_0)$ and $c$ is the constant from Proposition 9, then, with probability at least $1 - \delta$, $\frac{1}{T} \sum_{k=0}^{T-1} \|\nabla J(\theta_k)\|_2^2 \leq \varepsilon^2$. In particular,*

---

[4]Note that boundedness of the utilities follows from continuity of these functions and boundedness of the reward function.

$\min_{0 \le k < T} \|\nabla J(\theta_k)\|_2 \le \varepsilon$ *and the total number of sampled trajectories* $N_{\text{tot}} := T(n+m)$ *sufficient to reach an $\varepsilon$-first-order stationary point with probability at least $1 - \delta$ satisfies $N_{\text{tot}} = \tilde{\mathcal{O}}(\varepsilon^{-4})$ where $\tilde{\mathcal{O}}$ hides a polylogarithmic dependence on $1/\varepsilon$ and constants of the problem, specified in Appendix E.5 (see (32)).*

**Remark 16.** *Diminishing step sizes are used for exact asymptotic convergence because the noise must vanish. The finite-time theorem instead targets approximate FOSP guarantees, for which constant step sizes give the standard sharper descent-lemma bound. A decreasing-step-size version is possible, but the bound scales with $1/\sum_{t<T} \alpha_t$; for Robbins–Monro steps $\alpha_t$ of order $t^{-a}$, this gives $\mathcal{O}(T^{-(1-a)})$ for $a < 1$ or $\mathcal{O}(1/\log T)$ for $a = 1$, worse than the constant-step-size $\mathcal{O}(1/T)$ guarantee.*

## 5 Numerical Simulations

While our main contributions are theoretical and methodological, we provide simulations to illustrate our findings. Our main goals in this section are: (a) to show how our CPT-PG algorithm produces policies illustrating more nuances in capturing human behavior compared to standard expected utility theory and risk sensitive RL; (b) to show that, as expected, our algorithm scales better to larger state spaces than existing zeroth-order methods in a grid MDP with increasing size; (c) to test our algorithm on a finance application to show the flexibility of CPT-RL.

**(a) Nuances of CPT-RL.** We test our CPT-PG algorithm and compare it to vanilla PG (vPG) and an exponential risk-sensitive PG (ERS-PG) algorithm on two simple 2-bandit action problem instances. In *Environment 1 (Gain Bandit):* A safe action guarantees a certain gain of 2 and a risky action gives a reward 5 or 0 each with probability 1/2. Clearly the risky action has a higher expected return of 2.5. In *Environment 2 (Loss Bandit):* We flip the signs of the rewards. The safe action guarantees a certain negative reward of -2 whereas the risky action yields either -5 or 0 with probability 1/2 each. In this case, the risky action has a smaller expected return of -2.5.

*Setting:* We train a softmax policy using CPT-PG, vPG and ERS-PG. Vanilla PG optimizes for the standard expected return (identity for $u$ and $w$), CPT-PG using a KT model (as defined in section 2-2.1) with S-shaped utility using parameters ($\alpha = 0.6$, $\lambda = 2.5$) and an S-shaped probability function $w$ under-weighting the probability 1/2. ERS-PG is run using an exponential (concave) risk sensitive utility $u(x) = \eta^{-1}(1 - \exp(-\eta x))$ with $\eta = 0.5$. The reference point is set to be $x_0 = 0$ in this experiment. We use the *exact same* parameters for both environments.

*Results and interpretation:* As shown in Fig. 1 (right), in the Gain Bandit vPG selects the risky arm (higher expected return), whereas both CPT-PG and ERS-PG choose the safe arm—reflecting human risk aversion over gains. In the Loss Bandit, only CPT-PG "flips" to the risky arm, capturing human risk-seeking over losses. Crucially, CPT-PG does this with the *same* S-shaped utility and probability-weighting parameters in *both* settings, whereas the concave ERS-PG objective remains risk-averse throughout. This exact "safe in gains, risky in losses" pattern—known as the reflection effect—is the hallmark of Prospect Theory (Kahneman & Tversky, 1979; Tversky & Kahneman, 1992) from which our simple example is inspired. Neither expected-utility nor a single-parameter risk-sensitive criterion can reproduce both behaviors simultaneously, but CPT-RL can, using a fixed, psychologically-grounded model. While risk-sensitive RL can, with appropriate parameter tuning, replicate some aspects of human decision-making, it often lacks the asymmetric treatment of gains and losses and the probabilistic distortion that characterize human behavior. Overall our main message is that CPT-RL offers more flexibility in modeling. Note also that CPT captures several risk measures as particular cases using discontinuous weights (see App. I.3- I.4).

**(b) Robustness to state space size.** We compare our PG algorithm with the zeroth-order (CPT-SPSA-G) of L.A. et al. (2016) on MDPs with increasing size $n \times n$. Fig. 3 shows that our CPT-PG algorithm scales better to larger grid sizes than CPT-SPSA-G as expected due to its use of (first-order) gradient information. See App. J.5 for details.

**(c) Application to finance.** The goal is to train RL trading agents using our PG algorithm in the CPT-RL setting using a gym trading environment and data from the Bitcoin USD market. See App. J.8 for more details. We test several utility and probability weighting functions including a risk averse exponential of the form $x \mapsto \frac{1}{\beta}(1 - \exp(-\beta x))$ with different values of $\beta$ as well as the KT (Kahneman and Tversky)

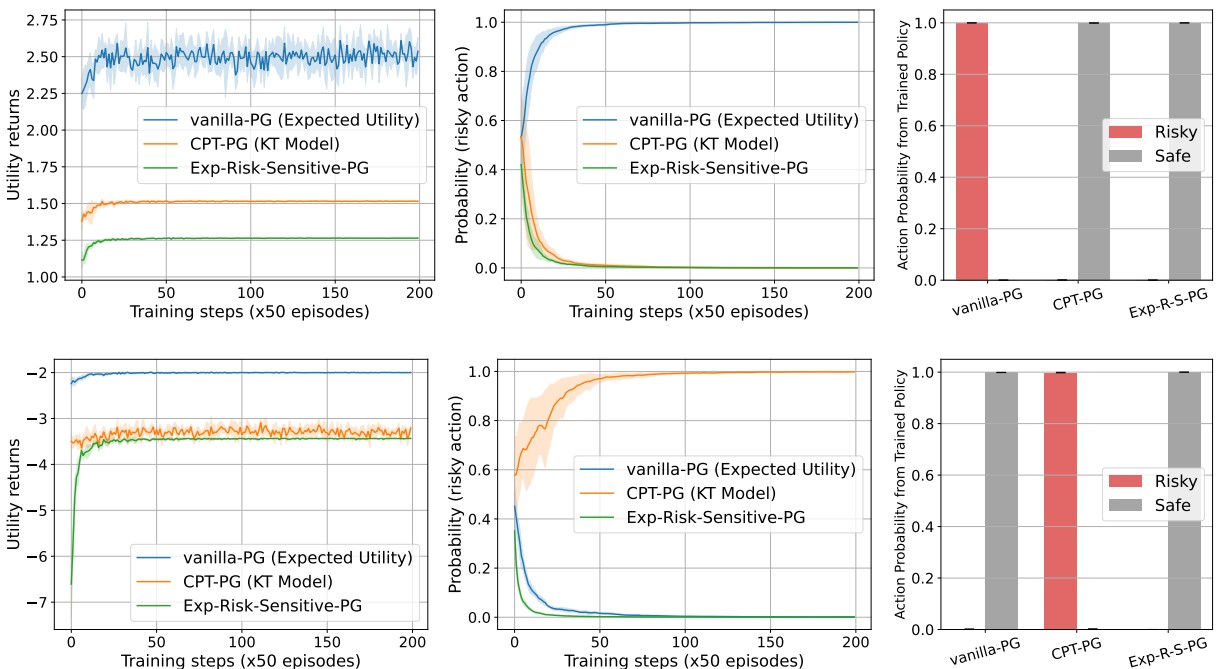

Figure 1: Comparison of our CPT-PG algorithm with vanilla PG (vPG) and exponential risk-sensitive (ERS-PG) on a simple 2-action bandit setting. (Upper fig.) Gain lottery setting: vPG trains a policy picking the risky action whereas CPT-PG and ERS-PG choose the safe one. (Lower fig.) Loss lottery: Only CPT picks the risky action. (Left) Recorded distorted returns, (center) evolution of probability of risky action along training steps, (right) Actions prescribed by trained policies. The shaded area is a range of $\pm$ one standard deviation with 5 independent runs with different seeds.

function with different values of the reference point $x_0$ to illustrate its influence. In Figure 2, we make three observations. First, the reference point shifts the values of the achieved CPT returns: The smaller the reference point, the larger are the returns (Fig. 2, left). This is because only values larger than the reference point are perceived as positive returns. This illustrates how the subjective perception of the agent of the returns is taken into account by the model. Second, different values of $\beta$ lead to different trajectories overall which can translate to different levels of risk aversion. In particular, the curves do not match the identity utility case in the first episodes and show more or less risk taken towards optimizing the CPT returns. Third, the exponent $\alpha$ in the utility distorts the function and shifts the returns significantly (Fig. 2, right). Lower values of $\alpha$ lead to higher returns in this setting. This parameter $\alpha$ provides a degree of freedom to model the behavior of the agent as per their perception of the returns. Different values of $\alpha$ modify the utility function curvature (w.r.t. the reference point $x_0 = 0$ here) which is concave for gains and convex for losses.

**Remark 17. *(Reference point).*** *This point is typically learned from the data in specific human-related applications by personalized tuning. In our experiment (see Fig. 2), we vary this reference point to show how the performance is influenced by this parameter of the model. If the agent has different perceptions of gains and losses with a different reference point, then policy optimization takes this into account.*

**Remark 18. *(History dependent policies and non-Markovianity).*** *In our experiments, we incorporated partial history by expanding the input window of the networks to represent multiple past states. Typically, the input size was chosen to capture sufficient context without encoding the full trajectory length. This limited form of history dependence was sufficient in our settings, although incorporating more expressive temporal models could further enhance performance. The question of scaling to higher dimensional problems deserves further investigation. It is worth noting that the memory requirement is heavily correlated with the temporal structure of the data, especially in time series data like in finance. In our simple simulations,*

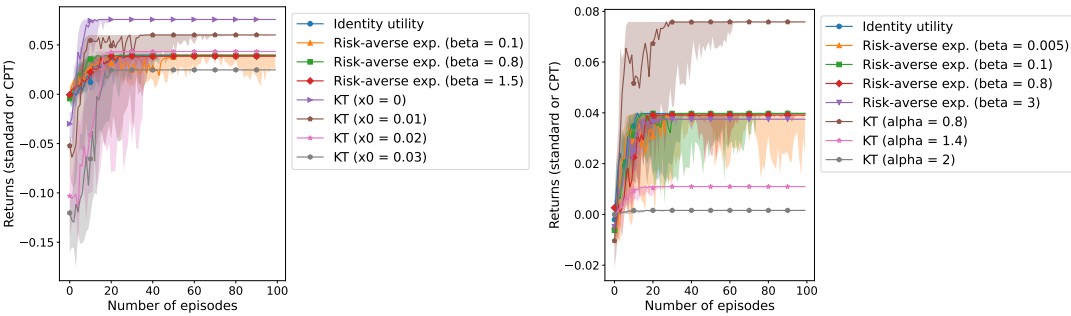

Figure 2: Performance of our PG algorithm on a financial trading application. KT refers to Kahneman and Tversky's utility function, $x_0$ is the reference point used in that utility, exp. refers to exponential and $\alpha$ is the parameter used in the definition of KT's utility. Shaded areas are interquantile (25-75%) margins and curves report the median values over 10 different runs.

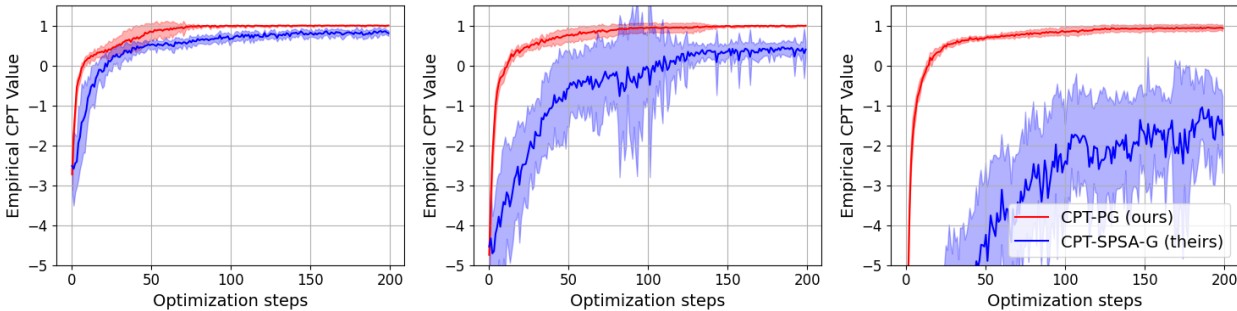

Figure 3: Compared performance of our algorithm and CPT-SPSA-G for $n = 3, 5, 9$. The shaded area is a range of $\pm$ one standard deviation over 10 runs.

*a small expanded input window was enough to obtain a descent performance. Our main goal was to show sensitivity of the KT model to different hyperparameters rather than optimizing performance.*

**Remark 19.** *(Markovian vs non-Markovian policies). In App. J.4 (see Fig. 22), we provide a simple example where a non-Markovian policy performs better than a Markovian one when running CPT-PG. Note here that the Markovian policy does not include the accumulated reward. Hence, this example does not contradict Proposition 3, which shows sufficiency of reward-augmented Markovian policies in $\Pi_{\Sigma,NS}$, but does not claim that ordinary Markovian policies in $\Pi_{M,NS}$ are sufficient.*

**Additional simulations.** We provide more simulations demonstrating the applicability of our CPT-PG algorithm in different settings, including illustrations of our theoretical results (App. J.3-J.4) and applications to a traffic control example over a grid (App. J.6), an electricity management example (App. J.7) and a control application (App. J.9).

## 6 Related Work

Prospect Theory and its sibling, CPT (Kahneman & Tversky, 1979; Tversky & Kahneman, 1992; Barberis, 2013), were first integrated with RL by L.A. et al. (2016). Since then, only a few studies have explored the CPT-RL framework (Borkar & Chandak, 2021; Ramasubramanian et al., 2021; Ethayarajh et al., 2024). Notably, Borkar & Chandak (2021) proposed a Q-learning algorithm for CPT-based policy optimization, while Ramasubramanian et al. (2021) developed value-based methods estimating CPT values using dynamic programming. Their approach optimizes a sum of CPT-transformed period costs, making it amenable to dynamic programming (see remark 1 therein). In contrast, our CPT formulation is different: we maximize the CPT value of the return of a policy CPT-PO. This objective lacks an additive structure, hence does not

satisfy a Bellman equation, rendering dynamic programming approaches inapplicable. Additionally, prior value-based methods are limited to finite state-action spaces, whereas our PG algorithm is also suitable for continuous state action settings, as shown in our experiments. More recently, Ethayarajh et al. (2024) incorporated CPT (without probability distortion) for fine-tuning large language models with human feedback. Our work complements prior CPT-RL studies (L.A. et al., 2016; Jie et al., 2018) that rely on zeroth-order SPSA methods (Spall, 1992). Instead, we introduce a PG algorithm that leverages first-order information, exploiting the structure of CPT values applied to cumulative rewards (see Section 3 for further comparison). Unlike existing PG approaches in risk-sensitive RL, our method explicitly accounts for probability distortion and S-shaped utility transformations, key aspects of CPT. For the special case of DRMs, as previously discussed in more details, Vijayan & L.A (2024) proposed a policy gradient method for maximizing DRM objectives and provided non-asymptotic first-order stationary guarantees. Markowitz et al. (2023) proposed a risk-sensitive policy gradient algorithm. While their objective is CPT-inspired, it is not the same as the standard CPT value of the return studied here and in L.A. et al. (2016). The key distinction is that the utility transformation enters the two objectives in different places. In the standard CPT objective studied here, returns are first transformed into subjective gain/loss variables, and the probability distortions are applied to the survival functions of these transformed variables. Hence, the distorted distributions are those of the perceived gains and losses. By contrast, Markowitz et al. (2023) used a single CDF-weighted objective over raw full-episode returns: the weighting function acts on the CDF/rank of the raw return, while the utility is attached to the return level being integrated. This leads to different gradient estimators: their method produces rank-based weights over ordered returns, whereas our method estimates a CPT marginal valuation using order statistics of the transformed gain/loss variables and then uses a REINFORCE-style score average. The theoretical contributions are also distinct. Markowitz et al. (2023) focused on a practical PPO/KL-regularized risk-sensitive algorithm. We provide a CPT policy-gradient theorem for the standard gain–loss CPT objective, a Monte Carlo estimator tailored to that theorem, consistency and non-asymptotic gradient-estimation bounds, and convergence guarantees for the resulting CPT-PG algorithm.

Recently, Pachal et al. (2025) proposed a (second-order) policy Newton algorithm for distortion riskmetrics and established a non-asymptotic guarantee for convergence to approximate second-order stationary policies. In particular, they proved that their method finds an $\varepsilon$-second-order stationary policy using $\mathcal{O}(\varepsilon^{-3})$ samples for distortion riskmetrics, while we obtain a $\mathcal{O}(\varepsilon^{-4})$ sample complexity to obtain a first-order stationary policy for CPT objectives using a first-order CPT-PG policy gradient algorithm. Their setting covers general distortion riskmetrics, including deviation-type functionals, whereas our work focuses on CPT objectives which encode reference dependence, separate gain/loss utilities, and general asymmetric probability weighting. Our contribution is complementary: we develop a first-order, REINFORCE-style policy gradient method for CPT, without using Hessian information and second-order policy smoothness assumptions. Moreover, our policy gradient representation (Theorem 5) and estimator (Algorithm 1) are not plug-in empirical CDF-gradient estimators: we derive a REINFORCE-style stochastic policy gradient using a trajectory-level CPT marginal valuation. We expect that alternative PG gradient estimators extending the CDF-gradient plug-in approach in prior DRM work may be considered. We leave such investigation and comparison to our distinct estimator for future work.

For a broader discussion on CPT-RL, convex RL, and risk-sensitive RL, see App. B. A diagram summarizing these connections is provided in App. I.2.

## 7 Conclusion

We developed a policy-gradient framework for CPT-based policy optimization in finite-horizon MDPs, including a policy gradient theorem, a Monte Carlo gradient estimator, and convergence guarantees for a first-order CPT-PG algorithm. Our simulations illustrate qualitative behaviors induced by CPT objectives and compare first-order updates to existing zeroth-order approaches.

We conclude by highlighting several opportunities for future work. From the technical viewpoint, our study is restricted to finite state-action spaces, relies on Lipschitzness assumptions on the derivatives of the weighting functions for the estimator analysis, and provides first-order stationarity guarantees. These limitations suggest several directions: extending the theory to continuous state-action spaces; treating endpoint-singular

KT and Prelec weighting functions, beyond their regularized counterparts; and deriving second-order stationarity guarantees using policy Hessian information and additional smoothness assumptions, along the lines of Pachal et al. (2025), which follows a different approach from ours. In light of recent work on gradient domination in robust MDPs (see, e.g., Kitamura et al. (2026) and references therein), another interesting direction is to investigate whether analogous gradient domination conditions can be established for CPT objectives. This appears nontrivial because CPT objectives have a distinct structure: nonlinear probability distortion acts on the terminal-return distribution, but such results could potentially strengthen guarantees to global optimality beyond first-order stationarity. From the modeling viewpoint, a natural direction is to relax the assumption that the utility and probability-distortion functions are known, e.g., by learning or calibrating them from data and studying the resulting statistical and optimization trade-offs. Finally, another direction is to extend CPT-based objectives to multi-agent or social settings, where reference points and probability distortions may interact with strategic incentives.

### Broader Impact Statement

This work develops optimization tools for sequential decision-making objectives inspired by behavioral economics, specifically Cumulative Prospect Theory (CPT), which models probability distortion and asymmetric valuation around a reference point. Such objectives are relevant in settings where decision-makers exhibit systematic deviations from expected-utility assumptions, including human-in-the-loop or preference-driven applications. Our primary contribution is theoretical and methodological: we provide a policy-gradient framework for CPT-RL and demonstrate it in simulation environments drawn from several application themes (e.g., finance, traffic control, and energy) as illustrative case studies.

Potential risks include mis-specification of CPT components (utility and distortion functions), reinforcement of undesirable biases if these components are learned from data, and misuse of behavioral models to manipulate users. Any deployment in real-world, human-facing systems would require careful calibration and validation, transparency about modeling assumptions, and appropriate safeguards to ensure responsible use.

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

## Contents

# A  Notation for Policy Classes

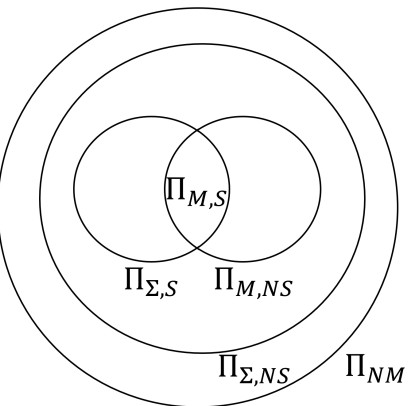

Figure 4: Policy classes (see Rem. 1).

Throughout this work, we will consider the following sets of policies:

- $\Pi_{NM} := \{\mathcal{H} \to \Delta(\mathcal{A})\}$ is the set of non-Markovian policies,[5]

- $\Pi_{\Sigma,NS} := \{\mathcal{S} \times \mathbb{R} \times \mathbb{N} \to \Delta(\mathcal{A})\}$ is the set of policies that only depend on the current state, the timestep and the sum of discounted rewards accumulated so far: The RL agent in state $s$ at timestep $t$ following policy $\pi \in \Pi_{\Sigma,NS}$ samples its next action from the distribution $\pi(s, \sum_{k=0}^{t-1} \gamma^k r_k, t)$,

- $\Pi_{\Sigma,S} := \{\mathcal{S} \times \mathbb{R} \to \Delta(\mathcal{A})\}$ is the set of policies that only depend on the state and the sum of discounted rewards: The RL agent in state $s$ at timestep $t$ following policy $\pi \in \Pi_{\Sigma,S}$ samples its next action from the distribution $\pi(s, \sum_{k=0}^{t-1} \gamma^k r_k)$,

- $\Pi_{M,NS} := \{\mathcal{S} \times \mathbb{N} \to \Delta(\mathcal{A})\}$ is the set of Markovian policies: An agent in state $s$ at timestep $t$ following policy $\pi \in \Pi_{M,NS}$ samples its next action from the distribution $\pi(s, t)$.

- $\Pi_{M,S} := \{\mathcal{S} \to \Delta(\mathcal{A})\}$ is the set of stationary Markovian policies, i.e. Markovian policies which are time-independent.

# B  Extended Related Work Discussion

## B.1  Risk-sensitive RL

There is a rich literature around risk sensitive control and RL that we do not hope to give justice to here. We refer the reader to recent comprehensive surveys on the topic (Garcıa & Fernández, 2015; Prashanth et al., 2022) and the references therein. Let us briefly mention that there exist several approaches to risk sensitive RL. These include formulations such as constrained stochastic optimization to control the tolerance to perturbations and stochastic minmax optimization to model robustness with respect to worst case perturbations for instance. Another approach which is more relevant to our paper discussion consists in regularizing or modifying objective functions. Such modifications are based on considering different statistics of the return deviating from the standard expectation such as the variance or the conditional value at risk (e.g. Tamar et al. (2012); Chow & Ghavamzadeh (2014); Chow et al. (2018)) or even considering the entire distribution of the returns like distributional RL (Bellemare et al., 2023). Another popular objective modification consists in maximizing an exponential criterion (e.g. Borkar (2002); Noorani et al.

---

[5]By 'non-Markovian', we mean '*non necessarily* Markovian' policies including Markovian ones. Elements of $\Pi_{NM} - \Pi_{M,NS}$ can be designated as 'strictly non-Markovian' policies. Likewise, by 'non stationary', we mean 'non necessarily stationary', and by 'stochastic' we mean 'non necessarily deterministic'.

(2022)) to obtain robust policies w.r.t noise and perturbations of system parameters or variations in the environment. Noorani et al. (2022) designed a model-free REINFORCE algorithm and an actor-critic variant of the algorithm leveraging an (approximate) multiplicative Bellman equation induced by the exponential objective criterion. Moharrami et al. (2024) recently proposed and analyzed similar PG algorithms for the same exponential objective. Vijayan & L.A. (2023) introduced a PG algorithm for solving risk-sensitive RL for a class of smooth risk measures including some distortion risk measures and a mean-variance risk measure. Their approach is based on simultaneous perturbation stochastic approximation (SPSA) (Bhatnagar et al., 2013) using zeroth-order information to estimate gradients. Our CPT-PO problem covers several of the aforementioned objectives including smooth distortion risk measures and exponential utility as particular cases (see App. I for more details).

## B.2 Convex RL/RL with General Utilities

In the last few years, convex RL (a.k.a. RL with general utilities) (Hazan et al., 2019; Zhang et al., 2020; Zahavy et al., 2021; Geist et al., 2022) has emerged as a framework to unify several problems of interest such as pure exploration, imitation learning or experiment design. More precisely, this line of research is concerned with maximizing a given functional of the state(-action) occupancy measure w.r.t. a policy. To solve this problem, several policy gradient algorithms have been proposed in the literature (Zhang et al., 2021; Bai et al., 2022; Barakat et al., 2023; 2025). Mutti et al. (2022b;a; 2023) challenged the initial problem formulation and proposed a finite trial version of the problem which is closer to practical concerns as it consists in maximizing a functional of the empirical state(-action) distribution rather than its true asymptotic counterpart. The particular case of our CPT policy optimization problem without probability distortion (see EUT-PO below) coincides with a particular case of the single trial convex RL problem (Mutti et al., 2023) in which the function of the empirical visitation measure is a linear functional of the reward function (see App. I.6 for details). However, our general problem is not a particular case of convex RL which does not account for probability distortions. Furthermore, our utility function is in general nonconvex in our setting (see example in Fig 11) and our policy gradient algorithm is not model-based in the sense that we do not estimate the state transition model. More recently, De Santi et al. (2024) introduced a *global* RL problem formulation where rewards are globally defined over trajectories instead of locally over states and used submodular optimization tools to solve the resulting non-additive policy optimization problem. While global RL allows to account for trajectory-level global rewards, it does not take into consideration probability distortions. In addition, their investigation is restricted to the setting where the transition model is known whereas our PG algorithm does not require to know or estimate the transition model.

## B.3 Cumulative Prospect Theoretic RL

Motivated by Prospect Theory and its sibling CPT (Kahneman & Tversky, 1979; Tversky & Kahneman, 1992; Barberis, 2013), L.A. et al. (2016) first proposed to combine CPT with RL to obtain a better model for human decision making. Following this first research effort, only few isolated works (Borkar & Chandak, 2021; Ramasubramanian et al., 2021; Ethayarajh et al., 2024) considered a similar CPT-RL setting. In particular, Borkar & Chandak (2021) proposed and analyzed a Q-learning algorithm for CPT policy optimization. Ramasubramanian et al. (2021) further developed value-based algorithms for CPT-RL by estimating the CPT value of an action in a given state via dynamic programming. More precisely, they were concerned with maximizing a sum of CPT value period costs which is amenable to dynamic programming. In contrast to their accumulated CPT-based cost (see their remark 1), our CPT policy optimization problem formulation is different: we maximize the CPT value of the return of a policy (see CPT-PO). In particular, this objective does not enjoy an additive structure and hence does not satisfy a Bellman equation. Moreover, their work relying on value-based methods is restricted to finite discrete state action spaces. Our PG algorithm is also suitable for continuous state action settings as we demonstrate in our experiments. More recently, Ethayarajh et al. (2024) incorporated CPT (without probability distortion) into RL from human feedback for fine-tuning large language models. CPT has also been recently exploited for multi-agent RL (Danis et al., 2023). Our work is complementary to this line of research, especially to L.A. et al. (2016) and its extended version (Jie et al., 2018) which are the most closely related work to ours. While their algorithm design makes use of simultaneous perturbation stochastic approximation (SPSA) (Spall, 1992) using only zeroth

order information, we rather propose a PG algorithm exploiting first-order information thanks to our special problem structure involving the CPT value of a cumulative sum of rewards. See section 3 for further details regarding this comparison.

We refer the reader to App. I.2 for a summarizing diagram illustrating the relationships between CPT-RL, convex RL and risk-sensitive RL.

## C  Proofs for Section 2.3

### C.1  The need for stochastic policies

To prove the result (i.e. the need for stochastic policies), we consider a simple MDP with only two states (an initial state and a terminal one) and two actions (A and B). See Fig. 14a below. We choose the identity as utility. Action A yields reward 1 with probability 1 and action B yields either 0 or $\frac{3}{2}$ with probability $\frac{1}{2}$ each. We further consider the following probability distortion function $w_+ : [0,1] \to [0,1]$ defined for every $x \in [0,1]$ as follows:

$$w_+(x) = \begin{cases} 5x & \text{if } x \leq 0.1, \\ \frac{1}{2} + \frac{5}{9}(x - 0.1) & \text{otherwise}, \end{cases} \tag{3}$$

and we set $w_- = 0$. All the policies can be described with a single scalar $p \in [0,1]$, the probability of choosing B instead of A.

The CPT value of the reward $X$ is:

$$\mathbb{C}(X) = w_+\left(1 - \frac{p}{2}\right) + \frac{1}{2}w_+\left(\frac{p}{2}\right). \tag{4}$$

There are only two possible deterministic policies:

- For the policy corresponding to $p = 0$, $\mathbb{C}(X) = 1$.

- For the policy corresponding to $p = 1$, $\mathbb{C}(X) = \frac{3}{2}w_+\left(\frac{1}{2}\right) = \frac{13}{12} \approx 1.08$.

However, with the non-deterministic policy $p = 0.2$, we get:

$$\mathbb{C}(X) = w_+(0.9) + \frac{1}{2}w_+(0.1) = \frac{17}{18} + \frac{1}{4} = \frac{43}{36} \approx 1.19$$

which is larger than the CPT values of both deterministic policies. We conclude that there are no deterministic policies solving the CPT problem in this case.

**Remark 20.** *We provided a counterexample with random rewards, but there also exist counterexamples with deterministic rewards. One way to build such a counterexample is to start from the MDP we just studied and 'transfer' the randomness from the reward functions to the probability transition, by constructing a larger -but equivalent- MDP, with intermediate states like in Fig. 6.*

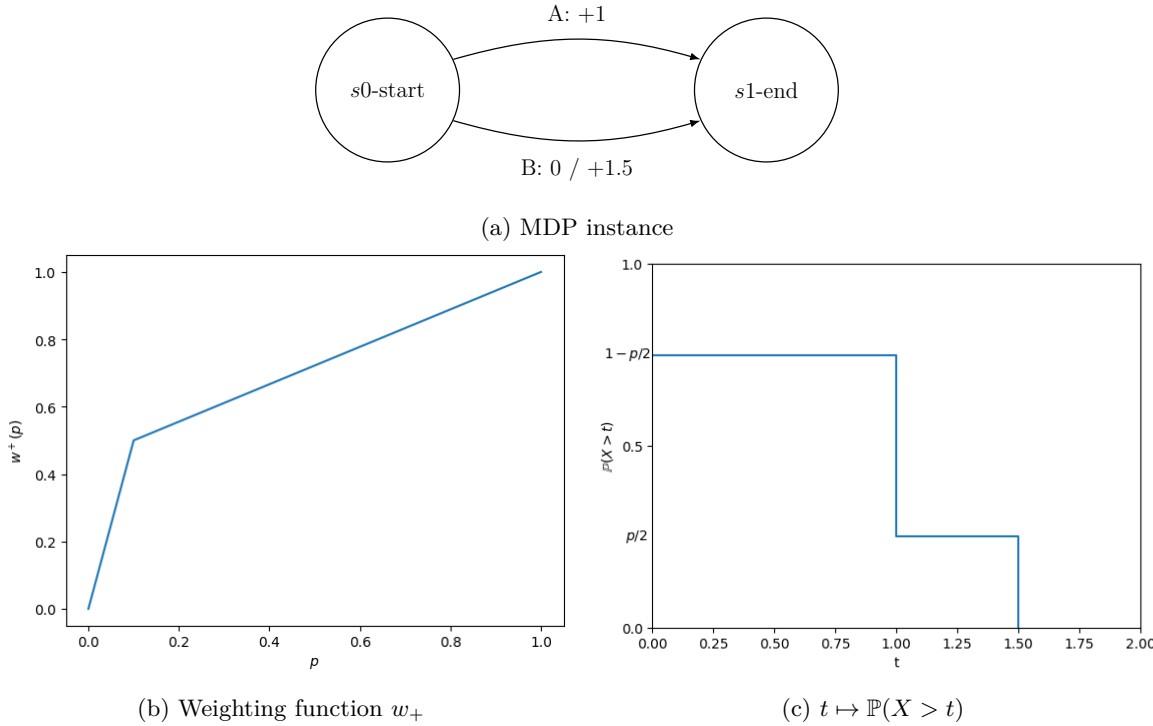

(a) MDP instance

(b) Weighting function $w_+$

(c) $t \mapsto \mathbb{P}(X > t)$

Figure 5: Problem instance for the proof.

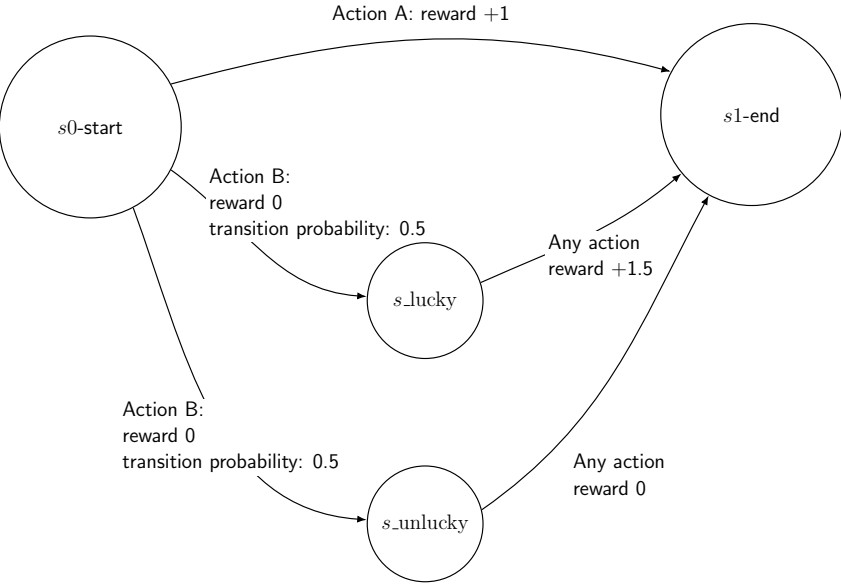

Figure 6: An equivalent example with deterministic rewards

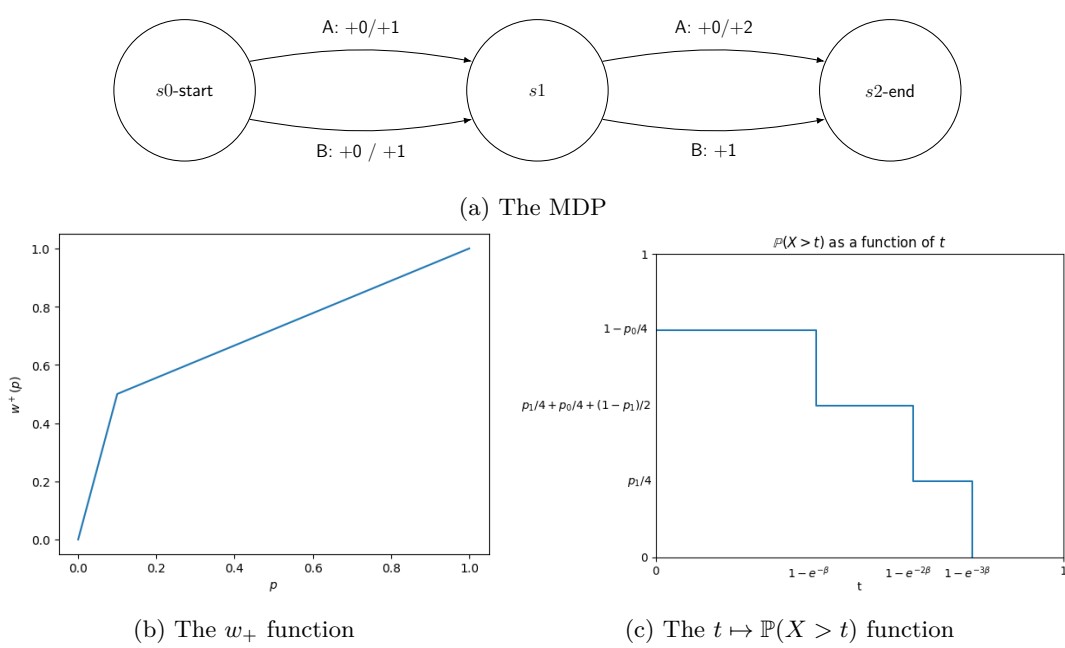

(a) The MDP

(b) The $w_+$ function

(c) The $t \mapsto \mathbb{P}(X > t)$ function

Figure 7: Figures for the proof of Proposition 2

## C.2    Proof of Proposition 2

We proceed by providing a counterexample. We consider the utility function $\mathcal{U} : x \mapsto 1 - \exp(-\beta x)$ with $\beta = \frac{1}{2}$, and the weight function:

$$w_+(x) = \begin{cases} 5x & \text{if } x \leq 0.1, \\ \frac{1}{2} + \frac{5}{9}(x - 0.1) & \text{otherwise.} \end{cases}$$

We also set $w_- = 0$. Our MDP has three states: an initial state $s_0$, an intermediate state $s_1$, and a terminal state $s_2$. There are two actions: A and B. All trajectories start in $s_0$. Any action from $s_0$ leads to $s_1$ with probability 1 and yields reward $+1$ with probability $\frac{1}{2}$ and 0 otherwise. The action taken when in $s_0$ is completely irrelevant. Any action taken in $s_1$ leads to $s_2$ with probability 1 and the episode stops as soon as $s_2$ is reached. When taking action A in $s_1$, the reward is either 0 or $+2$, with probability $\frac{1}{2}$ each. When taking action B in $s_1$, the reward is $+1$ with probability 1. All policies in $\Pi_{NM}$ can be described by $(p_{\text{start}}, p_0, p_1)$, where $p_{\text{start}}$ is the probability of choosing action A when in $s_0$, $p_0$ is the probability of choosing action A in $s_1$ if the transition from $s_0$ to $s_1$ yielded reward 0 and $p_1$ is the probability of choosing action A in $s_1$ if the transition from $s_0$ to $s_1$ yielded reward 1. $p_{\text{start}}$ is irrelevant to the performance of the policy so we can ignore it. The set of Markovian policies here is the set of policies such as $p_0 = p_1$. $\mathbb{C}(\pi)$ is a piecewise affine function of $p_0$ and $p_1$ and it can therefore be directly maximized. We omit the calculations here: one can check that the best achievable CPT value for Markovian policies is $\approx 0.616$ for $p_0 = p_1 = 0.4$ but that a CPT value of $\approx 0.625$ is achievable for $p_0 = 0$ and $p_1 = 0.4$, proving the lemma.

### C.3 Proof of Proposition 3

Let $\pi$ be an arbitrary non-Markovian (history-dependent) policy. Let $Y_t := (s_t, Z_t) \in \bar{\mathcal{S}} = \mathcal{S} \times \mathbb{R}$ be the state-augmented random variable induced by the original MDP under policy $\pi$.

The distribution of $Y_{t+1} = (s_{t+1}, Z_{t+1})$ is entirely determined by $Y_t = (s_t, Z_t)$ and the action $a_t$ selected by the policy $\pi$ given the current history $h_t$, i.e. $Y_t$ is a controlled Markov state. To characterize this distribution, we first define a transition kernel defined on the augmented state space $\bar{\mathcal{S}}$. Given an augmented state $y = (s, z) \in \bar{\mathcal{S}}$ and an action $a \in \mathcal{A}$, the next state is sampled according to the original MDP kernel as $s' \sim P(\cdot \mid s, a)$, and the accumulated reward is updated deterministically as $z' = z + r(s, a)$. Thus, we define the augmented transition kernel $\bar{\mathcal{P}}$ as follows:

$$\bar{\mathcal{P}}(s', dz'|s, z, a) := \mathcal{P}(s'|s, a)\, \delta_{z+r(s,a)}(dz'),$$

where $\delta_x$ denotes the Dirac measure at $x \in \mathbb{R}$, $s, s' \in \mathcal{S}, a \in \mathcal{A}$, and $z \in \mathbb{R}$. Equivalently, for every measurable set $B \subseteq \mathcal{S} \times \mathbb{R}$,

$$\bar{\mathcal{P}}(B \mid s, z, a) = \sum_{s' \in \mathcal{S}} \mathcal{P}(s' \mid s, a)\mathbf{1}\{(s', z + r(s, a)) \in B\}.$$

Note that $\bar{\mathcal{P}}$ is completely determined by the original environment dynamics and reward function and independent of the policy $\pi$ used to choose actions.

We now construct a Markovian policy on the augmented state space $\bar{\mathcal{S}}$ by matching the conditional action distribution induced by the non-Markovian policy $\pi$. Under the original MDP and the policy $\pi$, the pair $(Y_t, a_t)$ has a well-defined joint distribution. Let $\nu_t^\pi(dy) := \mathbb{P}_\pi(Y_t \in dy)$ denote the marginal distribution of $Y_t$, and let $\mu_t^\pi(dy, a) := \mathbb{P}_\pi(Y_t \in dy, \ a_t = a)$ denote the joint distribution of $(Y_t, a_t)$. Since $\mathcal{A}$ is finite and $\bar{\mathcal{S}}$ is a standard Borel space, there exists a regular conditional distribution $q_t^\pi(\cdot \mid y) \in \Delta(\mathcal{A})$ such that, for every measurable set $B \subseteq \bar{\mathcal{S}}$ and every $a \in \mathcal{A}$,

$$\mathbb{P}_\pi(Y_t \in B, \ a_t = a) = \int_B q_t^\pi(a \mid y)\, \nu_t^\pi(dy).$$

Equivalently, $q_t^\pi(\cdot \mid y)$ is a version of the conditional distribution of $a_t$ given $Y_t = y$ under the trajectory distribution generated by $\pi$, i.e. we disintegrate the joint distribution of the tuple $(Y_t, a_t)$ into a product of a distribution of $Y_t$ and a conditional distribution over the action space $\mathcal{A}$ given $Y_t$. We define the Markovian policy $\bar{\pi}$ over the augmented state space $\bar{\mathcal{S}}$ by:

$$\bar{\pi}_t(a|y) := q_t^\pi(a|y),$$

for any $a \in \mathcal{A}, y \in \bar{\mathcal{S}}$. The kernel $q_t^\pi$ is determined only $\nu_t^\pi$-almost surely; on augmented states outside the support of $\nu_t^\pi$, we define $\bar{\pi}_t(\cdot|y)$ arbitrarily (this does not affect the induced distribution of the process). This defines a valid randomized Markovian policy $\bar{\pi}$ on the augmented state space $\bar{\mathcal{S}}$.

In the rest of the proof, we show that the Markovian policy $\bar{\pi}$ over the augmented state space $\bar{\mathcal{S}}$ induces an augmented state process $\bar{Y}_t$ with the same distribution as $Y_t$ generated by the original MDP under the policy $\pi$, i.e. $\bar{Y}_t = (\bar{s}_t, \bar{Z}_t)$ is the augmented process generated by the augmented MDP with transition kernel $\bar{\mathcal{P}}$, policy $\bar{\pi}$, and initial condition $\bar{Y}_0 = Y_0 = (s_0, Z_0 = 0)$ where $s_0 \sim \rho$ and $Z_0 = 0$, then the distributions of $Y_t$ and $\bar{Y}_t$ coincide for any $t = 0, \ldots, H$. The proof proceeds by induction on $t$.

At $t = 0$, both processes have the same initial distribution: $s_0 \sim \rho, Z_0 = 0$. Therefore, the distributions of $Y_0$ and $\bar{Y}_0$ coincide.

Assume now that, for some $t \in \{0, \ldots, H - 1\}$, the distributions of $Y_t$ and $\bar{Y}_t$ coincide. We prove that the distributions of $Y_{t+1}$ and $\bar{Y}_{t+1}$ also coincide.

Let $B \subseteq \bar{\mathcal{S}}$ be any measurable set. Since $Y_t = (s_t, Z_t)$ is a controlled Markov state, the original MDP dynamics and the additive reward recursion imply that:

$$\mathbb{P}_\pi(Y_{t+1} \in B \mid h_t, a_t) = \bar{\mathcal{P}}(B \mid Y_t, a_t).$$

Therefore, it follows that:
$$\mathbb{P}_\pi(Y_{t+1} \in B) = \mathbb{E}_\pi\left[\bar{\mathcal{P}}(B \mid Y_t, a_t)\right].$$

Using the disintegration of the joint law of $(Y_t, a_t)$, we obtain:

$$\mathbb{P}_\pi(Y_{t+1} \in B) = \sum_{a \in \mathcal{A}} \int_{\bar{\mathcal{S}}} \bar{\mathcal{P}}(B \mid y, a) q_t^\pi(a|y) \nu_t^\pi(dy) = \sum_{a \in \mathcal{A}} \int_{\bar{\mathcal{S}}} \bar{\mathcal{P}}(B \mid y, a) \bar{\pi}_t(a|y) \nu_t^\pi(dy).$$

By the induction hypothesis, the distribution $\nu_t^\pi$ of $Y_t$ coincides with the distribution of $\bar{Y}_t$ which we will denote by $\bar{\nu}_t^{\bar{\pi}}$. Hence,

$$\mathbb{P}_\pi(Y_{t+1} \in B) = \sum_{a \in \mathcal{A}} \int_{\bar{\mathcal{S}}} \bar{\mathcal{P}}(B|y, a) \bar{\pi}_t(a|y) \bar{\nu}_t^{\bar{\pi}}(dy).$$

The right-hand side is exactly the one-step law of $\bar{Y}_{t+1}$ under the Markov policy $\bar{\pi}$. Therefore,

$$\mathbb{P}_\pi(Y_{t+1} \in B) = \mathbb{P}_{\bar{\pi}}(\bar{Y}_{t+1} \in B).$$

Since this holds for every measurable set $B$, it follows that $Y_{t+1}$ and $\bar{Y}_{t+1}$ have the same distribution. By induction, we have proved that $Y_{t+1}$ and $\bar{Y}_{t+1}$ have the same distribution for any $t = 0, \dots, H$. In particular, since $Z_H$ and $\bar{Z}_H$ are the second components of $Y_H$ and $\bar{Y}_H$, we have proved that $Z_H$ and $\bar{Z}_H$ have the same distributions. As a consequence, any objective depending only on the distribution of $Z_H$ or $\bar{Z}_H$ (which have the same distribution as we have shown) induces the same policy optimization objectives.

# D  Proofs and Additional Details for Section 3

## D.1  Proof of Theorem 5: CPT Policy Gradient Theorem

The CPT value is a difference between two integrals; see (1). We prove the result for the gain term. The loss term is handled identically and is subtracted at the end.

Let $\mathcal{T}$ denote the set of feasible trajectories of length $H$. Since the state and action spaces are finite and the horizon is finite, the set $\mathcal{T}$ of trajectories is finite. For a trajectory $\tau = (s_0, a_0, \dots, s_{H-1}, a_{H-1}, s_H) \in \mathcal{T}$, write $R(\tau) := \sum_{t=0}^{H-1} r_t$, and $Y^+(\tau) := u^+(R(\tau))$. Let $\rho_\theta(\tau)$ be the trajectory probability induced by $\pi_\theta$:

$$\rho_\theta(\tau) = \rho(s_0) \prod_{t=0}^{H-1} \pi_\theta(a_t|h_t) \, p(s_{t+1}|h_t, a_t). \tag{5}$$

Only the policy terms depend on $\theta$.

We denote the gain part of the CPT objective as follows:

$$J^+(\theta) := \int_0^{+\infty} w_+\big(\mathbb{P}_\theta(Y^+ > z)\big) \, dz, \tag{6}$$

where we highlight the dependence on the policy parameter $\theta$ using the subscript $\theta$ in the probability measure notation $\mathbb{P}_\theta$ in view of the differentiation. Since rewards are bounded and $H < \infty$, $Y^+$ is bounded by a constant $M_+$. Moreover, since $\mathcal{T}$ is finite, $Y^+$ takes only finitely many values. Let

$$0 = y_0 < y_1 < \cdots < y_K = M_+ \tag{7}$$

be the distinct values needed to partition the range of $Y^+$, where $M_+ := \max_{\tau \in \mathcal{T}} Y^+(\tau)$. If $M_+ = 0$, then $J^+(\theta) = 0$ and the gain contribution to the gradient is zero. Hence assume $M_+ > 0$.

For every $k = 1, \dots, K$, define the set of trajectories:

$$\mathcal{T}_k^+ := \{\tau \in \mathcal{T} : Y^+(\tau) \geq y_k\}.$$

Then for every $k = 1, \dots, K$, we show that:

$$\{\tau \in \mathcal{T} : Y^+(\tau) > z\} = \mathcal{T}_k^+, \quad \forall z \in [y_{k-1}, y_k). \tag{8}$$

*Proof of* (8). To show the set identity, we show both inclusions. Since $Y^+(\tau)$ can only take values in $\{y_0, \cdots, y_K\}$, if $Y^+(\tau) > z$ for $z < y_k$, the smallest possible value above $z$ is $y_k$. Hence $Y^+(\tau) \geq y_k$, which means that $\tau \in \mathcal{T}_k^+$. Conversely, if $\tau \in \mathcal{T}_k^+$ for $k = 1, \ldots, K$, then $Y^+(\tau) \geq y_k$ and for $z \in [y_{k-1}, y_k)$, it follows that $Y^+(\tau) > z$, which shows the second inclusion. $\square$

Therefore, we are now ready to show that the gain part of the CPT objective (6) reduces to a finite sum:

$$
\begin{aligned}
J^+(\theta) &= \int_0^{+\infty} w_+\big(\mathbb{P}_\theta(Y^+ > z)\big)\, dz && \text{(by definition, see (6))} \\
&= \int_0^{M_+} w_+\big(\mathbb{P}_\theta(Y^+ > z)\big)\, dz && \text{(by boundedness of } Y^+ \text{ by } M_+) \\
&= \sum_{k=1}^K \int_{y_{k-1}}^{y_k} w_+\big(\mathbb{P}_\theta(Y^+ > z)\big)\, dz && \text{(by decomposition of the integral using (7))} \\
&= \sum_{k=1}^K \int_{y_{k-1}}^{y_k} w_+(\mathbb{P}_\theta(\mathcal{T}_k^+))dz && \text{(using (8))} \\
&= \sum_{k=1}^K (y_k - y_{k-1}) w_+\big(\mathbb{P}_\theta(\mathcal{T}_k^+)\big) && \text{(since } w_+(\mathbb{P}_\theta(\mathcal{T}_k^+)) \text{ is a constant independent of } z). && (9)
\end{aligned}
$$

Note that endpoint values do not affect the integral.

Then, we differentiate the finite sum (9). By the chain rule,

$$
\nabla_\theta J^+(\theta) = \sum_{k=1}^K (y_k - y_{k-1}) w_+'\big(\mathbb{P}_\theta(\mathcal{T}_k^+)\big)\, \nabla_\theta \mathbb{P}_\theta(\mathcal{T}_k^+). \tag{10}
$$

For each $k$, the set $\mathcal{T}_k^+ \subseteq \mathcal{T}$ is fixed. Its probability depends on $\theta$ only through the trajectory distribution $\rho_\theta$. Hence,

$$
\nabla_\theta \mathbb{P}_\theta(\mathcal{T}_k^+) = \nabla_\theta \sum_{\tau \in \mathcal{T}} \rho_\theta(\tau) \mathbf{1}\{\tau \in \mathcal{T}_k^+\} = \sum_{\tau \in \mathcal{T}} \nabla_\theta \rho_\theta(\tau) \mathbf{1}\{\tau \in \mathcal{T}_k^+\} = \sum_{\tau \in \mathcal{T}} \rho_\theta(\tau) \mathbf{1}\{\tau \in \mathcal{T}_k^+\} \nabla_\theta \log \rho_\theta(\tau), \tag{11}
$$

where the second equality is valid because the sum over $\mathcal{T}$ is finite, and the last identity uses the standard log trick. Substituting (11) into (10), we obtain:

$$
\begin{aligned}
\nabla_\theta J^+(\theta) &= \sum_{k=1}^K (y_k - y_{k-1}) w_+'\big(\mathbb{P}_\theta(\mathcal{T}_k^+)\big) \sum_{\tau \in \mathcal{T}} \rho_\theta(\tau) \mathbf{1}\{\tau \in \mathcal{T}_k^+\} \nabla_\theta \log \rho_\theta(\tau) \\
&= \sum_{\tau \in \mathcal{T}} \rho_\theta(\tau) \left[ \sum_{k=1}^K (y_k - y_{k-1}) w_+'\big(\mathbb{P}_\theta(\mathcal{T}_k^+)\big) \mathbf{1}\{\tau \in \mathcal{T}_k^+\} \right] \nabla_\theta \log \rho_\theta(\tau), \tag{12}
\end{aligned}
$$

where the order of summation can be exchanged because all sums are finite.

We now identify the bracketed term by proving the following identity:

$$
\sum_{k=1}^K (y_k - y_{k-1}) w_+'\big(\mathbb{P}_\theta(\mathcal{T}_k^+)\big) \mathbf{1}\{\tau \in \mathcal{T}_k^+\} = \int_0^{Y^+(\tau)} w_+'\big(\mathbb{P}_\theta(Y^+ > z)\big)\, dz. \tag{13}
$$

*Proof of* (13). Fix $\tau \in \mathcal{T}$ and suppose that $Y^+(\tau) = y_i$ for $i \in \{0, \ldots, K\}$. Then, it follows that:

$$
\mathbf{1}\{\tau \in \mathcal{T}_k^+\} = \mathbf{1}\{Y^+(\tau) \geq y_k\} = \mathbf{1}\{k \leq i\}. \tag{14}
$$

Then, we have

$$\int_0^{Y^+(\tau)} w'_+\big(\mathbb{P}_\theta(Y^+ > z)\big)\ dz = \sum_{k=1}^i \int_{y_{k-1}}^{y_k} w'_+\big(\mathbb{P}_\theta(Y^+ > z)\big)\ dz \qquad (Y^+(\tau) = y_i \text{ and partition of } [0, y_i])$$

$$= \sum_{k=1}^i (y_k - y_{k-1}) w'_+\big(\mathbb{P}_\theta(\mathcal{T}_k^+)\big) \qquad (\forall z \in [y_{k-1}, y_k), \mathbb{P}_\theta(Y^+ > z) = \mathbb{P}_\theta(\mathcal{T}_k^+))$$

$$= \sum_{k=1}^K (y_k - y_{k-1}) w'_+\big(\mathbb{P}_\theta(\mathcal{T}_k^+)\big)\, \mathbf{1}\{\tau \in \mathcal{T}_k^+\} \quad \text{(using (14))}. \tag{15}$$

$$\square$$

Combining (12) and (13) yields

$$\nabla_\theta J^+(\theta) = \sum_{\tau \in \mathcal{T}} \rho_\theta(\tau) \left[\int_0^{u^+(R(\tau))} w'_+\big(\mathbb{P}_\theta(u^+(R(\tau')) > z)\big)\ dz\right] \nabla_\theta \log \rho_\theta(\tau)$$

$$= \mathbb{E}_{\tau \sim \rho_\theta}\left[\phi_\theta^+(R(\tau)) \nabla_\theta \log \rho_\theta(\tau)\right], \tag{16}$$

where $\tau'$ denotes an independent trajectory distributed according to $\rho_\theta$, and

$$\phi_\theta^+(v) = \int_0^{u^+(v)} w'_+\big(\mathbb{P}_\theta(u^+(R(\tau')) > z)\big)\ dz.$$

It remains to expand the gradient of the score function using (5) as follows:

$$\log \rho_\theta(\tau) = \log \rho(s_0) + \sum_{t=0}^{H-1} \log \pi_\theta(a_t | h_t) + \sum_{t=0}^{H-1} \log p(s_{t+1} | h_t, a_t), \tag{17}$$

$$\nabla_\theta \log \rho_\theta(\tau) = \sum_{t=0}^{H-1} \nabla_\theta \log \pi_\theta(a_t | h_t), \tag{18}$$

where the last step follows from observing that only the policy terms involve a dependence on the parameter $\theta$ whereas the initial distribution and transition kernel do not depend on $\theta$.

Substituting (18) into (16) gives the desired identity:

$$\nabla_\theta J^+(\theta) = \mathbb{E}_{\tau \sim \rho_\theta}\left[\phi_\theta^+(R(\tau)) \sum_{t=0}^{H-1} \nabla_\theta \log \pi_\theta(a_t | h_t)\right].$$

Repeating the same argument for the loss term in the CPT value:

$$J^-(\theta) := \int_0^{+\infty} w_-\big(\mathbb{P}_\theta(u^-(R(\tau)) > z)\big)\ dz$$

yields:

$$\nabla_\theta J^-(\theta) = \mathbb{E}_{\tau \sim \rho_\theta}\left[\phi_\theta^-(R(\tau)) \sum_{t=0}^{H-1} \nabla_\theta \log \pi_\theta(a_t | h_t)\right],$$

where $\phi_\theta^-(v) := \int_0^{u^-(v)} w'_-(\mathbb{P}_\theta(u^-(R(\tau')) > z))\ dz$. Since the CPT objective is $J(\theta) = J^+(\theta) - J^-(\theta)$, we obtain:

$$\nabla_\theta J(\theta) = \mathbb{E}_{\tau \sim \rho_\theta}\left[\phi_\theta(R(\tau)) \sum_{t=0}^{H-1} \nabla_\theta \log \pi_\theta(a_t | h_t)\right],$$

where $\phi_\theta(v) = \phi_\theta^+(v) - \phi_\theta^-(v)$. This concludes the proof.

### D.2 Proof of Proposition 7: Empirical approximation of the CPT-gradient integral

Fix $\sigma \in \{+, -\}$, and consider the shorthand notation:

$$Y := Y^\sigma = u^\sigma(X), \qquad Y_j := Y_j^\sigma, \qquad \widehat{S}_n := \widehat{S}_n^\sigma, \qquad S(z) := \mathbb{P}(Y > z).$$

We first prove the almost-sure convergence of the empirical integral. For any $v \in [0, M_u]$, by the $L_w$-Lipschitz continuity of $w'_\sigma$, we have:

$$\left| \int_0^v w'_\sigma\left(\widehat{S}_n(z)\right) dz - \int_0^v w'_\sigma(S(z)) \, dz \right| \leq \int_0^v \left| w'_\sigma\left(\widehat{S}_n(z)\right) - w'_\sigma(S(z)) \right| dz$$

$$\leq L_w \int_0^v \left| \widehat{S}_n(z) - S(z) \right| dz$$

$$\leq v L_w \sup_{z \in \mathbb{R}} \left| \widehat{S}_n(z) - S(z) \right|$$

$$\leq M_u L_w \sup_{z \in \mathbb{R}} \left| \widehat{S}_n(z) - S(z) \right|.$$

Taking the supremum over $v \in [0, M_u]$ yields:

$$\sup_{v \in [0, M_u]} \left| \int_0^v w'_\sigma\left(\widehat{S}_n(z)\right) dz - \int_0^v w'_\sigma(S(z)) \, dz \right| \leq M_u L_w \sup_{z \in \mathbb{R}} \left| \widehat{S}_n(z) - S(z) \right|.$$

By the Glivenko–Cantelli theorem,

$$\sup_{z \in \mathbb{R}} \left| \widehat{S}_n(z) - S(z) \right| \to 0 \qquad \text{almost surely as } n \to \infty.$$

Therefore,

$$\sup_{v \in [0, M_u]} \left| \int_0^v w'_\sigma\left(\widehat{S}_n(z)\right) dz - \int_0^v w'_\sigma(S(z)) \, dz \right| \to 0 \qquad \text{almost surely as } n \to \infty.$$

It remains to derive the order-statistic expression. Let $Y_{(1)} \leq \cdots \leq Y_{(n)}$ be the order statistics, and set $Y_{(0)} := 0$. For

$$k_v := \max\{k \in \{0, \ldots, n\} : Y_{(k)} \leq v\},$$

the empirical survival function is constant on each interval $[Y_{(k)}, Y_{(k+1)})$, namely

$$\widehat{S}_n(z) = \frac{n-k}{n}, \qquad z \in [Y_{(k)}, Y_{(k+1)}), \qquad k = 0, \ldots, n-1.$$

Consequently,

$$\int_0^v w'_\sigma\left(\widehat{S}_n(z)\right) dz = \sum_{k=0}^{k_v-1} \int_{Y_{(k)}}^{Y_{(k+1)}} w'_\sigma\left(\widehat{S}_n(z)\right) dz + \int_{Y_{(k_v)}}^v w'_\sigma\left(\widehat{S}_n(z)\right) dz$$

$$= \sum_{k=0}^{k_v-1} w'_\sigma\left(\frac{n-k}{n}\right) \left(Y_{(k+1)} - Y_{(k)}\right) + w'_\sigma\left(\frac{n-k_v}{n}\right) \left(v - Y_{(k_v)}\right).$$

Restoring the superscript $\sigma$ gives the stated formula. Since the argument does not depend on the choice of $\sigma$, the result holds for both $\sigma = +$ and $\sigma = -$.

## E Proofs for Section 4

The proofs in this section use similar techniques as the proofs developed in L.A. et al. (2016). Nevertheless, our policy gradient estimator is different and based on first-order information (using our policy gradient theorem) and this entails some technical differences in the analysis.

### E.1 Proof of Proposition 8: Consistency of the PG estimator

We now prove consistency of the full gradient estimator jointly as $n, m \to \infty$. Recall the CPT policy gradient estimator for any $\theta \in \mathbb{R}^d$:

$$\widehat{\nabla}_{n,m} J(\theta) = \frac{1}{m} \sum_{\ell=1}^{m} \left( \widehat{\phi}_n^{+,\ell} - \widehat{\phi}_n^{-,\ell} \right) S(\tau^\ell), \quad S(\tau^\ell) := \sum_{t=0}^{H-1} \nabla_\theta \log \pi_\theta(a_t^\ell | h_t^\ell).$$

We decompose this estimator as follows:

$$\widehat{\nabla}_{n,m} J(\theta) - \nabla J(\theta) = A_{n,m} + B_m, \tag{19}$$

where $A_{n,m}$ and $B_m$ are defined as follows for any $n, m \geq 1$:

$$A_{n,m} := \frac{1}{m} \sum_{\ell=1}^{m} \left[ \left( \widehat{\phi}_n^{+,\ell} - \widehat{\phi}_n^{-,\ell} \right) - \phi(R(\tau^\ell)) \right] S(\tau^\ell), \quad B_m := \frac{1}{m} \sum_{\ell=1}^{m} \phi(R(\tau^\ell)) S(\tau^\ell) - \mathbb{E}[\phi(R(\tau))S(\tau)].$$

We first control $A_{n,m}$ uniformly in $m$. For any $x \in \mathbb{R}$ and for every $n \geq 1$, define:

$$F^{\pm}(x) := \mathbb{P}(u^{\pm}(R(\tau)) \leq x), \qquad \widehat{F}_n^{\pm}(x) := \frac{1}{n} \sum_{j=1}^{n} \mathbf{1}\{u^{\pm}(R(\tilde{\tau}^j)) \leq x\}.$$

Observe now that:

$$\widehat{\phi}_n^{\pm,\ell} = \int_0^{u^{\pm}(R(\tau^\ell))} w'_{\pm} \left( 1 - \widehat{F}_n^{\pm}(x) \right) dx, \quad \phi^{\pm}(R(\tau^\ell)) := \int_0^{u^{\pm}(R(\tau^\ell))} w'_{\pm} \left( 1 - F^{\pm}(x) \right) dx,$$

where the first identity follows because $\widehat{F}_n^{\pm}$ is a step function whose jumps occur at the order statistics of $\{u^{\pm}(R(\tilde{\tau}^i))\}_{i=1}^n$. On each interval between two consecutive order statistics, $1 - \widehat{F}_n^{\pm}$ is constant, and evaluating the integral over these intervals gives the finite-sum expression in Algorithm 1.

Since $w'_{\pm}$ are $L_w$-Lipschitz and $0 \leq u^{\pm}(R(\tau^\ell)) \leq M_u$, we have, for every $\ell$,

$$\left| \widehat{\phi}_n^{\pm,\ell} - \phi^{\pm}(R(\tau^\ell)) \right| \leq \int_0^{u^{\pm}(R(\tau^\ell))} \left| w'_{\pm} \left( 1 - \widehat{F}_n^{\pm}(x) \right) - w'_{\pm} \left( 1 - F^{\pm}(x) \right) \right| dx$$

$$\leq L_w \int_0^{u^{\pm}(R(\tau^\ell))} \left| \widehat{F}_n^{\pm}(x) - F^{\pm}(x) \right| dx$$

$$\leq M_u L_w \sup_{x \in \mathbb{R}} \left| \widehat{F}_n^{\pm}(x) - F^{\pm}(x) \right|.$$

Therefore,

$$\left| \left( \widehat{\phi}_n^{+,\ell} - \widehat{\phi}_n^{-,\ell} \right) - \phi(R(\tau^\ell)) \right| \leq M_u L_w \left( \sup_x |\widehat{F}_n^{+}(x) - F^{+}(x)| + \sup_x |\widehat{F}_n^{-}(x) - F^{-}(x)| \right).$$

Using the bounded score assumption $\|S(\tau^\ell)\| \leq H M_\psi$, we get

$$\|A_{n,m}\| \leq \frac{1}{m} \sum_{\ell=1}^{m} \left| \left( \widehat{\phi}_n^{+,\ell} - \widehat{\phi}_n^{-,\ell} \right) - \phi(R(\tau^\ell)) \right| \|S(\tau^\ell)\|$$

$$\leq H M_\psi M_u L_w \left( \sup_x |\widehat{F}_n^{+}(x) - F^{+}(x)| + \sup_x |\widehat{F}_n^{-}(x) - F^{-}(x)| \right).$$

The right-hand side does not depend on $m$. By the Glivenko–Cantelli theorem, $\sup_x |\widehat{F}_n^{\pm}(x) - F^{\pm}(x)| \to 0$ a.s. as $n \to \infty$. Hence,

$$A_{n,m} \to 0 \quad \text{a.s. as } n \to \infty,$$

uniformly over $m$.

We now control $B_m$. Since $u^\pm$ are bounded and $w'_\pm$ are continuous on $[0,1]$, $\overline{w} = \max\{\|w'_+\|_\infty, \|w'_-\|_\infty\} < \infty$ and thus $|\phi(R(\tau^\ell))| \leq 2M_u\overline{w}$. Together with $\|S(\tau^\ell)\| \leq HM_\psi$, this implies

$$\|\phi(R(\tau^\ell))S(\tau^\ell)\| \leq 2HM_\psi M_u\overline{w}.$$

Therefore $\phi(R(\tau^\ell))S(\tau^\ell)$ is integrable. By the strong law of large numbers,

$$B_m \to 0 \quad \text{a.s. as } m \to \infty.$$

Combining both a.s. convergence results in (19), we obtain:

$$\widehat{\nabla}_{n,m}J(\theta) - \nabla J(\theta) = A_{n,m} + B_m \to 0 \qquad \text{almost surely as } n, m \to \infty.$$

This proves the consistency of the CPT-PG estimator.

### E.2 Proof of Proposition 9: Sample complexity of the PG estimator

Observe first that our policy gradient estimator can be written as follows:

$$\hat{\nabla}_{n,m}J(\theta) = \frac{1}{m}\sum_{l=1}^{m}Y_{n,l} \cdot X_l, \quad \text{where} \quad Y_{n,l} := \hat{\phi}_n^{+,\ell} - \hat{\phi}_n^{-,\ell}, \quad X_l := \sum_{t=0}^{H-1}\nabla_\theta \log\pi_{\theta_k}(a_t^\ell|h_t^\ell), \qquad (20)$$

see Algorithm 1. Then we can bound the CPT policy gradient estimation error as follows:

$$\|\hat{\nabla}_{n,m}J(\theta) - \nabla J(\theta)\| = \left\|\frac{1}{m}\sum_{\ell=1}^{m}\phi(R(\tau^\ell))X_\ell - \mathbb{E}\left[\phi(R(\tau))\sum_{t=0}^{H-1}\nabla_\theta\log\pi_\theta(a_t|h_t)\right] + \frac{1}{m}\sum_{\ell=1}^{m}(Y_{n,l} - \phi(R(\tau^\ell)))X_\ell\right\|$$

$$\leq \left\|\frac{1}{m}\sum_{\ell=1}^{m}\phi(R(\tau^\ell))X_\ell - \mathbb{E}\left[\phi(R(\tau))\sum_{t=0}^{H-1}\nabla_\theta\log\pi_\theta(a_t|h_t)\right]\right\|$$

$$+ \frac{1}{m}\sum_{\ell=1}^{m}|Y_{n,l} - \phi(R(\tau^\ell)))| \cdot \|X_\ell\|.$$

Let $Z_\ell := \phi(R(\tau^\ell))X_\ell$ for $\ell \in \{1,\ldots,m\}$. Since $\mathbb{E}[Z_1] = \mathbb{E}[Z_\ell] = \nabla J(\theta)$, and $\|X_\ell\| \leq HM_\psi$ for any $\ell \in \{1,\ldots,m\}$ by the bounded score assumption, we obtain:

$$\|\hat{\nabla}_{n,m}J(\theta) - \nabla J(\theta)\| \leq \left\|\frac{1}{m}\sum_{\ell=1}^{m}Z_\ell - \mathbb{E}[Z_1]\right\| + HM_\psi\frac{1}{m}\sum_{\ell=1}^{m}|Y_{n,\ell} - \phi(R(\tau^\ell))|. \qquad (21)$$

Our task now is to control the bound above with high probability by controlling each one of the terms using concentration inequalities. Specifically, we prove the following two statements:

(a) If $m \geq \frac{16(cHM_\psi M_u\overline{\omega})^2}{\varepsilon^2}\log\left(\frac{4d}{\delta}\right)$, then with probability at least $1 - \frac{\delta}{2}$, $\left\|\frac{1}{m}\sum_{\ell=1}^{m}Z_\ell - \mathbb{E}[Z_1]\right\| \leq \frac{\varepsilon}{2}$.

(b) If $n \geq \frac{8(HM_\psi M_u L_w)^2}{\varepsilon^2}\log\left(\frac{8m}{\delta}\right)$, then with probability at least $1 - \frac{\delta}{2}$, $HM_\psi\frac{1}{m}\sum_{\ell=1}^{m}|Y_{n,\ell} - \phi(R(\tau^\ell))| \leq \frac{\varepsilon}{2}$.

Given (21) and using both statements above and a union bound yields the result of Proposition 9, i.e. there exists $c > 0$ s.t. for any $\varepsilon > 0, \delta \in (0,1)$ and for all $\theta \in \mathbb{R}^d$, we have $\|\hat{\nabla}_{n,m}J(\theta) - \nabla J(\theta)\|_2 \leq \varepsilon$ with probability at least $1 - \delta$, if $n \geq \frac{8(HM_\psi M_u L_w)^2}{\varepsilon^2}\log\left(\frac{8m}{\delta}\right)$ and $m \geq \frac{4(cHM_\psi M_u\overline{\omega})^2}{\varepsilon^2}\log\left(\frac{4d}{\delta}\right)$. Now we prove the statements (a) and (b).

**Proof of (a).** Define $M_u := \max_{x \in [-Hr_{\max}, Hr_{\max}]}\{|u^-(x)|, |u^+(x)|\}$ and $\overline{w} := \max\{\|w'_+\|_\infty, \|w'_-\|_\infty\}$. We also use the shorthand notation: $\phi^\pm(R(\tau^\ell)) := \int_0^{u^\pm(R(\tau^\ell))} w'_\pm(\mathbb{P}(u^\pm(R(\tau)) > z))dz$ where $\tau$ is a random trajectory generated by policy $\pi_\theta$. Then the random variable $Z_\ell$ is bounded as follows for any $\ell \in \{1, \ldots, m\}$:

$$\|Z_\ell\| = \|\phi(R(\tau^\ell))X_\ell\| \leq 2HM_\psi M_u \overline{w},$$

where the inequality follows from the score bound $\|X_\ell\| \leq HM_\psi$ and

$$|\phi^\pm(R(\tau^\ell))| \leq \int_0^{u^\pm(R(\tau^\ell))} |w'_\pm(\mathbb{P}(u^\pm(R(\tau)) > z)|dz \leq M_u \overline{w}.$$

Applying Corollary 7 in Jin et al. (2019) which provides a concentration inequality for bounded vector-valued random variables, there exists an absolute constant $c > 0$ such that with probability at least $1 - \frac{\delta}{2}$ (for any $\delta \in (0, 1)$), we have:

$$\left\|\frac{1}{m}\sum_{l=1}^m Z_\ell - \mathbb{E}[Z_1]\right\| \leq 2c\sqrt{\frac{(HM_\psi M_u \overline{w})^2 \log\left(\frac{4d}{\delta}\right)}{m}}.$$

As a consequence, this proves (a), i.e. if $m \geq \frac{16(cHM_\psi M_u \overline{w})^2}{\varepsilon^2}\log\left(\frac{4d}{\delta}\right)$, then with probability at least $1 - \frac{\delta}{2}$, $\left\|\frac{1}{m}\sum_{\ell=1}^m Z_\ell - \mathbb{E}[Z_1]\right\| \leq \frac{\varepsilon}{2}$.

**Proof of (b).** For all $\ell \in \{1, \ldots, m\}$, since $Y_{n,\ell} = \hat{\phi}_n^{+,\ell} - \hat{\phi}_n^{-,\ell}$ and $\phi(R(\tau^\ell)) = \phi^+(R(\tau^\ell)) - \phi^-(R(\tau^\ell))$,

$$\begin{aligned}|Y_{n,\ell} - \phi(R(\tau^\ell))| &= |\hat{\phi}_n^{+,\ell} - \phi^+(R(\tau^\ell)) + \phi^-(R(\tau^\ell)) - \hat{\phi}_n^{-,\ell}| \\ &\leq |\hat{\phi}_n^{+,\ell} - \phi^+(R(\tau^\ell))| + |\phi^-(R(\tau^\ell)) - \hat{\phi}_n^{-,\ell}|.\end{aligned} \tag{22}$$

The rest of the proof follows similar lines to the proof of Proposition 3 in L.A. et al. (2016). Recalling the definition of $\hat{\phi}_n^{+,\ell}$ in Algorithm 1, since $u^+(\tau)$ is supposed to be bounded by $M_u$ and $w'_+$ (note here this is the derivative of $w_+$ unlike in Proposition 3 in L.A. et al. (2016)) is $L_w$-Lipschitz, we have

$$\begin{aligned}|\hat{\phi}_n^{+,\ell} - \phi^+(R(\tau^\ell))| &= \left|\int_0^{u^+(R(\tau^\ell))} w'_+(\mathbb{P}(u^+(R(\tau)) > z)dz - \int_0^{u^+(R(\tau^\ell))} w'_+(1 - \hat{F}_n^+(z))dz\right| \\ &\leq \int_0^{M_u}\left|w'_+(\mathbb{P}(u^+(R(\tau)) > z) - w'_+(1 - \hat{F}_n^+(z))\right|dz \\ &\leq \int_0^{M_u} L_w\left|\mathbb{P}(u^+(R(\tau)) < z) - \hat{F}_n^+(z)\right|dz \\ &\leq L_w M_u \sup_{z \in \mathbb{R}}\left|\mathbb{P}(u^+(R(\tau)) < z) - \hat{F}_n^+(z)\right|,\end{aligned} \tag{23}$$

where $\tau$ is a random variable (trajectory) generated by policy $\pi_\theta$.

Recall now the following classical inequality.

**Lemma 21. (Dvoretzky-Kiefer-Wolfowitz (DKW) inequality)** *Let $\hat{F}_n(u) = \frac{1}{n}\sum_{i=1}^n 1_{\{u(X_i) \leq u\}}$ denote the empirical distribution of a random variable $U$, with $u(X_1), \ldots, u(X_n)$ being sampled from the random variable $u(X)$. Then, for any $n$ and any $\epsilon > 0$, we have*

$$\mathbb{P}\left(\sup_{x \in \mathbb{R}}|\hat{F}_n(x) - F(x)| > \epsilon\right) \leq 2e^{-2n\epsilon^2}. \tag{24}$$

Using the DKW inequality (Lemma 21) yields for any $\ell \in \{1, \ldots, m\}$,

$$\mathbb{P}\left(|\hat{\phi}_n^{+,\ell} - \phi^+(R(\tau^\ell))| > \varepsilon/2\right) \leq \mathbb{P}\left(L_w M_u \sup_{z \in \mathbb{R}}\left|\mathbb{P}(u^+(R(\tau^\ell)) < z) - \hat{F}_n^+(z)\right| > \varepsilon/2\right) \leq 2e^{-n\frac{\varepsilon^2}{2L_w^2 M_u^2}}. \tag{25}$$

A similar inequality holds for $|\hat{\phi}_n^{-,\ell} - \phi^-(R(\tau^\ell))|$.

A union bound over $\ell \in \{1, \ldots, m\}$ yields:

$$\mathbb{P}\left(\bigcap_{\ell=1}^m \left\{|\hat{\phi}_n^{+,\ell} - \phi^+(R(\tau^\ell))| \leq \frac{\varepsilon}{2}\right\}\right) \geq 1 - \sum_{\ell=1}^m \mathbb{P}\left(|\hat{\phi}_n^{+,\ell} - \phi^+(R(\tau^\ell))| > \frac{\varepsilon}{2}\right) \geq 1 - 2m\, e^{-n\frac{\varepsilon^2}{2L_w^2 M_u^2}}\,.$$

Similarly, we have:

$$\mathbb{P}\left(\bigcap_{\ell=1}^m \left\{|\hat{\phi}_n^{-,\ell} - \phi^-(R(\tau^\ell))| \leq \frac{\varepsilon}{2}\right\}\right) \geq 1 - 2m\, e^{-n\frac{\varepsilon^2}{2L_w^2 M_u^2}}\,.$$

As a consequence, it follows from (22) and a union bound that:

$$\mathbb{P}\left(\bigcap_{\ell=1}^m \left\{|Y_{n,\ell} - \phi(R(\tau^\ell))| \leq \varepsilon\right\}\right) \geq 1 - 4m\, e^{-n\frac{\varepsilon^2}{2L_w^2 M_u^2}}\,.$$

Hence, we obtain:

$$\mathbb{P}\left(HM_\psi \frac{1}{m}\sum_{\ell=1}^m |Y_{n,\ell} - \phi(R(\tau^\ell))| \leq \frac{\varepsilon}{2}\right) \geq \mathbb{P}\left(\bigcap_{\ell=1}^m \left\{|Y_{n,\ell} - \phi(R(\tau^\ell))| \leq \frac{\varepsilon}{2HM_\psi}\right\}\right) \geq 1 - 4m\, e^{-n\frac{\varepsilon^2}{8H^2 M_\psi^2 M_u^2 L_w^2}}\,.$$

Therefore, if $n \geq \frac{8(HM_\psi M_u L_w)^2}{\varepsilon^2} \log\left(\frac{8m}{\delta}\right)$, then with probability at least $1 - \frac{\delta}{2}$, we have:

$$HM_\psi \frac{1}{m}\sum_{\ell=1}^m |Y_{n,\ell} - \phi(R(\tau^\ell))| \leq \frac{\varepsilon}{2}\,,$$

which proves statement (b).

*Proof of claim in Remark 10.* Recall first that a function $f : [0,1] \to \mathbb{R}$ is $\alpha$-Hölder continuous with $\alpha \in (0,1]$ if there exists $L_H > 0$ s.t. for all $x, y \in [0,1], |f(x) - f(y)| \leq L_H |x - y|^\alpha$. In Remark 10, we suppose that $w_+'$ and $w_-'$ are $\alpha$-Hölder continuous with constant $L_H$ rather than $L_w$-Lipschitz continuous. Under this assumption, the statement (a) above and its proof remain unchanged and statement (b) is replaced by the following statement:

If $n \geq \frac{(4HM_\psi M_u L_H)^{2/\alpha}}{2\varepsilon^{2/\alpha}} \log\left(\frac{8m}{\delta}\right)$, then with probability at least $1 - \frac{\delta}{2}$, $HM_\psi \frac{1}{m}\sum_{\ell=1}^m |Y_{n,\ell} - \phi(R(\tau^\ell))| \leq \frac{\varepsilon}{2}$.

We now prove the new statement above. The main change in the above proof is inequality (23) which is replaced by the following inequality using $\alpha$-Hölder continuity:

$$|\hat{\phi}_n^{+,\ell} - \phi^+(R(\tau^\ell))| \leq L_H M_u \left(\sup_{0 \leq z \leq M_u} \left|\mathbb{P}(u^+(R(\tau)) < z) - \hat{F}_n^+(z)\right|\right)^\alpha. \tag{26}$$

The rest of the proof follows similar lines, we provide the details for completeness.

Using the DKW inequality (Lemma 21) yields for any $\ell \in \{1, \ldots, m\}$,

$$\mathbb{P}\left(|\hat{\phi}_n^{+,\ell} - \phi^+(R(\tau^\ell))| > \varepsilon/2\right) \leq \mathbb{P}\left(L_H M_u \left(\sup_{0 \leq z \leq M_u} \left|\mathbb{P}(u^+(R(\tau^\ell)) < z) - \hat{F}_n^+(z)\right|\right)^\alpha > \varepsilon/2\right)$$

$$= \mathbb{P}\left(\sup_{0 \leq z \leq M_u} \left|\mathbb{P}(u^+(R(\tau^\ell)) < z) - \hat{F}_n^+(z)\right| > \left(\frac{\varepsilon}{2L_H M_u}\right)^{\frac{1}{\alpha}}\right)$$

$$\leq 2e^{-2n\left(\frac{\varepsilon}{2L_H M_u}\right)^{2/\alpha}}. \tag{27}$$

A similar inequality holds for $|\hat{\phi}_n^{-,\ell} - \phi^-(R(\tau^\ell))|$. Then, a union bound over $\ell \in \{1, \ldots, m\}$ yields:

$$\mathbb{P}\left(\bigcap_{\ell=1}^m \left\{|\hat{\phi}_n^{+,\ell} - \phi^+(R(\tau^\ell))| \leq \frac{\varepsilon}{2}\right\}\right) \geq 1 - \sum_{\ell=1}^m \mathbb{P}\left(|\hat{\phi}_n^{+,\ell} - \phi^+(R(\tau^\ell))| > \frac{\varepsilon}{2}\right) \geq 1 - 2m\, e^{-2n\left(\frac{\varepsilon}{2L_H M_u}\right)^{2/\alpha}}.$$

Similarly, we have:

$$\mathbb{P}\left(\bigcap_{\ell=1}^m \left\{|\hat{\phi}_n^{-,\ell} - \phi^-(R(\tau^\ell))| \leq \frac{\varepsilon}{2}\right\}\right) \geq 1 - 2m\, e^{-2n\left(\frac{\varepsilon}{2L_H M_u}\right)^{2/\alpha}}.$$

As a consequence, it follows from (22) and a union bound that:

$$\mathbb{P}\left(\bigcap_{\ell=1}^m \left\{|Y_{n,\ell} - \phi(R(\tau^\ell))| \leq \varepsilon\right\}\right) \geq 1 - 4m\, e^{-2n\left(\frac{\varepsilon}{2L_H M_u}\right)^{2/\alpha}}.$$

Hence, we obtain:

$$\mathbb{P}\left(H M_\psi \frac{1}{m}\sum_{\ell=1}^m |Y_{n,\ell} - \phi(R(\tau^\ell))| \leq \frac{\varepsilon}{2}\right) \geq \mathbb{P}\left(\bigcap_{\ell=1}^m \left\{|Y_{n,\ell} - \phi(R(\tau^\ell))| \leq \frac{\varepsilon}{2H M_\psi}\right\}\right) \geq 1 - 4m\, e^{-2n\left(\frac{\varepsilon}{4L_H M_u H M_\psi}\right)^{2/\alpha}}.$$

Therefore, if $n \geq \frac{(4H M_\psi M_u L_H)^{2/\alpha}}{2\varepsilon^{2/\alpha}}\log\left(\frac{8m}{\delta}\right)$, then with probability at least $1 - \frac{\delta}{2}$, we have:

$$H M_\psi \frac{1}{m}\sum_{\ell=1}^m |Y_{n,\ell} - \phi(R(\tau^\ell))| \leq \frac{\varepsilon}{2},$$

which concludes the proof of the statement and Remark 10.

$\square$

### E.3 Proof of Lemma 13: Smoothness of the CPT objective

Recall that the CPT objective for $\theta \in \mathbb{R}^d$ is given by:

$$J(\theta) = \int_0^\infty w_+\left(q_z^+(\theta)\right) dz - \int_0^\infty w_-\left(q_z^-(\theta)\right) dz,$$

where the gain-tail probability $q_z^+(\theta)$ and $q_z^-(\theta)$ are defined for any $z \geq 0$ as follows:

$$q_z^+(\theta) := \mathbb{P}_{\rho,\pi_\theta}(u^+(R(\tau)) > z), \qquad q_z^-(\theta) := \mathbb{P}_{\rho,\pi_\theta}(u^-(R(\tau)) > z).$$

Let $\mathfrak{T}_H$ denote the finite set of trajectories of length $H$. For $\tau \in \mathfrak{T}_H$, let $p_\theta(\tau) := \mathbb{P}_{\rho,\pi_\theta}(\tau)$ be the probability of the trajectory $\tau$, and let $R(\tau) := \sum_{t=0}^{H-1} r(s_t, a_t)$ be its return. Since the transition kernel and the reward function do not depend on $\theta$, the dependence of $p_\theta(\tau)$ on $\theta$ only comes from the policy terms and it follows that:

$$\nabla_\theta \log p_\theta(\tau) = \sum_{t=0}^{H-1} \nabla_\theta \log \pi_\theta(a_t|h_t), \quad \nabla_\theta^2 \log p_\theta(\tau) = \sum_{t=0}^{H-1} \nabla_\theta^2 \log \pi_\theta(a_t|h_t).$$

Using Assumption 12, we obtain:

$$\|\nabla_\theta \log p_\theta(\tau)\|_2 \leq H M_\psi, \quad \|\nabla_\theta^2 \log p_\theta(\tau)\|_{\mathrm{op}} \leq H M_{\psi,2}.$$

Since $\nabla_\theta p_\theta(\tau) = p_\theta(\tau)\nabla_\theta \log p_\theta(\tau)$ and

$$\nabla_\theta^2 p_\theta(\tau) = p_\theta(\tau)\left[\nabla_\theta \log p_\theta(\tau)\nabla_\theta \log p_\theta(\tau)^\top + \nabla_\theta^2 \log p_\theta(\tau)\right],$$

we obtain the following bounds:

$$\|\nabla_\theta p_\theta(\tau)\|_2 \le p_\theta(\tau) H M_\psi, \quad \|\nabla_\theta^2 p_\theta(\tau)\|_{\mathrm{op}} \le p_\theta(\tau) \left( H^2 M_\psi^2 + H M_{\psi,2} \right). \tag{28}$$

Then recalling that $q_z^+(\theta) = \mathbb{P}_{\rho,\pi_\theta}(u^+(R(\tau)) > z) = \sum_{\tau \in \mathfrak{T}_H : u^+(R(\tau)) > z} p_\theta(\tau)$ and using (28), we obtain the following bounds:

$$\|\nabla_\theta q_z^+(\theta)\|_2 \le H M_\psi, \quad \|\nabla_\theta^2 q_z^+(\theta)\|_{\mathrm{op}} \le H^2 M_\psi^2 + H M_{\psi,2}.$$

The same bounds hold for $q_z^-(\theta) := \mathbb{P}_{\rho,\pi_\theta}(u^-(R(\tau)) > z)$.

Now using the notation $F_z^+(\theta) := w_+(q_z^+(\theta))$, it follows from the chain rule that:

$$\nabla_\theta^2 F_z^+(\theta) = w_+''(q_z^+(\theta)) \nabla_\theta q_z^+(\theta) \nabla_\theta q_z^+(\theta)^\top + w_+'(q_z^+(\theta)) \nabla_\theta^2 q_z^+(\theta).$$

Using again Assumption 12, we get:

$$\|\nabla_\theta^2 F_z^+(\theta)\|_{\mathrm{op}} \le L_w H^2 M_\psi^2 + B_w \left( H^2 M_\psi^2 + H M_{\psi,2} \right).$$

The same bound holds for $F_z^-(\theta) := w_-(q_z^-(\theta))$.

Since $u^+(R(\tau))$ and $u^-(R(\tau))$ are bounded in absolute value by $M_u$, each CPT integral is over an interval of length at most $M_u$. Hence, integrating the above Hessian bound over the gain and loss parts gives:

$$\|\nabla_\theta^2 J(\theta)\|_{\mathrm{op}} \le 2 M_u \left[ L_w H^2 M_\psi^2 + B_w \left( H^2 M_\psi^2 + H M_{\psi,2} \right) \right].$$

Therefore $J$ is $L_J$-smooth with

$$L_J := 2 M_u \left[ L_w H^2 M_\psi^2 + B_w \left( H M_{\psi,2} + H^2 M_\psi^2 \right) \right].$$

This completes the proof.

### E.4 Proof of Theorem 14: Asymptotic convergence

This result follows from a standard application of the stochastic approximation framework with vanishing bias (see e.g. Borkar (2008); Benaïm (2006)). Under our regularity assumptions on the utility and weight functions, we can show asymptotic convergence of the iterates to the set of stationary points of our CPT objective. The idea is to show that the iterates track a gradient flow defined by the gradient of the CPT-PO objective with vanishing step sizes. The proof follows similar steps as the proof of Theorem 1 in L.A. et al. (2016) (see App. D, p. 25 therein), up to the difference that our policy gradients are not estimated using zeroth-order information but rather using our PG Thm. 5 and Prop. 7. Let $G_k := \widehat{\nabla}_{n_k,m_k} J(\theta_k)$ denote the CPT-PG estimator used at iteration $k$ with batch sizes $n_k$ and $m_k$ rather than constants $n, m$ in Algorithm 1. Let $\mathcal{F}_k$ be the filtration generated by the iterates and all samples up to iteration $k$. We decompose the estimator as follows:

$$G_k = \nabla J(\theta_k) + b_k + M_{k+1},$$

where $b_k := \mathbb{E}[G_k \mid \mathcal{F}_k] - \nabla J(\theta_k)$ is the conditional bias and $M_{k+1} := G_k - \mathbb{E}[G_k \mid \mathcal{F}_k]$ is a martingale-difference noise term (s.t. $\mathbb{E}[M_{k+1} \mid \mathcal{F}_k] = 0$). The update can therefore be written as follows:

$$\theta_{k+1} = \theta_k + \alpha_k \left( \nabla J(\theta_k) + b_k + M_{k+1} \right).$$

We now verify that the perturbation terms satisfy the standard stochastic approximation conditions.

**Martingale noise bound.** First, the martingale noise has bounded second moment. Indeed, by the bounded score assumption, $\left\| \sum_{t=0}^{H-1} \nabla_\theta \log \pi_\theta(a_t \mid h_t) \right\| \le H M_\psi$. Moreover, since $u^+$ and $u^-$ are bounded by $M_u$, and since $w_+'$ and $w_-'$ are continuous on the compact interval $[0, 1]$, we can define

$\overline{w} := \max\{\|w'_+\|_\infty, \|w'_-\|_\infty\} < \infty$. Both the true distorted coefficient and its empirical order-statistic approximation are therefore uniformly bounded by $2M_u\overline{w}$. We prove this bound in the following. We first bound the empirical CPT coefficient. For the gain term, let

$$\widehat{S}_{n,+}(z) := \frac{1}{n} \sum_{i=1}^{n} \mathbf{1}\{u^+(R(\tilde{\tau}^i)) > z\},$$

computed from the $n$ trajectories used for the order-statistic approximation (see Algorithm 1, steps 4 and 5). The empirical order-statistic approximation appearing in the algorithm can equivalently be written as:

$$\widehat{\phi}_n^{+,l} = \int_0^{u^+(R(\tau^\ell))} w'_+\left(\widehat{S}_{n,+}(z)\right) dz, \qquad 1 \le \ell \le m.$$

Since $\|w'_+\|_\infty \le \overline{w}$ and $\overline{w} < +\infty$ (as $\widehat{S}_{n,+}(z) \in [0,1]$ for every $z$), we have:

$$\left|\widehat{\phi}_n^{+,\ell}\right| \le \int_0^v \left|w'_+\left(\widehat{S}_{n,+}(z)\right)\right| dz \le \overline{w}M_u.$$

The same argument for the loss term gives $\left|\widehat{\phi}_n^{-,l}\right| \le \overline{w}M_u$. Therefore, uniformly over $n$, over the sampled trajectories, and over $\ell \in \{1, \ldots, m\}$,

$$\left|\widehat{\phi}_n^{+,l} - \widehat{\phi}_n^{-,l}\right| \le 2\overline{w}M_u.$$

Combining the above bound with the bounded score assumption yields:

$$\|G_k\| \le \frac{1}{m} \sum_{\ell=1}^m \left|\widehat{\phi}_n^+ - \widehat{\phi}_n^-\right| \cdot \sum_{t=0}^{H-1} \left\|\nabla_\theta \log \pi_{\theta_k}(a_t^\ell | h_t^\ell)\right\| \le 2\overline{w}M_uHM_\psi.$$

We have proved that there exists a positive constant $G_{\max} := 2\overline{w}M_uHM_\psi < \infty$, independent of $k, n_k, m_k$, such that $\|G_k\| \le G_{\max}$ a.s. Consequently, $\|M_{k+1}\| = \|G_k - \mathbb{E}[G_k \mid \mathcal{F}_k]\| \le 2G_{\max}$ a.s., and thus $\mathbb{E}\left[\|M_{k+1}\|^2 \mid \mathcal{F}_k\right] \le 4G_{\max}^2$ a.s.

**Vanishing bias.** Second, we control the bias $(b_k)$. The finite-sample CPT-PG estimator is generally biased because of the order-statistic/Riemann-sum approximation of the CPT-gradient integral. However, the same high-probability bounds used in the proof of the gradient estimation sample complexity result (Proposition 9) imply the following bound in an expectation: there exists a constant $C > 0$ such that, for every $\theta$,

$$\mathbb{E}\left[\left\|\widehat{\nabla}_{n,m}J(\theta) - \nabla J(\theta)\right\|\right] \le C\left(\frac{1}{\sqrt{n}} + \frac{\sqrt{\log d}}{\sqrt{m}}\right). \tag{29}$$

We prove the above inequality by a tail integration argument converting the high-probability guarantee into a guarantee in expectation.

Set $X := \left\|\widehat{\nabla}_{n,m}J(\theta) - \nabla J(\theta)\right\|$ for some $\theta \in \mathbb{R}^d$ where the stochastic policy gradient $\widehat{\nabla}_{n,m}J(\theta)$ is computed as in Algorithm 1, define $A_n := \frac{\sqrt{2}HM_\psi M_uL_w}{\sqrt{n}}$ and $B_m := \frac{cHM_\psi M_u\overline{w}}{\sqrt{m}}$.

Then we have already proved in Proposition 9 that with probability at least $1 - \delta$ (for any $\delta \in (0,1)$),

$$X \le A_n\sqrt{\log\left(\frac{2}{\delta}\right)} + B_m\sqrt{\log\left(\frac{2d}{\delta}\right)}.$$

It follows that with probability at least $1 - \delta$,

$$X \le A_n\sqrt{\log 2 + \log\left(\frac{1}{\delta}\right)} + B_m\sqrt{\log(2d) + \log\left(\frac{1}{\delta}\right)} \le \left(A_n\sqrt{\log 2} + B_m\sqrt{\log(2d)}\right) + (A_n + B_m)\sqrt{\log\left(\frac{1}{\delta}\right)}.$$

Setting $Z := \max\left\{X - \left(A_n\sqrt{\log 2} + B_m\sqrt{\log(2d)}\right), 0\right\}$, we have for any $\delta \in (0,1)$:

$$\mathbb{P}\left(Z > (A_n + B_m)\sqrt{\log\left(\frac{1}{\delta}\right)}\right) \leq \delta\,.$$

Equivalently, for all $t \geq 0$,

$$\mathbb{P}(Z > t) \leq \exp\left(-\frac{t^2}{(A_n + B_m)^2}\right)\,.$$

Therefore, by integrating the tail, we obtain (noting that $Z \geq 0$),

$$\mathbb{E}[Z] = \int_0^{+\infty} \mathbb{P}(Z > t)dt \leq \int_0^{+\infty} \exp\left(-\frac{t^2}{(A_n + B_m)^2}\right)dt = \frac{\sqrt{\pi}}{2}(A_n + B_m)\,. \tag{30}$$

We have proved that $\mathbb{E}[Z] \leq \frac{\sqrt{\pi}}{2}(A_n + B_m)$ which implies that:

$$\mathbb{E}[X] \leq \left(A_n\sqrt{\log 2} + B_m\sqrt{\log(2d)}\right) + \frac{\sqrt{\pi}}{2}(A_n + B_m)\,.$$

Substituting the expressions of $A_n$ and $B_m$, we obtain:

$$\mathbb{E}\left[\left\|\widehat{\nabla}_{n,m}J(\theta) - \nabla J(\theta)\right\|\right] \leq \left(\sqrt{\log 2} + \frac{\sqrt{\pi}}{2}\right)A_n + \left(\sqrt{\log(2d)} + \frac{\sqrt{\pi}}{2}\right)B_m$$

$$= \left(\sqrt{\log 2} + \frac{\sqrt{\pi}}{2}\right)\frac{\sqrt{2}HM_\psi M_u L_w}{\sqrt{n}} + \left(\sqrt{\log(2d)} + \frac{\sqrt{\pi}}{2}\right)\frac{cHM_\psi M_u \overline{w}}{\sqrt{m}}\,,$$

which proves (29). Applying this bound conditionally on $\mathcal{F}_k$, with $\theta = \theta_k$, gives:

$$\mathbb{E}\left[\|G_k - \nabla J(\theta_k)\|\,|\,\mathcal{F}_k\right] \leq C\left(\frac{1}{\sqrt{n_k}} + \frac{\sqrt{\log d}}{\sqrt{m_k}}\right)\,,$$

where $C$ is some constant independent of $n_k$ and $m_k$. Therefore, using Jensen's inequality, we obtain:

$$\|b_k\| = \|\mathbb{E}[G_k\,|\,\mathcal{F}_k] - \nabla J(\theta_k)\| \leq \mathbb{E}\left[\|G_k - \nabla J(\theta_k)\|\,|\,\mathcal{F}_k\right] \leq C\left(\frac{1}{\sqrt{n_k}} + \frac{\sqrt{\log d}}{\sqrt{m_k}}\right)\,,$$

where the last inequality holds almost surely. Since $n_k \to +\infty$ and $m_k \to +\infty$, it follows that $b_k \to 0$ a.s.

We have all the conditions required to apply the standard ODE method of stochastic approximation with the (standard) gradient flow $\dot{\theta}(t) = \nabla J(\theta(t))$ using the function $J$ as a strict Lyapunov function for this flow (see e.g. Theorem 2, Corollary 3 and extensions of section 2.2, chapter 2 in Borkar (2008) or Proposition 4.1, Remark 4.5 and Proposition 6.4 in Benaïm (2006)).

### E.5 Proof of Theorem 15: Total sample complexity for approximate first-order stationarity

The proof mainly follows from standard analysis of smooth nonconvex optimization with inexact gradients. We provide a full proof here for completeness.

Let $g_k := \widehat{\nabla}_{n,m}J(\theta_k)$ as per Algorithm 1 with constant step size $\alpha > 0$. By $L_J$-smoothness (Lemma 13),

$$J(\theta_{k+1}) \geq J(\theta_k) + \alpha\langle\nabla J(\theta_k), g_k\rangle - \frac{L_J\alpha^2}{2}\|g_k\|_2^2. \tag{31}$$

Using the shorthand notation $e_k := g_k - \nabla J(\theta_k)$, we further have:

$$\langle\nabla J(\theta_k), e_k\rangle \geq -\frac{1}{4}\|\nabla J(\theta_k)\|_2^2 - \|e_k\|_2^2, \quad \|g_k\|_2^2 \leq 2\|\nabla J(\theta_k)\|_2^2 + 2\|e_k\|_2^2.$$

Using the above inequalities in (31), we obtain:

$$J(\theta_{k+1}) \geq J(\theta_k) + \left(\frac{3\alpha}{4} - L_J\alpha^2\right)\|\nabla J(\theta_k)\|_2^2 - \left(\alpha + L_J\alpha^2\right)\|e_k\|_2^2.$$

Since $\alpha \leq 1/(4L_J)$, we have $\frac{3\alpha}{4} - L_J\alpha^2 \geq \frac{\alpha}{2}$ and $\alpha + L_J\alpha^2 \leq \frac{5\alpha}{4}$. Together with $\|e_k\|_2 \leq \varepsilon_g$, we get

$$J(\theta_{k+1}) \geq J(\theta_k) + \frac{\alpha}{2}\|\nabla J(\theta_k)\|_2^2 - \frac{5\alpha}{4}\varepsilon_g^2.$$

Rearranging and summing from $k = 0$ to $T - 1$ yields:

$$\frac{1}{T}\sum_{k=0}^{T-1}\|\nabla J(\theta_k)\|_2^2 \leq \frac{2(J(\theta_T) - J(\theta_0))}{\alpha T} + \frac{5}{2}\varepsilon_g^2 \leq \frac{2\Delta_0}{\alpha T} + \frac{5}{2}\varepsilon_g^2,$$

where the last inequality uses $J(\theta_T) \leq J_{\max}$. The same upper bound holds for $\min_{0 \leq k < T}\|\nabla J(\theta_k)\|_2^2$. Choosing $\varepsilon_g = \varepsilon/\sqrt{5}$ and $T \geq \frac{4\Delta_0}{\alpha\varepsilon^2}$ yields:

$$\min_{0 \leq k < T}\|\nabla J(\theta_k)\|_2^2 \leq \varepsilon^2.$$

It remains to account for the number of sampled trajectories required to ensure $\|g_k - \nabla J(\theta_k)\|_2 \leq \varepsilon_g$ uniformly over $k = 0, \ldots, T - 1$. We ensure that the gradient-estimation error is uniformly controlled along the optimization trajectory. For each iteration $k = 0, \ldots, T - 1$, define the event

$$\mathcal{E}_k := \{\|g_k - \nabla J(\theta_k)\|_2 \leq \varepsilon_g\}.$$

Although $\theta_k$ is random, it is measurable with respect to the past randomness before drawing the fresh trajectories used to construct $g_k$. Therefore, Proposition 9 can be applied conditionally on the past. With failure probability $\delta/T$, we have $\mathbb{P}(\mathcal{E}_k^c \mid \theta_k) \leq \frac{\delta}{T}$, provided that $m \geq \frac{16(cHM_\psi M_u\overline{\omega})^2}{\varepsilon_g^2}\log\left(\frac{4dT}{\delta}\right)$ and $n \geq \frac{8(HM_\psi M_u L_w)^2}{\varepsilon_g^2}\log\left(\frac{8mT}{\delta}\right)$. Taking expectation over $\theta_k$ yields $\mathbb{P}(\mathcal{E}_k^c) \leq \frac{\delta}{T}$. Then, a union bound over the $T$ iterations gives the uniform gradient estimation error,

$$\mathbb{P}\left(\bigcup_{k=0}^{T-1}\mathcal{E}_k^c\right) \leq \sum_{k=0}^{T-1}\mathbb{P}(\mathcal{E}_k^c) \leq T \cdot \frac{\delta}{T} = \delta.$$

Equivalently, $\mathbb{P}\left(\bigcap_{k=0}^{T-1}\mathcal{E}_k\right) \geq 1 - \delta$. Thus, with probability at least $1 - \delta$,

$$\|g_k - \nabla J(\theta_k)\|_2 \leq \varepsilon_g, \qquad \text{for all } k = 0, \ldots, T - 1.$$

Substituting $\varepsilon_g = \varepsilon/\sqrt{5}$ and $T = 4L_J\Delta_0/\varepsilon^2$ gives the stated total trajectory complexity:

$$N_{\text{tot}} = T(n + m) \geq \frac{4L_J\Delta_0}{\varepsilon^2} \cdot \left(\frac{40(HM_\psi M_u L_w)^2}{\varepsilon^2}\log\left(\frac{8mT}{\delta}\right) + \frac{80(cHM_\psi M_u\overline{w})^2}{\varepsilon^2}\log\left(\frac{4dT}{\delta}\right)\right). \tag{32}$$

## F  CPT-RL vs Risk-sensitive RL: An Illustration

In this section, we illustrate key features of CPT-RL compared to existing risk-sensitive RL approaches using a simple environment. This serves to provide intuition through an easy-to-grasp example. Specifically, we highlight how CPT-RL inflates low-probability (high-risk) events, distinguishing it from other methods. For comparison, we focus on an exponential risk-sensitive RL policy gradient algorithm, though similar results can also be shown for other risk-sensitive measures.

**RiskyGridworld environment and reward structure.** We consider a $5 \times 5$ custom gridworld environment The agent starts at $(0,0)$ and must navigate to the goal state $(4,4)$, choosing between safe and risky paths.

- States: each cell represents a state. There are 3 risky states, two of which $((1,0), (2,2))$ correspond to penalty states and one is low probability high reward state (see reward description below), starting and goal states and the rest of the states are considered safe.

- Actions: The agent can move up, down, left, or right.

- Rewards: Safe steps incur a small penalty $(-10)$. The risky state $(2,4)$ offers a high reward $(10^6)$ with low probability $(0.01)$, otherwise a large penalty $(-10^3)$. The remaining risky states give the same penalty. Reaching the goal provides a reward of $50,000$.

- Transitions: Movement is deterministic given the actions which are sampled according to the trained policy (which is not deterministic in our setting).

**Algorithms for policy optimization.** We compare our CPT-PG algorithm with an exponential risk-sensitive PG algorithm (using an exponential utility $(\mathcal{U}(x) = \frac{1}{\beta} \exp(-\beta x)), \beta = 0.1$, without probability weighting). For our CPT-PG algorithm, we use Kahneman and Tversky's utility function with parameters $(x_0 = 1, \lambda = 3, \alpha = 0.6)$ (see section 2) and a probability weight function which is piecewise affine given by the following coefficients $w = [4, 0, 0.8, 0.2, 2.7, -1.7, 0.1, 0.9]$ where $[a_1, a_2, a_3, b_1, b_2, b_3, c_1, c_2]$ stands for a piecewise affine $w$ function with $w(x) = a_i x + b_i$ for $c_{i-1} < x < c_i$. Note that this weight function inflates low probability events. This is one of the key features of CPT that we illustrate. We use a step size $\alpha = 0.01$ for both algorithms. For the policy network, we use a simple two-layer feed-forward neural network with a Leaky ReLU activation in the hidden layer and a softmax output layer for action selection.

**Policy visualization.** We provide heat maps visualizations of trained policy networks in the RiskyGridworld environment, representing the probability of selecting a risky action at each state. For each state we define risky actions as the actions leading to risky states. Therefore, the only states possibly leading to risky states are the states adjacent to risky ones. For each one of these states we encode the risky action as the one leading to the risky state (choose the riskier one if there are multiple ones). Then the heatmap assigns to each state the probability of selecting the risky action associated to it (probability as provided by the trained policy under consideration in that state).

**Results.** See figures 8 and 9 and their captions in the next page.

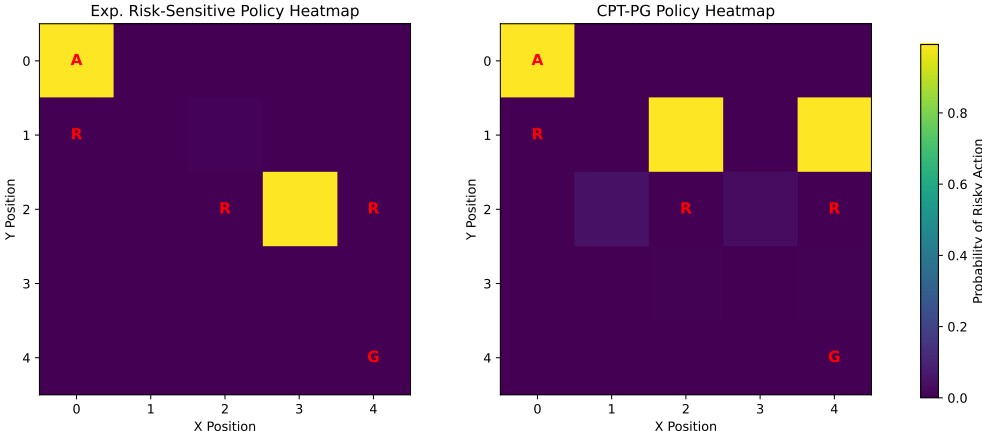

Figure 8: Heat maps representing the policies trained using our algorithm CPT-PG and an exponential risk-sensitive PG algorithm. Each cell in the $5 \times 5$ grid corresponds to a state, risky states are denoted by the letter 'R' in the cell, 'A' stands for the initial state and 'G' for the goal state, all the other states are considered safe (with a zero probability assigned). The color represents the probability of selecting a risky action at each state. The risky state (2,4) is risky in the sense that with low probability 0.01 it leads to high reward of $10^6$ and a penalty of $-10^3$ otherwise. The main observation here is that CPT takes more risk in trying to end up in this risky state (low probability high reward) as you can see with the yellow cell above the risky state. Overall the risk profile is different for both methods. This is partly explained by the fact that CPT inflates low probability events thanks to the probability weight function.

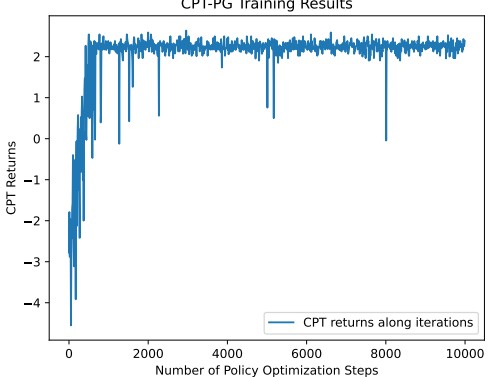 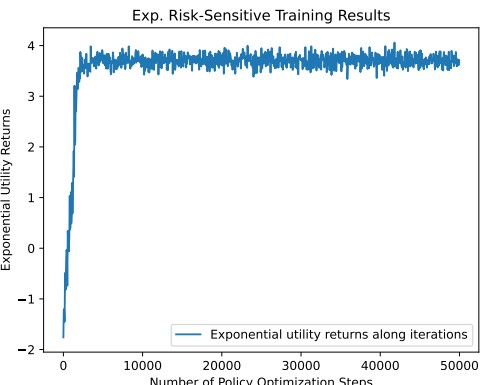

Figure 9: Policy training curves: (**Left**) CPT returns for CPT-PG. (**Right**) Exponential utility returns for Exponential Risk-Sensitive PG. Quantities are not comparable as one shows the CPT returns optimized in CPT policy optimization whereas the exponential risk-sensitive PG optimizes the exponential utility of the returns. Hence we represent them in separate graphs. This figure mainly serves the purpose of showing that the policies have been trained to maximize their respective objectives.

# G   Applications of CPT: From Prior Work in Stateless Settings to CPT-RL

In this section, we provide a discussion regarding the applications where CPT has already been successfully used (mainly in the static stateless setting) and potential applications in the dynamic (RL) setting with state transitions.

We highlight that CPT has been tested and effectively used in a large number of compelling behavioral studies that we cannot hope to give justice to here. Besides the initial findings of Tversky and Kahneman for which the latter won the Nobel Prize in economics in 2002, please see a few recent references below for a

broad spectrum of real-world applications ranging from economics to transport, security and energy, mostly in the stateless (static) setting.

- Risk preferences across 53 countries worldwide in an international survey (Rieger et al., 2017). Estimates of CPT parameters from data illustrate economic and cultural differences whereas probability weighting also reflects gender differences as well as economic and cultural impacts. Note here the explainability feature of CPT.

### G.1 Energy: Renovation and Home Energy Management

- A study of homeowners in the Netherlands to investigate energy retrofit decision using CPT (Ebrahimigharehbaghi et al., 2022). CPT is shown to predict the number of homeowners decisions to renovate their homes more accurately than Expected Utility Theory (EUT).

- Home energy management (Dorahaki et al., 2022). This work proposes a behavioral home energy management model to increase the user's satisfaction.

### G.2 Security: Building Evacuation

- Application of CPT to building evacuation (Gao et al., 2023). CPT allows to take into account individual psychology and irrational behavior in modeling evacuations via pedestrian movement modeling. This is particularly important for designing and optimizing emergency and safety management strategies.

### G.3 Urban Planning and Mobility

- **Parking:** Understanding private parking space owners' propensity to share their parking spaces by considering their psychological concerns as well as their socio-demographic and revenue characteristics for instance (Yan et al., 2020). This might be useful to help developing shared parking services.

- **Traffic routing.** Gao et al. (2010) model the travelers' strategic behavior for route choice in a stochastic network when adapting to traffic conditions which are revealed en route.

### G.4 Finance

- Empirical study about financial decision making in two universities in Argentina (Ladrón de Guevara Cortés et al., 2023). In particular, it is shown that the financial decisions of the participants under uncertainty are more consistent with Prospect Theory than expected utility theory.

### G.5 Example: Personalized Treatment for Pain Management

We illustrate our CPT-RL problem formulation with a concrete example in healthcare to give the reader more intuition about the different features of CPT-RL, its importance in applications when human perception and behavior matter and its differences compared to risk-sensitive RL. The goal is to help a physician manage a patient's chronic pain by suggesting a personalized treatment plan over time. The challenge here is to balance pain relief and the risk of opioid dependency or other side effects that might be due to the treatment, i.e. short-term relief and longer term risks. The idea is to train a CPT-RL agent to help the physician.

**1. *Why RL?*** (a) The physician needs to adjust treatment at each time step depending on the patient's reported pain level as well as the observed side effects. This is relevant to dynamic treatment regimes in general (such as for chronic diseases, see e.g. Yu et al. (2021) for a survey) in which considering delayed effects of treatments is also important and RL does account for such effects. (b) Decisions clearly impact the patient's immediate pain relief, dependency risks in the future and their overall health condition. A state is described by e.g. current pain level, dependency risk and side effect severity. Actions are treatments, e.g. no treatment, opioid or alternative treatment.

**2. *Why CPT-RL?*** Patients and clinicians make decisions influenced by psychological biases. We illustrate the importance of each one of the three features of CPT in section 2 (reference point, utility and probability distortion weight functions) via this example:

- *Reference points:* Patients assess and report pain levels according to their subjective (psychologically biased) baseline. Incorporating reference point dependence leads to a more realistic model of human decision-making taking into account *perceived* gains and losses. In our example, reducing pain from a level of 7 to 5 is not perceived the same way if the reference point of the patient is 3 or 5. In contrast, risk-sensitive RL treats every pain reduction as a uniform gain, regardless of the patient's starting reference pain level.

- *Utility transformation:* Patients might often show a loss averse behavior, i.e., they might perceive pain increase or withdrawal symptoms as worse than equivalent gains in pain relief. Note here that loss aversion should not be confused with risk aversion (Schmidt & Zank, 2005). In short, loss aversion can be defined as a *cognitive bias* in which the emotional impact of a loss is more intense than the satisfaction derived from an equivalent gain. For instance, in our example, a 2-point increase in pain might be seen as much worse than a 2-point reduction even if the change is the same in absolute value. This loss aversion concept is a cornerstone of Kahneman and Tversky's theory. In contrast, risk aversion rather refers to the *rational* behavior of undervaluing an uncertain outcome compared to its expected value. Risk sensitive approaches might be less adaptive to a patient's subjective preferences if they deviate from objective risk assessments.

- *Probability weighting:* Low probability events such as severe side effects (e.g., opioid overdose or dependency) might be overweighted or underweighted based on the patient's psychology.

**3. *CPT-RL vs Risk-averse RL.*** In terms of policies, risk-averse RL would favor non-opioid treatments unless extreme pain levels make opioids justifiable. In contrast, CPT-RL policies would prescribe opioids if pain significantly exceeds the patient's reference point. As dependency risk increases, CPT-RL policies would transition to non-opioid treatments as a consequence of overweighting the probability of rare catastrophic outcomes. Notably, CPT-RL policies can oscillate between risk-seeking (to address high pain) and risk-averse (to avoid severe side effects). In contrast, a risk-sensitive agent focuses on minimizing variability in health states and dependency risks and would likely avoid opioids in most cases unless pain levels become extreme. Such risk-sensitive policies favor stable strategies (e.g., consistent non-opioid use), prioritizing low variance in patient outcomes.

### G.6 Further Applications of CPT-RL

Our CPT-RL problem formulation finds applications in a number of diverse areas. A nonexhaustive list includes:

- **Traffic control.** We refer the reader to our toy example in the main part. simulations for specific CPT-RL applications in simple settings for traffic control, electricity management and financial trading that we will not discuss again here.

- **Electricity management.** Please see simulations in the main part (section 5 and App. J.7) in a simple example setting to illustrate our methodology.

- **Finance:** portfolio optimization, risk management, behavioral asset pricing (e.g. influence of investor sentiment on price dynamics via e.g. over-weighting of low-probability events, including their preferences). For recent applications of CPT to finance, we refer the reader to a recent paper (Luxenberg et al., 2024) using CPT for portfolio optimization (in a stateless static setting). We also applied our methodology to financial trading (see App. J.8).

- **Health:** personalized treatment plans, (e.g. health insurance design for specific groups modeling risk and factoring perceived fairness).

On a more high-level note, we would like to mention that CPT-RL is of practical relevance for finance and healthcare for several reasons: in short, CPT allows for (a) **modeling human biases**, (b) **factoring risk**, and (c) **capturing individual preferences for personalization.** All these three points are essential in the above applications.

Many other meaningful human-centric applications are yet to be explored, including legal and ethical decision making, cybersecurity and human-robot interaction.

## H  CPT-RL and Trajectory-Based Reward RL as Preference Learning Paradigms

In this section, we compare the CPT-RL and trajectory-based reward RL (using a single reward for the entire trajectory, such as Reinforcement Learning from Human Feedback) seen as preference learning paradigms. In particular, we also discuss the pros and cons of each one of them.

Regarding the structure of the final reward and the metric learning you mention, this is a fair point and we agree that Our present work requires so far access to utility and weight functions whereas trajectory-based reward RL learns the metric to be optimized using human preference data. However, let us mention a few points:

(a) These can be readily available in specific applications (for risk modeling or even chosen at will by the users themselves);

(b) CPT relies on a predefined model, this can be beneficial in applications such as portfolio optimization or medical treatment where trade-offs have to be made and models might be readily available;

(c) Furthermore, we argue that having such a model allows it to be more explainable compared to a model entirely relying on human feedback and fine tuning, let alone the discussion about the cost of collecting human feedback. We also note that some of the most widely used algorithms in RLHF (e.g. DPO) do rely on the fact that the reward follows a Bradley-Terry model for instance (either for learning the reward or at least to derive the algorithm to bypass reward learning);

(d) Let us mention that one can also learn the utility and weight functions. We mentioned this promising possibility in our conclusion although we did not pursue this direction in this work. One can for instance represent the utility and weight functions by neural networks and train models to learn them using available data with relevant losses, jointly with the policy optimization task. One can also simply fit the predefined functions (say e.g. Tversky and Kahneman's function) to the data by estimating the parameters of these functions (see $\eta$ with our notations and exponents of the utility function in Table 1 for the CPT row). This last approach is already commonly used in practice, see e.g. Rieger et al. (2017).

**CPT vs RLHF: General comparison.**  CPT has been particularly useful when modeling specific biases in decision making under risk to account for biased probability perceptions. It allows to *explicitly* model cognitive biases. In contrast, RLHF has been successful in training LLMs which are aligned with human preferences where these are complex and potentially evolving and where biases cannot be explicitly and reasonably modeled. RLHF has been rather focused on learning *implicit* human preferences through interaction (e.g. using rankings and/or pairwise comparisons). Overall, CPT can be useful for tasks where risk modeling is essential and critical whereas RLHF can be useful for general preference alignment although RLHF can also be adapted to model risk if human preferences are observable and abundantly available at a reasonable cost. This might not be the case in healthcare applications for instance, where one can be satisfied with a tunable risk model. On the other hand, so far CPT does not have this ability to adapt to evolving preferences over time unlike RLHF which can do so via feedback.

**CPT and RLHF: Pros and cons.** To summarize the pros and cons of both approaches, we provide the following elements. As for the pros, CPT directly models psychological human biases in decision making via a structured framework which is particularly effective for risk preferences. RLHF can generalize to different scenarios with sufficient feedback and handle complex preferences via learning from diverse human interactions, it is particularly useful in settings where preferences are not explicitly defined such as for LLMs for aligning the systems with human preferences and values. As for the cons, CPT is a static framework since the utility and probability weight functions are fixed, it is hence less adaptive to changing preferences. It uses a predefined model of human behavior which is not directly using feedback. It also requires to

estimate model parameters precisely, often for specific domains. As for RLHF on the other hand, the quality and the quantity of the human feedback is essential and this dependence on the feedback clearly impacts performance. This dependence can also cause undesirable bias amplification which is present in the human feedback. We also note that training such models is computationally expensive in large scale applications.

**CPT and RLHF are not mutually exclusive.** While CPT and trajectory-based RL (say e.g. RLHF) both offer frameworks for incorporating human preferences into decision making, we would like to highlight that CPT and RLHF are not mutually exclusive. We can for instance use CPT to design an initial reward structure reflecting human biases, then refine it with RLHF. We can also consider to further relax the requirement of sum of rewards (which already has several applications on its own) and think about incorporating CPT features to RLHF. Some recent efforts in the literature in this direction that we mentioned in our paper include the work of Ethayarajh et al. (2024) which combines prospect theory with RLHF (without probability weight distortion though, which limits its power). Note that the ideas of utility transformation and probability weighting are not crucially dependent on the sum of rewards structure and can also be applied to trajectory-based rewards or trajectory frequencies for instance. We believe this direction deserves further research, one interesting point would be how to incorporate risk awareness from human behavior to such RLHF models using ideas from CPT.

# I More About CPT Values and CPT Policy Optimization

## I.1 More about Optimal Policies in CPT-RL

**The need for stochastic policies.** We start our discussion by pointing out a stark difference between optimal policies in standard MDPs and CPT-PO. While there exists an optimal *deterministic* stationary policy for MDPs (see e.g. Thm. 6.2.10 in Puterman (2014)), this is not the case in general for CPT-PO. Indeed, there does not always exist an optimal policy for CPT-PO in $\Pi_{NM}^D$. In general, stochasticity of the policy is essential in solving our CPT-RL problem. To see this, consider an example built around a function $w_+$ that puts special emphasis on the 10% of the best outcomes. In this case, the optimal policy needs to be randomized to take advantage of this and obtain the highest returns with some probability without suffering from bad outcomes by deterministically committing to this riskier strategy (see App. C.1 for a proof).

**Importance of probability weighting.** When setting the probability weight distortion function $w$ to the identity, i.e. when considering the particular case EUT-PO of CPT-PO, it appears that an optimal policy is not necessarily stochastic. Indeed, it has been shown that there exists an optimal policy for EUT-PO in $\Pi_{\Sigma,NS}^D$ (see e.g. Theorem 1 in Bäuerle & Rieder (2014)). Therefore, the need for stochasticity in the optimal policy is clearly due to the probability distortions in the CPT value. The aforementioned result allows to safely restrict the policy search to $\Pi_{\Sigma,NS}$ which is a much smaller policy space than the set of non-Markovian policies $\Pi_{NM}$. The fact that an optimal deterministic policy exists is a fundamental difference with the general CPT-PO setting. Whether there are specific weight functions (apart from the identity) for which there always exist a deterministic optimal policy remains an open question left for future work.

**The need for non-Markovian policies.** We now ask the next natural question: Can we further restrict our policy search to a smaller policy class compared to $\Pi_{\Sigma,NS}$? In particular, are there specific utility functions for which the resulting EUT-PO problem has optimal *Markovian* policies? Bäuerle & Rieder (2014) provide a positive answer by establishing a precise characterization of such utility functions which turn out to be either affine or exponential (when $\mathcal{U}$ is continuous and increasing). This highlights the role of the (nonlinear) utility functions on the nature of optimal policies. However, these results cannot be extended to CPT-PO in general.

## I.2 Positioning CPT-RL in the literature

**Remark 22.** *For the infinite horizon discounted setting, the objective becomes the CPT value of the random variable $X = \sum_{t=0}^{+\infty} \gamma^t r_t$ recording the cumulative discounted rewards induced by the MDP and the policy $\pi$. The policy can further be parameterized by a vector parameter $\theta \in \mathbb{R}^d$.*

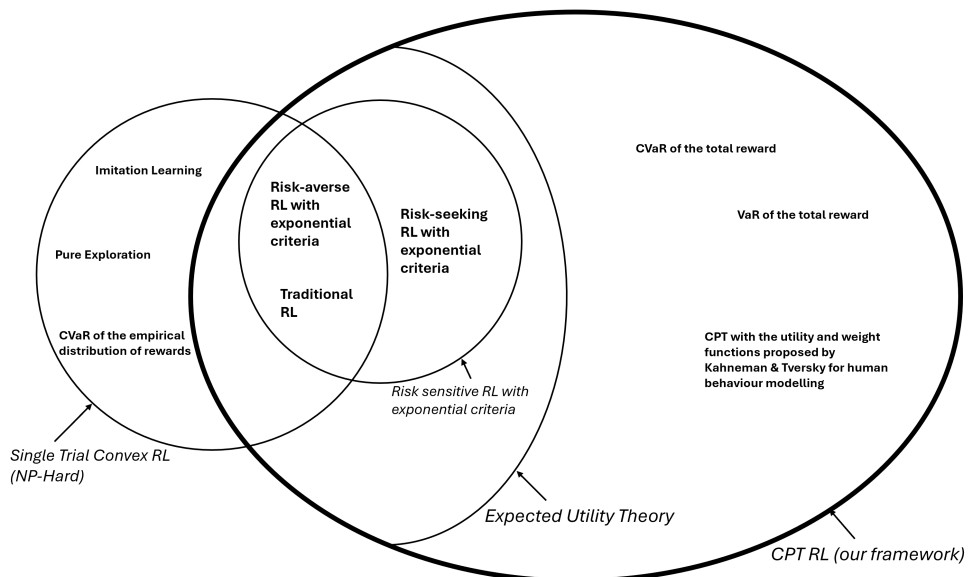

Figure 10: A Venn Diagram representing our framework and some other frameworks in the literature

### I.3 CPT value examples

| Setting | Utility function | $w_+$ | $w_-$ |
|---|---|---|---|
| CPT | Any | Any | Any |
| CPT (Functions proposed by Kahneman and Tversky) | $\begin{cases} (x - x_0)^\alpha & \text{if } x \geq 0, \\ -\lambda(x - x_0)^\alpha & \text{if } x < 0 \end{cases}$ | $\frac{p^\eta}{(p^\eta + (1-p)^\eta)^{\frac{1}{\eta}}}$ | $\frac{p^\delta}{(p^\delta + (1-p)^\delta)^{\frac{1}{\delta}}}$ |
| EUT | Any | Identity function | Identity function |
| Distortion risk measure | Identity function | Any | $1 - w_+(1 - t)$ |
| CVaR* ((Balbás et al., 2009)) | Identity function | $1 - w_-(1 - t)$ | $\begin{cases} \frac{x}{1-\alpha} & \text{if } 0 \leq x < 1 - \alpha, \\ 1 & \text{if } 1 - \alpha \leq x \leq 1 \end{cases}$ |
| VaR* (Balbás et al., 2009) | Identity function | $1 - w_-(1 - t)$ | $\begin{cases} 0 & \text{if } 0 \leq x < 1 - \alpha, \\ 1 & \text{if } 1 - \alpha \leq x \leq 1 \end{cases}$ |
| Risk-sensitive RL with exponential criteria (Noorani et al., 2022) | $\frac{1}{\beta} \exp(\beta x), \beta > 0$ | Identity function | Identity function |

Table 1: CPT value examples. *: Note that $w_+$ and $w_-$ are discontinuous for VaR and CVaR.

### I.4 Proof: CVaR, Var and distortion risk measures are CPT values

For a random variable $X$ and a non-decreasing function $g : [0, 1] \to [0, 1]$ with $g(0) = 0$ and $g(1) = 1$, the **distortion risk measure** (Sereda et al., 2010) is defined as:

$$\rho_g(X) := \int_{-\infty}^{0} \tilde{g}(F_{-X}(x))dx - \int_{0}^{+\infty} g(1 - F_{-X}(x))dx,$$

where $F_{-X} : t \mapsto \mathbb{P}(-X \leq t)$ and $\tilde{g} : t \mapsto 1 - g(1 - t)$.

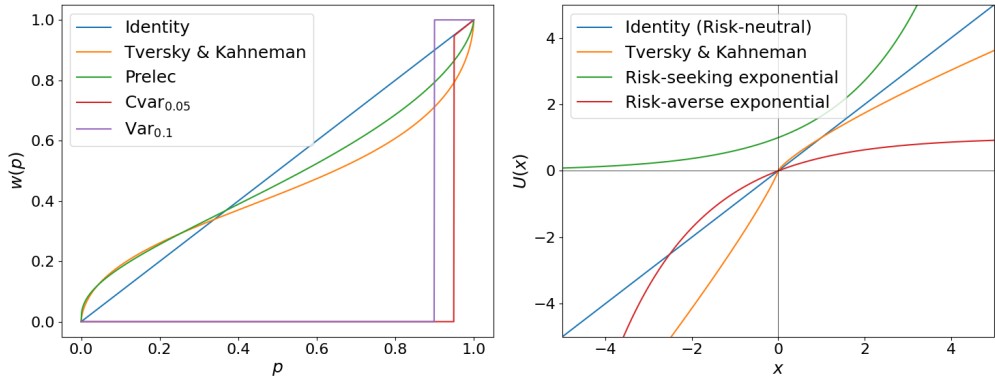

Figure 11: Various examples of probability weight functions (left) and utility functions (right).

**Proposition 23.** *Any distortion risk measure of a given random variable $X$ can be written as a CPT value with $u^+ = id^+$, $u^- = -id^-$, $w_+ = \tilde{g}$ and $w_- = g$.*

*Proof.* It follows from the definition of the distortion risk measure together with a simple change of variable $x \mapsto -x$ that:

$$
\begin{aligned}
\rho_g(X) &= \int_{-\infty}^{0} \tilde{g}(F_{-X}(x))dx - \int_{0}^{+\infty} g(1 - F_{-X}(x))dx \\
&= -\int_{+\infty}^{0} \tilde{g}(F_{-X}(-x))dx - \int_{0}^{+\infty} g(1 - F_{-X}(x))dx \\
&= \int_{0}^{+\infty} \tilde{g}(F_{-X}(-x))dx - \int_{0}^{+\infty} g(1 - F_{-X}(x))dx \\
&= \int_{0}^{+\infty} \tilde{g}(\mathbb{P}(-X \leq -x))dx - \int_{0}^{+\infty} g(1 - \mathbb{P}(-X \leq x))dx \\
&= \int_{0}^{+\infty} \tilde{g}(\mathbb{P}(X \geq x))dx - \int_{0}^{+\infty} g(\mathbb{P}(-X > x))dx \,.
\end{aligned}
$$

Since $g(\mathbb{P}(-X > x)) = g(\mathbb{P}(-X \geq x))$ almost everywhere (in a measure theoretic sense) on $[0, +\infty($, and $g$ is bounded, we obtain:

$$
\rho_g(X) = \int_{0}^{+\infty} \tilde{g}(\mathbb{P}(X \geq x))dx - \int_{0}^{+\infty} g(\mathbb{P}(-X \geq x))dx \,.
$$

We recognize the CPT-value of $X$ with $u^+ = \mathrm{id}^+$, $u^- = -\mathrm{id}^-$, $w_+ = \tilde{g}$ and $w_- = g$. $\qquad \square$

**Remark 24.** *When $X$ admits a density function, Value at Risk (VaR) and Conditional Value at Risk (CVaR) (Wirch & Hardy (2001)) have been shown to be special cases of distortion risk measures and are therefore also instances of CPT-values.*

### I.5 Simple examples and further insights about CPT

Human decision makers might not act rationally due to psychological biases and personal preferences, their decisions might not necessarily be dictated by expected utility theory. Consider this simple example as a first illustration: A player must choose between (A) receiving a payoff of 80 and (B) participating in a lottery and receive either 0 or 200 with equal probability. The player's preference depends on their attitude towards risk. While a risk-neutral agent will be satisfied with the immediate and safe payoff of 80, another individual might want to try to obtain the much higher 200 payoff. In particular, different agents might perceive the same utility and the same random outcome differently. Furthermore, they can exhibit both

risk-seeking and risk-averse behaviors depending on the context.

Therefore, due to its failure to capture such settings as a descriptive model, the standard expected utility theory has been called into question by the pioneering behavioral psychologist Daniel Kahneman together with his colleague Amos Tversky (Kahneman & Tversky, 1979). In particular, Daniel Kahneman has been awarded the Nobel Prize in Economic Sciences in 2002 "for having integrated insights from psychological research into economic science, especially concerning human judgment and decision-making under uncertainty". In their seminal works combining cognitive psychology and economics, they laid the foundations of the so-called prospect theory and its cumulative version later on (Tversky & Kahneman, 1992) to explain several empirical observations that challenge the standard expected utility theory.

Let us illustrate this in a simple example borrowed from Ramasubramanian et al. (2021) (example 2 in section IV therein) for the purpose of our exposition. Consider a game where one can either earn \$100 with probability (w.p.) 1 or earn 10000 w.p. 0.01 and nothing otherwise. A human might rather lean towards the first option which gives a certain gain. In contrast, if the situation is flipped, i.e., a loss of 100 w.p. 1 versus a loss of \$10000 w.p. 0.01, then humans might rather choose the latter option. In both settings, the expected gain or loss has the same value (100). The CPT paradigm allows to model the tendency of humans to perceive gains and losses differently. Moreover, the humans tend to deflate high probabilities and inflate low probabilities (Tversky & Kahneman, 1992; Barberis, 2013). For instance, as exposed in L.A. et al. (2016), humans might rather choose a large reward, say 1 million dollars w.p. $10^{-6}$ over a reward of 1 w.p. 1 and the opposite when rewards are replaced by losses.

### I.6 Connection to General Utility RL and Convex RL in finite trials

In this section, we elaborate in more details on one of the connections we noticed (and mentioned in related works) between our CPT-PO problem of interest and the literature of generality utility RL.

First, we recall a few notations complementing the preliminaries. Any fixed policy $\pi$ and any initial state distribution $\rho$ induce together a state occupancy measure $d_\rho^\pi$ recording the visitation frequency of each state, it is defined at each state $s \in \mathcal{S}$ by $d_\rho^\pi(s) := \sum_{t=0}^{H-1} \mathbb{P}_{\rho,\pi}(s_t = s)$. The corresponding state-action occupancy measure is defined for every state-action pair $(s,a) \in \mathcal{S} \times \mathcal{A}$ by $\mu_\rho^\pi(s,a) := d_\rho^\pi(s)\pi(a|s)$. Recall that $J(\pi) = \langle \mu_\rho^\pi, r \rangle := \sum_{s \in \mathcal{S}, a \in \mathcal{A}} \mu_\rho^\pi(s,a)r(s,a)$ for any policy $\pi$ and any initial state distribution $\rho$.

The general utility RL problem consists in maximizing a (non-linear in general) functional of the occupancy measure induced by a policy. More formally, the general utility RL can be written as follows:

$$\max_\pi F(d_\rho^\pi), \tag{33}$$

where $F$ is the real valued utility function defined on the set of probability measures over the state or state-action space, $\rho$ is the initial state distribution and $d_\rho^\pi$ is the state (or sometimes state-action) occupancy measure induced by the policy $\pi$. This problem captures the standard RL problem as a particular case by considering a linear functional $F$ defined using a fixed given reward function. Recently, motivated by practical concerns, Mutti et al. (2023) argued for the relevance of a variation of the problem under the qualification of *convex RL in finite trials*. They introduce for this the empirical state distributions $d_n \in \Delta(\mathcal{S})$ defined for every state $s \in \mathcal{S}$ by:

$$d_n^\pi(s) = \frac{1}{nT} \sum_{i=1}^{n} \sum_{t=0}^{T-1} \mathbb{1}(s_{t,i} = s), \tag{34}$$

where $s_{t,i}$ is the state at time $t$ resulting from the interaction with the MDP (with policy $\pi$) in the $i$-th episode, among $n$ independent trials. Their policy optimization problem is then as follows:

$$\max_\pi \xi_n(\pi) := \mathbb{E}[F(d_n^\pi)]. \tag{35}$$

Note that $d_n^\pi$ is a random variable as it is an empirical state distribution. Observe also that $\lim_{n \to \infty} \xi_n(\pi) = F(d_\rho^\pi)$ under mild technical conditions (e.g. continuity and boundedness of $F$). This shows the connection

between the above final trial convex RL objective and the general utility RL problem (33). The interesting differences between both problem formulations arise for small values of $n$. Of particular interest, both in this paper and in Mutti et al. (2023), is the *single trial RL* setting where $n = 1$.

Setting the probability distortion function $w$ to be the identity, our CPT-PO problem becomes EUT-PO, i.e.:

$$\max_\pi \mathbb{E} \left[ \mathcal{U} \left( \sum_{t=0}^{H-1} r_t \right) \right], \tag{36}$$

which is of the form $\xi_1(\pi)$, the single-trial RL objective as defined in Mutti et al. (2023). Indeed, it suffices to write the following to observe it:

$$\mathcal{U} \left( \sum_{t=0}^{H-1} r_t \right) = \mathcal{U}(\langle d_1^\pi, r \rangle), \tag{37}$$

where $r$ is the reward function seen as a vector in $\mathbb{R}^{|\mathcal{S}|}$, $\langle \cdot, \cdot \rangle$ is the standard Euclidean product in $\mathbb{R}^{|\mathcal{S}|}$. Therefore, it appears that the above objective is indeed a functional of the empirical distribution $d_1^\pi$. Single trial general utility RL is more general than EUT-PO since it does not necessarily consider an additive reward inside the non-linear utility and can accommodate any (convex) functional of the occupancy measure. However, CPT-PO does not appear to be a particular case of single trial convex RL because of the probability distortion function introduced.

### I.7 Choice of the utility function

We provide here additional comments regarding the choice of the utility function to complement our brief discussion in the main part of the paper. The problems themselves might dictate to the user or decision maker the utility function to be used. The user might also design their own according to their own beliefs, behaviors and objectives, based on the goal to be achieved (e.g. risk-seeking, risk-neutral, risk-averse). Specific applications might also suggest specific utility functions such as specific risk measures like in risk sensitive RL for instance. We have provided in table 1 a list of different examples one might consider. Learning the utility function is also an interesting direction to investigate as we mention in the conclusion. In practice, it is rather common to use the example we provide in table 1 (CPT row) with exponent parameters which are estimated using data. We provide a few concrete examples in the following. For instance, Rieger et al. (2017) adopt such an approach (see sections 3.1, 3.2 and 3.3 therein for a detailed discussion about parameter estimation). Ebrahimigharehbaghi et al. (2022) choose some similar variation of this utility (see eqs. 2-3 therein) while still using KT's probability weighting functions. Gao et al. (2023) compare different functions for different similar power utility functions with fitted parameters (see Tables 1, 2 and 3 therein p. 3, 4, 6 for extensive comparisons with the existing literature). Similar investigations were conducted in Yan et al. (2020). Dorahaki et al. (2022) consider psychological time discounted utility functions (variations of the same power functions) in their model with additional relevant hyperparameters, motivated by (domain-specific) psychological studies (see eq. (4) therein). It is worth noting that all these examples are only in the static stateless setting.

## J  More Details about Section 5 and Additional Experiments

**Batch size and quantile estimation.** Quantile estimation is generally more sensitive to batch size than expectation-based methods. However, in practice, we observed that relatively small batches (5–32 samples) performed robustly across all environments. As shown in Fig. 15, larger batches can improve performance, but small batches already yield competitive CPT values.

**History dependence of policy networks.** In most experiments, we used feed-forward networks without explicit history dependence. We found that Markovian policies performed well in our simple simulations for applications like financial markets and Mujoco control. In other domains (e.g., electricity management), we incorporated partial history by expanding the input window to represent multiple past states. Typically, the input size was chosen to capture sufficient context without encoding the full trajectory length ($H$). This limited form of history dependence was sufficient in our settings, though incorporating more expressive temporal models could enhance performance further.

**Hardware and execution time.** We have run part of the experiments on a laptop with a 13th Gen Intel Core i7-1360P2.20 GHz CPU and 32 GB of RAM, and the remaining simulations (section 5, App. J.8, J.9) on a MacBook Pro M4 with 48 GB of RAM. Experiments can take a few seconds for the simplest ones, to a few minutes for the ones in the main part. More complex ones take a couple of hours, e.g. for electricity management (App. J.7), Finance (App. J.8) and MuJoCo (App. J.9).

**Software.** Our experiments are coded in Python and mainly use Pytorch (Paszke et al., 2019).

**Weight functions.** The risk-neutral $w_+$ function is simply the identity function. As for the definition of other probability distortion functions $w_+$ we use for experiments, we define:

$$w_{ra}(x) := \begin{cases} 0.5x & \text{if } x \leq 0.9, \\ 5.5x - 4.5 & \text{otherwise.} \end{cases} \quad w_{rs}(x) := \begin{cases} 5x & \text{if } x \leq 0.1, \\ \frac{1}{2} + \frac{5}{9}(x - 0.1) & \text{otherwise.} \end{cases}$$

$$w_{sra}(x) := \begin{cases} 0.1x & \text{if } x \leq 0.9, \\ 9.1x - 8.1 & \text{otherwise.} \end{cases} \quad w_{srs}(x) := \begin{cases} 9x & \text{if } x \leq 0.1, \\ \frac{1}{9}x + \frac{8}{9} & \text{otherwise.} \end{cases}$$

**Optimizer.** Instead of vanilla stochastic gradient descent, we use the Adam optimizer to speed up convergence. In our Python implementation, we use the same batch of trajectories for estimating the function $\phi$ and for performing the stochastic gradient ascent step.

**Neural net activation function.** We use the tanh activation function before the last softmax layer to encourage exploration and reduce the risk of converging to local optima which may occasionally occur for some runs.

### J.1 More details about bandit simulations in Section 5

In addition to the parameters specified in the main part, we used the following set of parameters to conduct the bandit experiments in Section 5:

Table 2: Hyperparameters for the bandit experiments in Section 5

| Hyperparameter | Value |
|---|---|
| Optimizer | Adam |
| Learning rate | 0.01 |
| Number of episodes | 1000 |
| Batch size | 10 |

We used a piecewise linear probability weight function given by the coefficients: $w = [1.0, 0.0, 1.0, 0.0, 0.5]$ where the notation $[a_1, a_2, \ldots, a_i, b_1, b_2, \ldots, b_i, c_1, \ldots, c_{i-1}])$ refers to a piecewise affine $w$ function with $f(x) = a_i x + b_i$ for $c_{i-1} < x < c_i$.

In Fig. 12, we show a case where CPT-PG learns a stochastic policy.

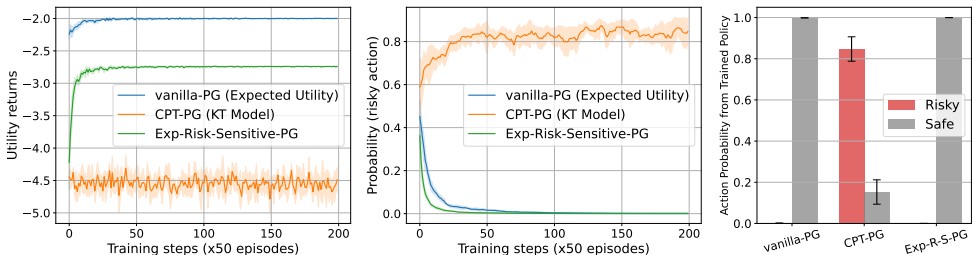

Figure 12: Comparison of our CPT-PG algorithm with exponential risk-sensitive PG and vanilla PG on a simple 2-action bandit setting. (Lower figure) Loss lottery setting: Only CPT picks the risky action with high probability. Note here that the policy trained by CPT-PG is stochastic.

### J.2 CPT compared to Distortion Risk Measures

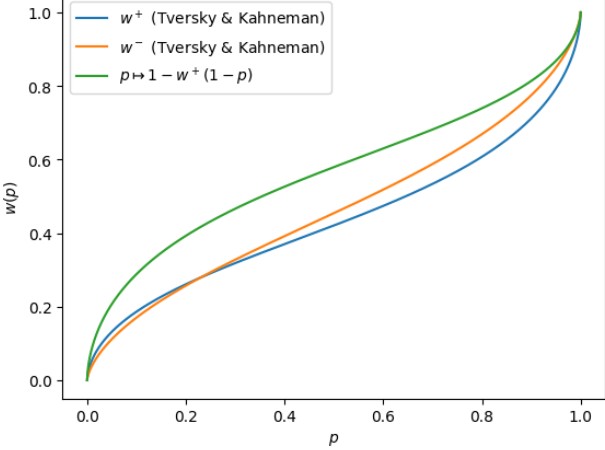

Figure 13: Illustration of the flexibility of CPT compared to the Distortion Risk Measure. Notice how $w_-$ is distinct from both $w_+$ and $p \mapsto 1 - w_+(1-p)$.

### J.3 Illustration of the need for stochastic policies in CPT-RL

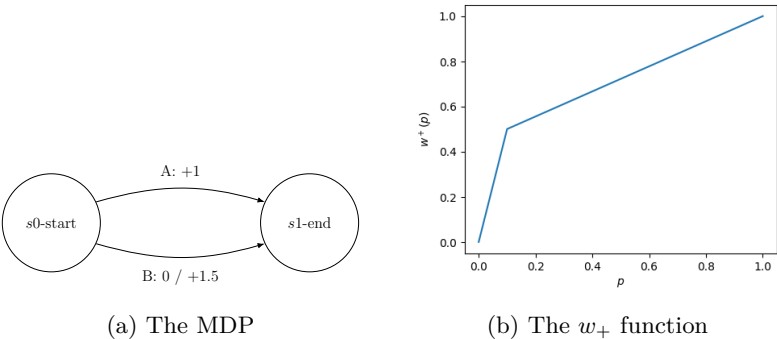

(a) The MDP         (b) The $w_+$ function

Figure 14: Setting of the experiment on non-deterministic policies and batch size influence

We study experimentally the behavior of our algorithm with regards to small batch sizes.

**Setting.** We use the barebones setting (Figure 14) with a $w$ function that aggressively focuses on the 10% of favorable outcomes. Denoting by $p$ the probability of choosing A for a given policy, we look at the value of $p$ at convergence (1000 optimization steps) for various batch sizes. The optimal policy corresponds to $p = 0.8$.

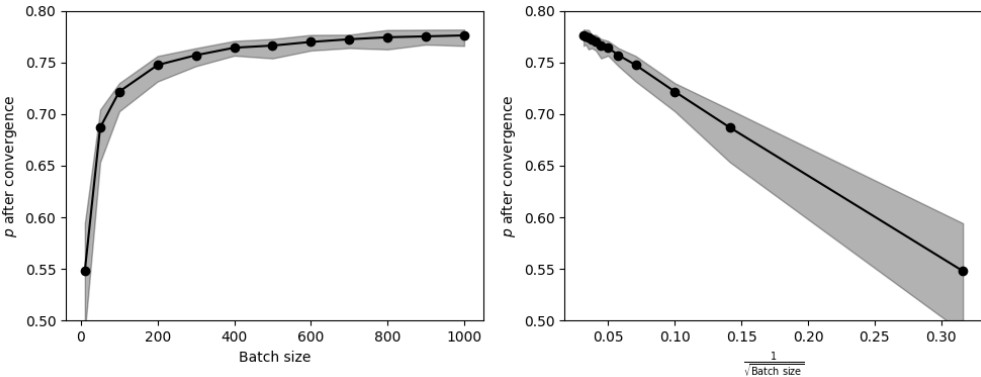

Figure 15: Results of the experiment on non-deterministic policies and batch size influence, over 100 runs. The black dots are the medians and the shaded area represents the interquartile range.

**Insights.** For each batch size we test, we run and plot a hundred training rounds (Figure 15). We fist observe that the policy we obtain with our algorithm indeed approaches the optimal $p = 0.8$ policy. The estimation error (w.r.t. the optimal theoretical value of $p = 0.8$) appears to be of order $\frac{1}{\sqrt{\text{batch size}}}$. It was to be expected that a small batch size would lead to a bias in CPT value and CPT gradient estimation, and, finally, in policy, as a small batch size renders impossible an accurate estimation of the probability distribution of the total return function. The fact that this bias appears to be proportional to the inverse of the square root of the batch size is in line with the standard statistical intuition (as e.g. per the central limit theorem). In our particular example, the estimated $p$ is below (and not above) the theoretical $p$. This is likely because our $w$ function places a strong weight on the top 10% of outcomes. Hence there is an imbalance between the impact of overestimating and underestimating the proportion of good outcomes in a given run: if we underestimate the probability of getting +1.5 with a given policy due to sampling, the effect will be stronger than the opposite effect we would get by overestimating the probability of the same error. As the batch size grows, the estimation error is reduced and the effect vanishes.

### J.4    Markovian vs Non-Markovian Policies for CPT-RL

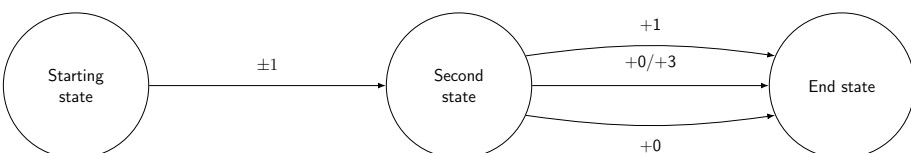

Figure 16: The environment for the experiment on non-Markovian policy

To illustrate the fundamental difference between memoryless utility functions and others we conduct a small experiment on a simple setting (Figure 16), similar to the one introduced in the proof of the theorem. We consider three states and three actions. From the starting state, any action leads to the second state with probability 1 and yields a reward of $+1$ with probability $\frac{1}{2}$ and of $-1$ with probability $\frac{1}{2}$. Once in the second state, the first action yields reward $+1$ with probability 1, the second action yields 0 or 3 with probability $\frac{1}{2}$ each, and the third action always yields 0. We compare the performance of a policy parametrized in $\Pi_{\Sigma,NS}$ and one in $\Pi_{M,NS}$.

**Insights.** The results (Figure 22) illustrate indeed the performance advantage of the non-Markovian policy compared to the Markovian one in the case of a non-affine, non-exponential utility function, and the absence thereof in the exponential setting. Note that the Markovian policy does not include the reward augmentation described in Proposition 3.

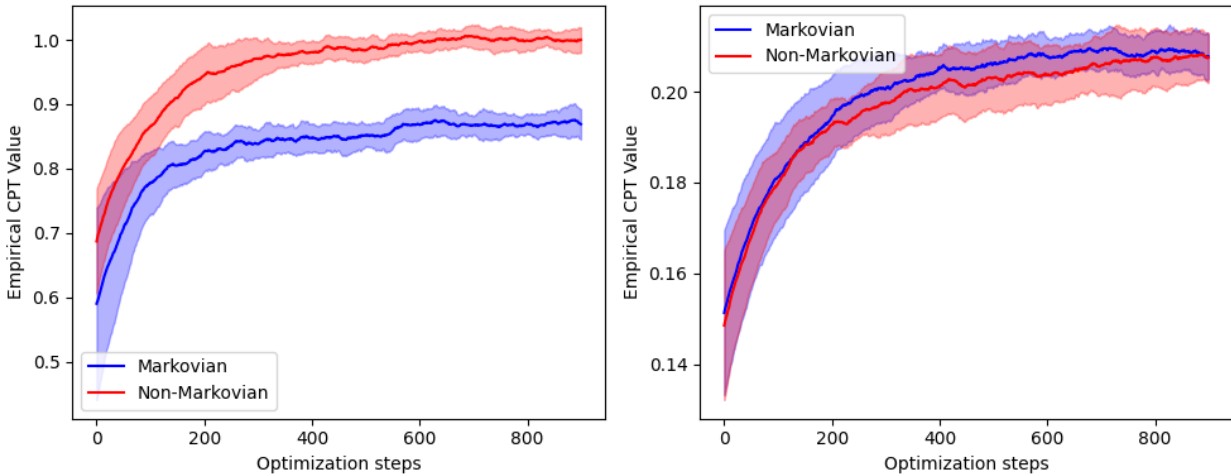

Figure 17: Comparison of Markovian and Non-Markovian policy performances for non-exponential (left) and exponential (right) utility functions. Shaded areas represent a range of $\pm$ one standard deviation over 20 runs.

### J.5 Grid Environment

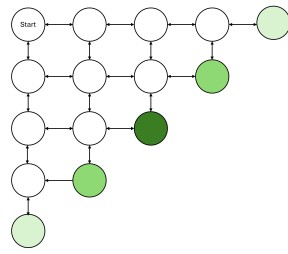

Figure 18: Scaling grid example.

| | | | |
|---|---|---|---|
| ↓ | ↓ | ↓ | ↓ |
| → | ↓ | ↓ | ↓ |
| → | → | → | ↓ |
| +5 | → | → | +6 |

(a) A risk-neutral optimal policy obtained with our algorithm

| | | | |
|---|---|---|---|
| ↓ | ↓ | ↓ | ↓ |
| ↓ | ↓ | ↓ | ↓ |
| ↓ | → | → | ↓ |
| +5 | → | → | +6 |

(b) An optimal policy obtained by training with the risk-averse utility $\mathcal{U} : x \mapsto \sqrt{x}$

Figure 19: Comparison of optimal policies under risk-neutral and risk-averse scenarios

**Exploration.** To avoid our gradient ascent algorithm getting stuck in a local optimum, we have to ensure enough exploration is going on. Therefore, we tweak the last layer of the neural network to prevent every action's probability from vanishing too soon. We choose a parameter $\alpha$, choose our last layer as $x \mapsto \text{softmax}(\alpha \tanh(x/\alpha))$, and we let $\alpha$ slowly grow with the iterations. A small $\alpha$ forces exploration, larger $\alpha$ allows for more exploitation: this is similar to an $\epsilon$-*greedy* scheme (with $\epsilon$ decaying as $\alpha$ grows), as it forces every action to be chosen with at least a small probability.

**Scalability to larger state spaces.** We consider a family of MDPs where the state space is a $n \times n$ grid for a given integer parameter $n$. The agent starts in the top right corner and has always four possible actions (up,down,left,right). Taking a step yields a reward of $\frac{-1}{n}$, attempting to leave the grid yields $\frac{-2}{n}$, and reaching the anti-diagonal ends the episode with a positive reward. All cells on the anti-diagonal yield the same expected reward, but with different levels of risk; the least risky reward is the deterministic one, in the center of the grid. We consider tabular policies and the initial policy is a random policy assigning the probability $1/4$ to each action. We test the sensitivity of the performance of both algorithms to the size of the state space. The steps sizes of both algorithms have been tuned to approach their possible performance; we wish to draw attention not to the absolute performance of either algorithm on any particular example, but rather to the evolution of the performance of both as the size of the problem increases. We observe that the performance of CPT-SPSA-G suffers for larger state space size whereas our PG algorithm is robust to state space scaling. While both algorithms are gradient ascent based algorithms in principle, our stochastic policy gradients are different.

**Influence of the utility function.** We consider a 4x4 grid for our illustration purpose. Our agent starts on a random square on one of the three upper rows of the grid, and can move in all four directions. Any move to an empty square will award it a random reward of $-1$ with probability $\frac{1}{2}$ and of $+0.8$ with probability $\frac{1}{2}$. Therefore, longer trajectories are slightly costly in expectation, and generate significant variance. In two corners of the grid, we add cells that yields rewards of $+5$ for one or $+6$ for the other, and conclude the episode. Illegal moves (attempting to leave the grid) are punished by a negative reward. Our parameterized policy is a neural network whose last layer is activated with softmax and has 4 coordinates corresponding to the 4 different possible moving actions. We consider solving CPT-PO with different utility functions: risk-neutral identity utility, risk-averse KT utility, as well as exponential utility function. The obtained policies differ depending on the utility function. For examples of risk-neutral/averse policies obtained, see Fig. 19b.

### J.6 Traffic Control over a Grid

We simulate a car agent navigating a city grid, where central roads are faster but risk higher delays (see fig. 20). The agent must balance speed against risk by avoiding the city center. In risk-neutral settings, the agent favors faster routes, while risk-averse policies avoid the risky central roads. Fig. 20 (center) demonstrates that our algorithm successfully adapts to different risk-weighted objectives. We also consider solving CPT-PO with different utility functions: risk-neutral identity utility, risk-averse KT utility, as well as exponential utility function. Fig. 23 show the corresponding different CPT returns observed. The obtained policies differ depending on the utility function. For examples of risk-neutral/averse policies obtained, see Fig. 19b in App. J.5.

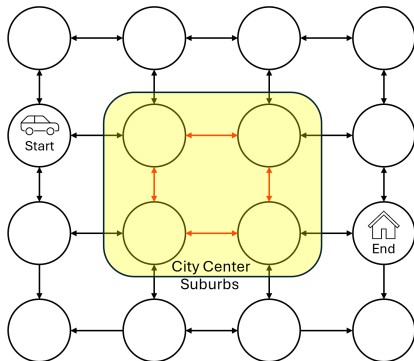

Figure 20: Traffic control environment: red roads in the city center are prone to congestion.

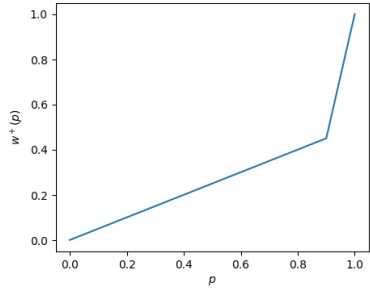

Figure 21: The probability distortion function $w_+$ used for the traffic control experiment.

|   | 0 | 1 | 2 |
|---|---|---|---|
| 0 | → | → | ↓ |
| 1 | ↑ | ? | 🏠 |
| 2 | ↑ | ? | ↑ |

(a) Training with our $w$ function for traffic control $(3 \times 3)$

|   | 0 | 1 | 2 |
|---|---|---|---|
| 0 | → | → | ↓ |
| 1 | → | → | 🏠 |
| 2 | → | → | ↑ |

(b) Risk-neutral reference

|   | 0 | 1 | 2 | 3 |
|---|---|---|---|---|
| 0 | → | → | ? | ↓ |
| 1 | ↑ | ? | → | ↓ |
| 2 | ↑ | ↑ | → | 🏠 |
| 3 | ↑ | ↑ | ? | → |

(c) Training with our $w$ function for traffic control $(4 \times 4)$

|   | 0 | 1 | 2 | 3 |
|---|---|---|---|---|
| 0 | → | → | → | ↓ |
| 1 | → | → | → | ↓ |
| 2 | → | → | → | 🏠 |
| 3 | → | → | → | ↑ |

(d) Risk-neutral reference

Figure 22: Examples of policies obtained with our algorithm. Question marks indicate a non-deterministic action selection in a given state.

**Implementation details.** In both cases, the risk-neutral optimal solution (going around the city center) is also a local optimum for the risk-averse objective, and, because it is a shorter path, is easier to stumble upon by chance when exploring the MDP. This means we have to implement special measures to force exploration. The algorithm used *as is* is prone to get stuck from time to time in local minima on this example. It would seem that our $w$ function, which is aggressively risk-averse, hinders exploration. To mitigate this, we introduce an entropy regularization term that we add to the score function with a decaying regularization

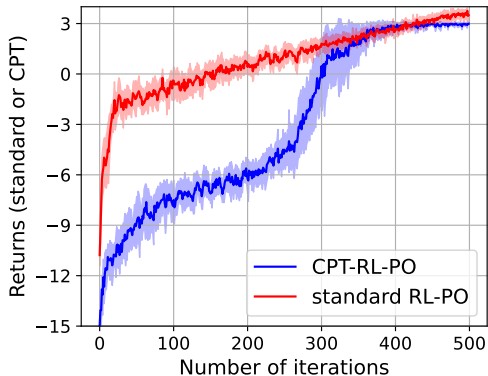

Figure 23: Returns along the iterations of our PG algorithm for CPT-PO for traffic control. Shaded areas indicate a range of ± one standard deviation over 20 runs. See App. J for details.

weight in the policy gradient found in Theorem 5, see App. J for further details. We incorporate entropy regularization in the policy gradient as follows:

$$\mathbb{E}\left[\phi\left(\sum_{t=0}^{H-1} r_t\right)\sum_{t=0}^{H-1}\nabla_\theta\Big(\log\pi_\theta(a_t|s_t) + \underbrace{\alpha_n H(\pi_\theta(a_t|s_t))}_{\text{Entropy regularization term}}\Big)\right], \tag{38}$$

where $\alpha_n$ is the weight of the regularization. We found that a decaying $\alpha_n$ yielded the best results.

On the $4\times4$ grid, we also start by pretraining our model with a risk-neutral method for a few steps, to accelerate training and avoid some bad local optima we can stumble upon due to unlucky policy initialization, before carrying on with our risk-aware method.

### J.7 Electricity Management

In this application involving *continuous* state and action spaces, we consider an individual home which has solar panels for producing electricity and a battery for storing energy (see Fig. 24, left). We consider a 24-hour time frame where the agent must decide the quantity of electricity to buy/sell, based on solar panel production, market prices, and battery levels. We use public data for selling prices recorded on a national electricity network. Public data is available online.[6] We experiment with risk-neutral, risk-averse, and risk-seeking objectives with 3 weight functions $w$ and a Gaussian neural network policy. Our algorithm performs well across these scenarios, with the risk-averse policy minimizing downside risk, and the risk-seeking policy maximizing potential gains. Fig. (24) (right) shows the distribution of total returns for different objectives. The most rewarding time to sell electricity is around 4pm (see prices in Fig. 24, center). However, selling too much too soon exposes us to the risk of falling short of battery during the night and risking to buy it later for a higher price. The risk-averse policy avoids selling a lot of electricity and tends to keep it stored until the end of the day. Conversely, the risk-seeking policy aggressively sells energy when the markets are high at the cost of possibly having to buy it again later in the day. The risk-averse policy has the distribution with the best left tail (worst cases are not too bad), the risk-seeking distribution has the best right tail (best cases are particularly good). The risk-neutral policy has the best mean value.

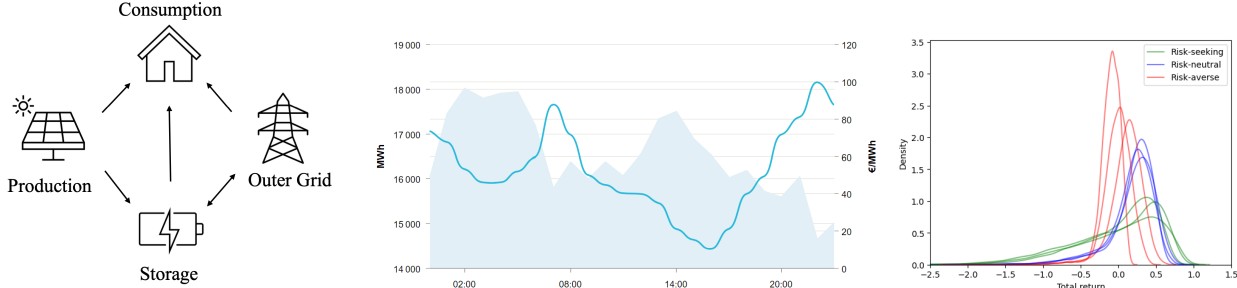

Figure 24: (**Left**) Electricity management environment: arrows refer to electricity flow. (**Center**) Electricity prices in a typical day, the blue line (right-hand side scale) records the electricity price on the European market, the shaded area (left-hand side scale) represents the total electricity production. (**Right**) Density of the empirical returns obtained by deploying different trained PG policies from different initializations, density computed using 10,000 runs for each curve.

---

[6]www.services-rte.com/en/view-data-published-by-rte/france-spot-electricity-exchange.html

### J.8 Trading in Financial Markets

We discuss here an application of our methodology to financial trading. The goal is to train RL trading agents using our general PG algorithm in the setting of our CPT-RL framework.

**Environment: general description.** We consider a gym trading environment available online, all the details about this environment are available here: https://gym-trading-env.readthedocs.io/en/latest/. This environment simulates stocks and allows to train RL trading agents. For the interest of the reader, we provide a brief summary explaining how the environment works. The environment is build from a given dataframe and a list of possible positions. The dataframe contains market data throughout a given period. The list of possible positions will represent the set of possible actions the agent can take, We provide more details about our specific environment in the following paragraph.

**Our trading environment.** We use data from the Bitcoin USD (BTC-USD) market between May 15th 2018 and March 1st 2022 available in the aforementioned website. We note that the data used follows the same pattern as publicly available data after a few preprocessing steps, the reader can find such data examples at https://finance.yahoo.com/quote/BTC-USD/history including the date, a few extracted features ('open', 'high', 'low', 'close') which respectively represent the open price, i.e. the price at which the first trade occurred for the asset at the beginning of the time period, the highest, lowest and last such prices, and the volume in USD which is the total value of all trades executed in a given time period. In particular, we will consider static features (computed once at the beginning of the data frame preprocessing) and dynamic features (computed at each time step) such as the last position taken as introduced by the Gym Trading Environment.

- State space: We consider a seven dimensional continuous state space. Features are constructed from the raw stock market data as previously explained. State transitions are described using the provided time series.

- Action space: We consider three classical types of positions the trader can take in a financial market: SHORT, OUT and LONG. These positions constitute the set of actions. These actions refer to whether the trader expects the price of an asset to rise or fall and how they are positioned to profit from that fluctuation. Extending this setting to a setting with a larger set of positions is straightforward as the environment implementation also supports more complex positions.

- Rewards: The rewards we consider are given by the log values of the ratio of the portfolio valuations at times $t$ and $t-1$. Borrowing interest rates and trading fees are also considered in the computation. The reward function can also be easily modified in the environment thanks to the implementation of the Gym Trading Environment which builds on the standard Gym environments.

**Remark 25.** *One can easily build their own environment by downloading their own dataframe for any historical stock market data and performing their desired preprocessing as for the features they would like to consider to build their states.*

**Experimental setting.** We have tested several utility and probability weighting functions including a risk averse exponential of the form $x \mapsto \frac{1}{\beta}(1 - \exp(-\beta x))$ with different values of $\beta$ as well as the KT (Kahneman and Tversky) function as defined in the main part with different values of the reference point $x_0$ to illustrate its influence.

**Hyperparameters.** We used the following set of parameters to conduct the experiments:

Additional hyperparameters used are directly reported in the legends of the figures below.

**Results.** We refer the reader to Fig. 2. We make a few observations:

Table 3: Hyperparameters

| Hyperparameter | Value |
|---|---|
| Optimizer | Adam |
| Learning rate | 0.05 |
| Number of episodes | 100 |
| Batch size | 5 |
| Number of steps per episode | 25 |

- Influence of the reference point: It can be seen that the reference point shifts the values of the achieved CPT returns: The smaller the reference point, the larger are the returns. This is because only values larger than the reference point are perceived as positive returns given the definition of the KT utility. This illustrates how the subjective perception of the agent of the returns is taken into account by the model.

- Different return trajectories for different risk averse functions: Different values of $\beta$ lead to different trajectories overall which can translate to different levels of risk aversion. In particular, the curves do not match the identity utility case in the first episodes and show more or less risk taken towards optimizing the CPT returns.

- Influence of the parameter $\alpha$ in KT's utility: Observe that the exponent $\alpha$ in the utility distorts the function and shifts the returns significantly. Lower values of $\alpha$ lead to higher returns in this setting where the returns (as per the ratio definition of the reward) are smaller than 1. This parameter $\alpha$ provides a degree of freedom to model the behavior of the agent as per their perception of the returns. Different values of $\alpha$ modify the curvature of the utility function (w.r.t. the reference point which is $x_0 = 0$ here) which is concave for gains and convex for losses.

### J.9   Control on MuJoCo Environments

In this section we test our algorithm on the INVERTEDPENDULUM-v5 environment (Todorov et al., 2012) to demonstrate that our PG algorithm is also applicable to other control benchmarks with continuous state and action spaces.

**Hyperparameters.** We used the following set of parameters to obtain our results:

| Hyperparameter | Value |
| --- | --- |
| Optimizer | Adam |
| Learning rate | 1e-4 |
| Number of episodes | 2000 |
| Batch size | 32 |
| Maximum number of steps per episode | 200 |

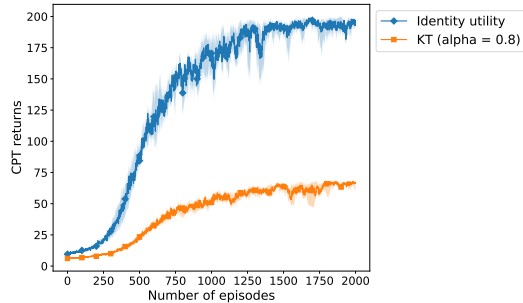

Figure 25: Performance of our PG algorithm on the INVERTEDPENDULUM-v5 environment (Todorov et al., 2012). KT refers to Kahneman and Tversky's utility function, alpha is the parameter used in the definition of KT's utility, exp. refers to exponential. Shaded areas are interquantile (25-75%) margins and curves report the median values over 10 different runs. All the CPT return curves are obtained with the same probability weighting function $w$ which is piecewise affine with three segments (hence different from the standard RL identity setting).

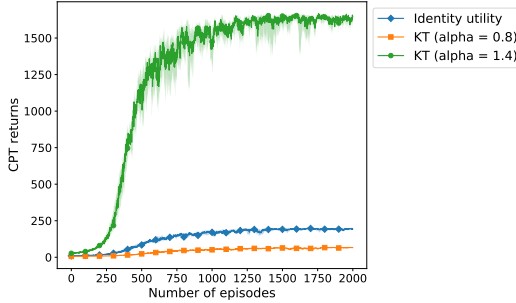

Figure 26: Performance of our PG algorithm on the INVERTEDPENDULUM-v5 environment (Todorov et al., 2012). This figure complements Fig 25 with the CPT returns using a KT utility with $\alpha = 1.4$. Notice that a much higher CPT return is achieved in that case. We also provide Fig. 25 for scaling purposes, the CPT returns being much higher for KT ($\alpha = 1.4$).

