# OpenReview forum: "Policy Gradients for Cumulative Prospect Theory in Reinforcement Learning"
_TMLR — Decision pending for TMLR_

### Review · Reviewer_bFSR · 2026-04-04

**Summary Of Contributions:**

The paper studies an policy optimization where the objective is not the expected cumulative reward, but rather the Cumulative Prospect Theory (CPT) value function. The CPT value function models asymmetric preferences and recovers the expected cumulative reward and various risk measures under particular choices of utility and probability weighting functions. The authors establish a few important properties of the CPT objective in Section 2.3, which highlights the difficulty of policy optimization under this objective. The authors also derive a closed-form expression for the policy gradient under the CPT objective, and propose a corresponding sample-based algorithm. The authors establish the convergence of the algorithm (in the first-order stationarity sense) and a sample complexity bound. A few numerical simulations are conducted that illustrate the characteristics of the CPT objective and the performance of the proposed algorithm over a zeroth-order-gradient baseline.

The paper is well-written and presents the main contributions in a very clear way. I also find the theoretical results and experimental results significant and enlightening.

**Audience:**

Yes

**Audience Explanation:**

Yes, I believe researchers in the RL community (especially those studying risk-aware RL) would find the work interesting.

**Broader Impact Concerns:**

No concerns.

**Claims And Evidence:**

Yes

**Claims Explanation:**

The policy gradient theorem, convergence and sample complexity analysis, and empirical validations are sound, and clearly support the theoretical claims and practical effectiveness of the proposed approach.

**Requested Changes:**

I do not have major concerns. A few questions/comments for the authors to consider:

1) What is the scale of the experiment on the financial problem? My understanding is that all experiments are conducted in small-scale/tabular settings. I believe the paper would benefit from empirical validations involving function approximation.

2) Can the authors clarify if it take non-trivial efforts if one would like to extend the generalized policy gradient theorem to infinite-horizon average- or discounted-reward MDPs?

3) Regarding the statement "In particular, our analysis does not require increasing batch sizes to establish asymptotic convergence." -- Isn't this the norm? In what problems are increasing batch sizes required for showing convergence?

---

> ### Author Response · Authors · 2026-05-25
> **Rebuttal**
>
> We thank the reviewer for their valuable feedback. We answer their questions below. Main additions are marked in blue, and we clarified statements, proofs, notation and assumptions.
>
> > 1. What is the scale of the experiment on the financial problem? My understanding is that all experiments are conducted in small-scale/tabular settings. I believe the paper would benefit from empirical validations involving function approximation.
>
> The financial experiment is not tabular: it uses a seven-dimensional state representation from Bitcoin USD market data and a neural-network policy. The action space has three positions and rewards are log portfolio-value ratios. Details are in Appendix J.8, p. 55–56.
>
> More generally, CPT-PG is a direct policy-gradient method and does not require a tabular value or Q-function. Our theory applies to differentiable parameterized policies, including neural-network policies subject to the stated regularity assumptions. We use feed-forward PyTorch policy networks.
>
> We agree that larger-scale experiments with richer function approximation would be valuable and leave them to future work. Given the paper’s theoretical and methodological focus, our empirical goal is to illustrate CPT-specific behavior, evaluate CPT-PG in representative settings, and compare it with CPT-SPSA-G, rather than provide a large-scale deep RL benchmark.
>
> > 2. Can the authors clarify if it take non-trivial efforts if one would like to extend the generalized policy gradient theorem to infinite-horizon average- or discounted-reward MDPs?
>
> We expect that extending the CPT policy-gradient theorem to the infinite-horizon discounted setting is possible but requires additional technical assumptions and a dedicated treatment. In particular, one can define the CPT objective for the discounted return $X_{\gamma} = \sum_{t \geq 0} \gamma^t r_t$, which is bounded when rewards are bounded and $\gamma < 1$. Under suitable regularity assumptions on the policy class, utility functions, and probability weighting functions, together with domination conditions allowing the exchange of differentiation and integration over the infinite trajectory space, the same likelihood-ratio argument should yield an analogous CPT policy-gradient expression.
>
> The average-reward setting is more subtle: the objective must be defined through a limiting random variable or limits of finite-horizon average-reward CPT values, e.g. $C(\frac1T\sum_{t=0}^{T-1}r_t)$ as $T\to\infty$. Since CPT is nonlinear in the return distribution, exchanging the long-run limit, CPT integral, and differentiation is not immediate and would likely require ergodicity/mixing and uniform integrability or domination assumptions. We therefore focus on finite horizons, where returns are bounded and the CPT objective is well-defined. We have added Remark 7 regarding extensions.
>
> > 3. Regarding the statement "In particular, our analysis does not require increasing batch sizes to establish asymptotic convergence." -- Isn't this the norm? In what problems are increasing batch sizes required for showing convergence?
>
> We removed this sentence, since CPT-PG also uses a biased finite-sample estimator and Theorem 14 now requires increasing batch sizes.
> The intended comparison was specifically with the zeroth-order CPT-SPSA-G algorithm [1]. In that work, increasing batch sizes are used to control the bias of CPT-value estimates inside a zeroth-order finite-difference gradient estimator (see Assumption A3 in section 4.2 therein). Because the finite-difference estimator divides differences of noisy CPT-value estimates by a vanishing perturbation radius, the estimation error must vanish sufficiently fast relative to this perturbation radius.
>
> In the revised manuscript, we clarify that the finite-sample CPT-PG estimator is also generally biased, due to the order-statistic/Riemann-sum approximation of the CPT-gradient integral. Accordingly, Theorem 14 now requires increasing batch sizes $n_k,m_k\to\infty$ to ensure that this bias vanishes and that the iterates converge asymptotically to the set of first-order stationary points.
>
> Thus, the key distinction is not avoiding increasing batch sizes, but avoiding zeroth-order finite-difference perturbations of CPT values. CPT-PG uses first-order likelihood-ratio information, so its bias is not amplified by division by a vanishing perturbation radius. Hence Theorem 14 only requires $n_k,m_k\to\infty$, rather than a rate condition coupling batch sizes to a perturbation parameter. We added this clarification after Theorem 14.
>
> [1] Prashanth L.A., Cheng Jie, Michael Fu, Steve Marcus, and Csaba Szepesvari. Cumulative prospect theory meets reinforcement learning: Prediction and control. In International Conference on Machine Learning, 2016.

---

> > ### Comment · Reviewer_bFSR · 2026-06-08
> >
> > I thank the authors' response to the questions. I have no further comments on 1) and 2).
> >
> > Regarding 3), I am concerned by the fact that Theorem 14 requires an increasing batch size. This suggests that, under stochastic gradients with bounded variance, Algorithm 1 does not converge exactly to a stationary point of J, but rather to a neighborhood around a stationary point with radius depending on the variance level. Can the authors clarify whether my understanding is correct?
> >
> > The standard step-size condition $\sum_{k=0}^{\infty}\alpha_k=\infty, \sum_{k=0}^{\infty}\alpha_k^2<\infty$ is typically sufficient for establishing exact asymptotic convergence of stochastic-gradient-type algorithms, since the variance term enters the analysis quadratically through the step size. What problem structure prevents the standard argument from being applied here and necessitates an increasing batch size?

---

> > > ### Author Response · Authors · 2026-06-08
> > >
> > > We thank the reviewer for the follow-up question. We answer it below.
> > >
> > > Regarding asymptotic convergence, the increasing batch size requirement is due to the finite-sample *bias* of our CPT-PG estimator rather than the variance.
> > >
> > > In standard stochastic approximation with *unbiased* stochastic gradients and bounded variance, the Robbins-Monro step size conditions are sufficient because the stochastic error is a martingale difference noise sequence and its cumulative effect is controlled by $\sum_k \alpha_k^2<\infty$. In our setting, however, the finite-sample estimator is generally biased for $\nabla J(\theta)$. The bias comes from the order-statistic approximation of the CPT marginal valuation $\phi(R(\tau))$, which depends *nonlinearly* on the survival probabilities inside $w_\pm'$. Thus the stochastic policy gradient can be decomposed as $\hat \nabla J(\theta_k) = \nabla J(\theta_k) + b_k + M_{k+1}$ where $M_{k+1}$ is a martingale difference noise sequence and $b_k$ is the finite-sample bias. The Robbins--Monro step size conditions control the martingale noise $M_{k+1}$, but they do not remove a persistent nonzero bias $b_k$. With fixed finite batch size, one should therefore expect convergence, at best, to the stationary points of a perturbed vector field, or to a neighborhood of the stationary set of $J$, with radius controlled by the **bias** level.
> > >
> > > The increasing batch size condition in Theorem 14 is a sufficient condition ensuring that this bias vanishes asymptotically. In particular, Proposition 8 establishes consistency of the CPT-PG estimator as the batch sizes grow, and Appendix E.4 gives the corresponding finite-sample bias control and asymptotic convergence proof building on the stochastic approximation framework. Once $b_k \to 0$, the remaining stochastic term can be handled by standard stochastic approximation arguments under Robbins-Monro steps. Thus the need for increasing batches comes from the biased CPT marginal-valuation estimator rather than the variance.

---

> > > > ### Comment · Reviewer_bFSR · 2026-06-12
> > > >
> > > > I appreciate the clarification. I have no more questions.

---

> > > > > ### Author Response · Authors · 2026-06-20
> > > > >
> > > > > We thank the reviewer for their follow-up and for their valuable comments, which helped us improve the manuscript.

---

### Review · Reviewer_oMLZ · 2026-04-22

**Summary Of Contributions:**

This paper integrates Cumulative Prospect Theory (CPT) —  into reinforcement learning via a policy gradient approach. The core contributions are a CPT policy gradient theorem, a first-order Monte Carlo algorithm (CPT-PG), and convergence/sample complexity guarantees.

Strengths:

1. The paper showed that CPT requires non-Markovian strategies for optimality.

2. The paper provides an asymptotic convergence guarantee over the policy gradient algorithm.

3. The paper provides a sample complexity guarantee for estimating the gradient.

4. The empirical results demonstrated the superiority of the proposed approach.

Weaknesses:

1. It is not a major weakness, but, the Markovian policy would not be optimal, which is not surprising.  We already know that for risk-sensitive RL (like CVaR), Markovian policies can be sub-optimal [A1].  [A1] shows that for a large number of risk-measure function (convex risk measures), Markovian policy in the augmented state-space is optimal, where the state is augmented with the total reward achieved so far. Can the authors show (or, provide counter example) to that for CPT? That would be a more interesting contribution.

2. The paper provides convergence guarantees; however, they require Lipschitzness condition on w. Can the authors comment on this assumption? For what types of $w$ is it satisfied?

3. Further on point 3, the convergence guarantee is only asymptotic; there is no finite-time guarantee. Hence, the overall sample complexity bound is rather confusing. In particular, sample complexity bound means how many samples are required to achieve a policy that is at most $\epsilon$-optimal; however, here it is used to estimate the gradient only, it does not guarantee a policy with a bounded sub-optimality gap.

4. Though the empirical results are appreciated, the baselines are rather weak. It would be better to see the performance compared to more risk-averse or distributionally robust RL policies.

5. The policy is history dependent, which raises concerns about tractability and the performance bound. If the authors can identify sufficient statistics of the history, that would be interesting, and practically significant.

[A1]. Wang, K., Liang, D., Kallus, N. and Sun, W., 2025, October. A Reductions Approach to Risk-Sensitive Reinforcement Learning with Optimized Certainty Equivalents. In International Conference on Machine Learning 2025, (pp. 63636-63661). PMLR

**Audience:**

Yes

**Audience Explanation:**

It would be of interest for robust RL or explainable AI community.

**Claims And Evidence:**

Yes

**Claims Explanation:**

The claims are supported by theoretical results.

**Requested Changes:**

See the weakness.

---

> ### Author Response · Authors · 2026-05-25
> **Rebuttal**
>
> We thank the reviewer for their time and feedback. We revised the manuscript with main additions marked in blue, including clarified statements, proofs, notation, and assumptions.
>
> **Response to 1:**
>
> We thank the reviewer for this suggestion. We now show that for terminal-return objectives such as CPT-PO, it is sufficient to optimize over Markov policies on a reward-augmented state space, where the augmentation tracks accumulated reward. We added this as Proposition 3 and Remark 4 (Sec. 2.3, p. 5), with proof in App. C.3 (pp. 26--27).
>
> **Response to 2:**
>
> The weighting-function regularity assumptions enter in two places. In Theorem 5, Lipschitzness and differentiability of $w_+$ and $w_-$ justify differentiating the CPT functional and yield the policy-gradient expression involving $w_+'$ and $w_-'$. In Propositions 8--10, Lipschitzness of $w_+'$ and $w_-'$ controls the empirical-survival/order-statistic approximation of the gradient integrals.
>
> These assumptions hold for many bounded-slope smooth distortions, including affine, quadratic/smooth polynomial distortions satisfying endpoint constraints, and endpoint-regularized CPT weights such as Tversky--Kahneman, Prelec, and Goldstein--Einhorn. Raw inverse-S CPT weights may have unbounded or steep endpoint derivatives and may require regularization. Similar assumptions appear in distortion-risk policy-gradient work; see Vijayan and L.A. (2024), Table 1. We added Remarks 6 and 11 for details.
>
> **Response to 3:**
>
> **Clarification regarding gradient estimation sample complexity.**
> Indeed, Proposition 10 is a gradient-estimation sample-complexity result: it controls the number of trajectories needed to ensure $\|\widehat{\nabla}J(\theta)-\nabla J(\theta)\|\le \varepsilon$ with high probability at a fixed $\theta$. It is not an optimization sample-complexity guarantee for producing an $\varepsilon$-optimal policy. We retitled it “Gradient estimation sample complexity” for clarity.
>
> **On finite-time global suboptimality.**
> We do not claim a finite-time global suboptimality-gap guarantee. Since the CPT objective is generally nonconvex in the policy parameter (due to nonconvexity of both the utility transformation and the probability distortion), guarantees of the form $J^\star-J(\theta)\le\varepsilon$ are not expected without additional structure. The more relevant target is approximate first-order stationarity.
>
> **New finite-time approximate first-order stationarity guarantee.**
> We now complement Theorem 14 with Lemma 13 and Theorem 15 (p. 10): Lemma 13 proves smoothness of $J$, and Theorem 15 gives a high-probability total sample complexity $\mathcal{O}(\varepsilon^{-4})$ for obtaining an $\varepsilon$-approximate first-order stationary policy parameter, using standard smooth nonconvex analysis with inexact gradients and Proposition 10. Proofs are in Apps. E.3 (p. 32-33) and E.5 (p. 36).
>
> **Response to 4:**
>
> We agree that broader benchmarking against additional risk-sensitive and distributionally robust RL algorithms would strengthen the empirical section when objectives are comparable; we leave this to future work. Our empirical goal is focused: to illustrate CPT-specific behavior and compare first-order CPT-PG with the closest prior CPT-RL baseline, the zeroth-order CPT-SPSA-G algorithm of L.A. et al. (2016). The paper’s main contribution is theoretical and methodological.
>
> CPT-RL and distributionally robust RL optimize different objectives. Distributionally robust RL typically optimizes worst-case performance over ambiguity sets for the environment, transition model, reward, or data distribution. Our setting has no ambiguity set or adversarial environment; CPT instead transforms the terminal-return distribution induced by the policy and MDP through reference dependence, asymmetric gain/loss utilities, and probability weighting.
>
> Thus CPT is not simply more risk-averse. With fixed CPT parameters, it can be risk-averse over gains and risk-seeking over losses, as shown in the reflection-effect bandit experiment (Fig. 1, p. 10--11). Standard risk-averse or robust objectives do not naturally reproduce this gain/loss asymmetry without changing the objective.
>
> **Response to 5:**
>
> The sufficient statistic is the current state, time, and accumulated reward. As noted above, Proposition 3 shows that every history-dependent policy admits a Markov policy on this augmented state space inducing the same terminal-return distribution, hence the same CPT value. Thus arbitrary history dependence can be reduced to reward-augmented Markov policies.
>
> Regarding tractability, our guarantees are for the chosen parameterized policy class. If $\pi_\theta$ is parameterized as a function of $(s_t,Z_t,t)$, then CPT-PG convergence/stationarity applies to the induced finite-dimensional objective $J(\theta)$. We do not claim global optimality over all measurable augmented-state policies, which would require additional structure and is generally out of reach due to nonconvexity.

---

> > ### Comment · Reviewer_oMLZ · 2026-06-06
> > **Still some issues**
> >
> > I would like to thank the authors for their responses. The new results definitely make the paper stronger and addressed some of my concerns. I have a few issues.
> >
> > **1. Differentiation in the proof of Theorem 5 (App. D.1).**
> > The derivation differentiates the outer trajectory measure $\rho_\theta$ but appears
> > to treat the weight $\phi^+(R(\tau))$ as independent of $\theta$, whereas by its
> > definition in (2) it contains the survival function $\mathbb{P}(u^+(R(\tau')) > z)$
> > with $\tau' \sim \pi_\theta$, which also depends on $\theta$. Could the authors
> > clarify why the contribution from the $\theta$-dependence of this inner survival
> > function does not appear in the final expression? A precise statement of which
> > quantities are held fixed when $\nabla_\theta$ is applied would resolve this.
> >
> > **2. Interchange of differentiation and integration in $z$.**
> > The proof moves $\nabla_\theta$ inside the $\int_z$ integral and subsequently swaps
> > the order of the $z$- and $\tau$-integrals. Remark 21 justifies free differentiation
> > by appealing to the finiteness of the trajectory space, but the $z$-integral is over
> > a continuous range, so this remark does not cover the relevant interchange. Please
> > state the domination/regularity condition used (e.g., boundedness of $w'_\pm$ and
> > integrability of the dominating function over the bounded support $[0, M_u]$) and
> > justify the Fubini step explicitly. Since Theorem 5 underlies the entire method,
> > this needs to be made fully rigorous rather than asserted.
> >
> > **3.** Remark 19 and App. J.4 report a setting where a non-Markovian policy
> > outperforms a Markovian one, yet Proposition 3 establishes that reward-augmented
> > Markov policies in $ \Pi_{\Sigma,NS} $ are sufficient. I take it the gap is between
> > $ \Pi_{M,NS} $ (state + time) and $ \Pi_{\Sigma,NS} $ (state + accumulated reward + time),
> > i.e., the experiment's "Markovian" baseline does not include the reward augmentation.
> > Could the authors confirm this explicitly, and clarify in the main text that the
> > empirical comparison is $ \Pi_{M,NS} $ vs. $ \Pi_{\Sigma,NS} $ rather than against the
> > sufficiency guarantee?
> >
> > **4.** The bias bound (19) is stated as $C(1/\sqrt{n} + \sqrt{\log d}/\sqrt{m})$.
> > Could the authors confirm the $\sqrt{\log d}$ (rather than $\log d$) dependence is
> > what propagates into the vanishing-bias condition of Theorem 14, and that no
> > additional dimension dependence is hidden in $C$?

---

> > > ### Author Response · Authors · 2026-06-08
> > >
> > > We thank the reviewer for their careful follow-up and additional questions. We address each point below.
> > >
> > > **1. Differentiation in the proof of Theorem 5 (App. D.1).**
> > > The quantity $\phi$ is not held fixed before differentiating the objective. It appears only after applying the chain rule to $w_\pm(\mathbb P_\theta(u^\pm(R)>z))$ and rewriting $\nabla_\theta\mathbb P_\theta(u^\pm(R)>z)$ by the likelihood-ratio identity. For each fixed $z$, the event $(\tau:u^\pm(R(\tau))>z)$ is a fixed subset of the finite trajectory space; only its probability under $\rho_\theta$ depends on $\theta$. This dependence is exactly accounted for by the score term. We have expanded the proof in Appendix D.1 to make the $\theta$-dependence and the order of differentiation explicit.
> > >
> > > **2. Interchange of differentiation and integration in $z$.**
> > > We do not require additional domination/regularity conditions to justify differentiating under the integral with respect to $\theta$. The reason is that we are dealing with a finite horizon and finite state-action space setting which implies that the trajectory space is finite. As a consequence, $u^\pm(R(\tau))$ takes finitely many values, and the CPT integrals reduce to finite sums over intervals on which the survival events are constant. Differentiation is therefore applied to finite sums, and all order exchanges are finite-sum interchanges. We revised Appendix D.1 by replacing the previous brief remark (Remark 21) with a detailed finite-sum proof that justifies each step of the gradient computation.
> > >
> > > **3. Markovian vs. reward-augmented Markovian policies.** This is correct. We have clarified this in the main text by revising Remark 19. In particular, the Markovian baseline in the example does not include the accumulated reward, so the example does not contradict Proposition 3, which establishes sufficiency of reward-augmented Markovian policies.
> > >
> > > **4. Dimension dependence in the bias bound.** This is correct. The $\sqrt{\log d}$ dependence comes from the vector concentration step in the gradient estimation bound. No additional dimension dependence is hidden in the constant $C$, beyond the explicitly stated problem-dependent constants such as $H,M_\psi,M_u,L_w$ and $\bar{w}$. A complete proof of the bound, including this dependence, is given below Eq. (27) in App. E.4, pp. 37--38.

---

> > > > ### Comment · Reviewer_oMLZ · 2026-06-15
> > > >
> > > > Thanks for the clarification.
> > > >
> > > > I still believe that the paper is over-claiming contributions, or maybe a lot of things have not been clear to me.
> > > >
> > > > 1. Since the authors have shown that the state augmentation is necessary, I am assuming that the policy space is now in the augmented state. Thus, even if the policy space is finite, the resulting state space would be infinite. So, when you are talking about parameterization, are you talking about the discretization on the augmented space? How will it affect the performance? I do not see any impact of the parameter space $d$ in the sample complexity bound.
> > > >
> > > > 2. The first-order condition does not mean global convergence. The paper should mention it as a limitation. Further, the sample complexity depends on $\Delta_0$, $M_{\psi}$, $L_J$, and other network parameters. It also depends on $B_w$? For what types of weighing functions are they smooth? How would one know all these parameters?

---

> > > > > ### Author Response · Authors · 2026-06-16
> > > > >
> > > > > We thank the reviewer for their follow-up comments.
> > > > >
> > > > > **1.** Proposition 3 does not show that state augmentation is necessary; it proves that reward-augmented Markov policies are sufficient to optimize terminal-return objectives such as CPT-RL (Prop. 3 in section 2): the current state, time, and accumulated reward are sufficient, so arbitrary history dependence is not needed. This result was added for completeness following the reviewer’s earlier suggestion.
> > > > >
> > > > > The PG theorem in section 3 is stated for parametrized history-dependent policies (non-Markovian policies), as stated at the beginning of the section. Thus, the algorithm and analysis do not require discretizing the reward-augmented state space; they work directly with sampled trajectories and a finite-dimensional parametrized policy. In other words, in the PG formulation in section 3, we do not replace the original finite state space by a discretized reward-augmented state space.
> > > > >
> > > > > As for the dependence on the policy parameter dimension $d$, the explicit dependence in our gradient estimation and finite-time bounds is logarithmic, through terms such as $\log(d)$ following from high-probability bounds, rather than polynomial as in zeroth-order gradient estimation methods (e.g. L.A. et al. 2016). Of course, the constants in the bound, such as score bounds, depend on the chosen policy parametrization. This is consistent with standard first-order PG analysis, and more generally stochastic gradient descent using gradient information. We note also that this is one of the main advantages of first-order PG methods compared to zeroth-order methods proposed in prior work, which incur an explicit polynomial dependence on dimension due to zeroth-order estimation of policy gradients.
> > > > >
> > > > > **2.** Indeed, convergence to a first-order stationary policy does not imply global policy convergence to an optimal policy. We show asymptotic and non-asymptotic convergence guarantees to first-order stationary policies of the CPT-RL objective in our non-convex smooth policy optimization setting, as stated in the abstract, our contributions, and the statements of our results. We do not claim global convergence to optimal policies. Such computational optimality guarantees are not expected due to nonconvexity of the utility and weight transformations, which precludes convex or even hidden-convex structure.
> > > > >
> > > > > As for the dependence of the total sample complexity on problem constants in Theorem 15, it depends on $B_w$ through the smoothness constant $L_J$, which depends on $B_w$ (see Eq. (32), p. 40, for the total sample complexity with all constants made explicit, and Lemma 13, p. 10, for the definition of $L_J$). Twice continuously differentiable nondecreasing weight functions $w_{\pm}: [0,1]\to[0,1]$ such that $w_{\pm}(0)=0$ and $w_{\pm}(1)=1$ satisfy our smoothness conditions, since their first and second derivatives are bounded on the compact interval $[0,1]$. We provide examples in Remark 11. The dependence of the sample complexity on the smoothness constant $L_J$, the initial optimality gap $\Delta_0$, and the score bound $M_{\psi}$ is standard in nonconvex smooth optimization guarantees and PG analysis in particular. The constants may or may not be known depending on the setting. This is akin to standard existing analysis for non-adaptive gradient methods in smooth nonconvex optimization.

---

> > > > > > ### Comment · Reviewer_oMLZ · 2026-06-19
> > > > > >
> > > > > > I thank the authors for their responses. I don't have any more questions. I want to elaborate a few things:
> > > > > >
> > > > > > My point was that while I agree that if the policy model is expressive enough, then it can model the history and the optimal policy, but it is a different question of how one can develop an expressive policy model.
> > > > > >
> > > > > > I agree that global convergence can be difficult, but I would suggest looking into the robust MDP literature, where they prove the gradient dominance condition even though the problem is non-convex. But I agree that this is beyond the scope of this paper.

---

> > > > > > > ### Author Response · Authors · 2026-06-20
> > > > > > >
> > > > > > > We thank the reviewer for the clarification and helpful comments. We agree that, beyond the sufficiency result for reward-augmented Markov policies, the design of expressive policy classes tailored to such augmented/history-dependent representations is a distinct and important question.
> > > > > > >
> > > > > > > We also appreciate the pointer to the robust MDP literature on gradient domination. We agree that this is an interesting direction. This question deserves a dedicated treatment in our setting, since the available robust-MDP arguments rely on robust-MDP-specific structure, such as performance-difference identities together with uniqueness of the worst-case transition kernel or the worst-case action-value function. CPT objectives are structurally different: nonlinear probability distortion is applied to survival probabilities of the terminal return, so the marginal contribution of a trajectory depends on the overall return distribution rather than only on that trajectory’s own return. This structure also prevents direct Bellman-style decompositions. We have added this direction as future work in the conclusion (in blue in the revised manuscript), in connection with the limitation that our current guarantees establish first-order stationarity.

---

### Review · Reviewer_zzN8 · 2026-05-28

**Summary Of Contributions:**

This paper proposes a policy gradient algorithm for optimizing the CPT of the cumulative reward in a finite horizon MDP setting. The contributions include a policy gradient theorem with CPT objective, a Monte Carlo estimate of the CPT gradient using order statistics, consistency guarantee for this estimate, asymptotic convergence of the overall CPT-PG algorihtm and some simulation experiments.

The paper is reasonably well-written and I could follow the technical aspects.

As the authors mention in the paper, the primary contributions of this work are theoretical and I had some serious concerns about the theoretical development, and the contributions of this work in comparison to the results already available in the literature.

## Major concerns/weaknesses
1. In comparison to (Vijayan & L.A, 2024), the authors claim CPT is more general as it has utility functions and a reference point. How do these impact *reinforcement learning* in general and the derivation of the policy gradient theorem as well as CPT gradient estimation, in particular? From a technical standpoint, utility functions do not pose much of a challenge for the derivation of the CPT policy gradient theorem, while the weight functions do and the latter are already part of DRMs. To reiterate, I see only completely parallel arguments in the former, and replacing return r.v. samples by those where an apt utility function is applied.
2. Previous work on DRM imposed a Lipschitzness requirement on the distortion function. The authors here impose the same, while claiming to handle CPT. The problem is that the distortion function underlying the celebrated work of Tversky and Kahnemann is Holder and not Lipschitz. Prelec's distortion/weight function is not Lipschitz either. In remark 11, the authors mention that one could use endpoint-regularized proxy of CPT weights. How does one choose the parameter $\epsilon$ in this proxy? Intuitively, CPT theory is based on the idea that rare events are overweighted, while the proxy variant of weight may be going against this idea.
3. While DRMs may be a special case of CPT, a more general distortion riskmetric (DR) is not. In particular, DRs include a variety of deviation measures, including Gini deviation and do not impose $w(1)=1$. General DRs have been handled in a RL context in reference [2] listed below.
4. The convergence guarantee for the proposed CPT-PG algorithm is to a first-order stationary point, and these are not necessarily local optima. In [2], for DRs, the authors have established convergence to a second-order stationary point (SOSP), which would coincide with local optima under a strict saddle condition. Given the non-convex nature of the objective, I find it perplexing that the authors dismiss SOSP convergence for CPT by saying "We
focus on first-order PG algorithms without resorting to higher-order Hessian information and ...". This justification is not convincing. Moreover, the sample complexity to find an $\epsilon$ approximate FOSP in this work is $O(1/\epsilon^4)$, while the Newton method finds an $\epsilon$ approximate SOSP with $O(1/\epsilon^3)$ for DRs.
5. The asymptotic convergence guarantee requires boundedness of iterates, which is imposed as an assumption in Theorem 14. Such an assumption cannot be verified, and the authors mention using projections.  The asymptotic convergence requires diminishing step sizes, while the finite time analysis uses a constant step size and this disconnect is not discussed. The finite time bounds do not require projections either. Thus, the convergence guarantees in the asymptotic and non-asymptotic regimes uses different parameter setting and different update iterations, and it is not clear if this difference is strictly necessary. For instance, why not have rates in Theorem 15 for diminishing step sizes?
6. Misleading sample complxity under Lipschitz assumption. CPT estimation requires more than the folklore $O(1/\epsilon^2)$ samples owing to Holder weight functions, see (Jie et al. 2018) for a lower bound. A similar claim should hold for CPT gradient estimation as well and the authors sidestep it by imposing a Lipschitz assumption on the weights.
7. Reference [1] is an important recent work that handles CPT using the policy gradient approach, and it is missed in the related work of this submission.
8. In [1], the CPT gradient estimate is improved by variance reduction. In particular, the CPT gradient estimate there has no cross terms, unlike the one in line 10 of Algorithm 1. These cross terms add to the variance owing to independence of trajectories.

On the empirical side, I am unable to infer the message from the results. For instance, in MuJoCo, vanilla REINFORCE results in better CPT returns in Figure 25, if I have read it correctly. In [1], the authors exhibit better risk profiles with a CPT-PG style algorihtm with a KL regularizer, while [2] shows better returns with risk-seeking DRs. In contrast, from the experiments,it is not clear how CPT reference point/utilities lead to better policies over vanilla REINFORCE. Apart from CPT returns, it is necessary to know why a CPT-PG policy is preferable and I did not find this in the finance/MuJoCo experiments.

## References
1. Markowitz, J., Gardner, R. W., Llorens, A., Arora, R., and Wang, I.-J. (2023). A Risk-Sensitive Approach to Policy Optimization. Proceedings of the AAAI Conference on Artificial Intelligence, 37(12), 15019–15027.
2. Pachal, S., Maniyar, M. P., and A., P. L. (2026). Policy Newton Methods for Distortion Riskmetrics. Proceedings of the AAAI Conference on Artificial Intelligence, 40(29), 24674–24681.

**Additional Comments:**

See the first box for the detailed comments.

**Audience:**

Yes

**Audience Explanation:**

Risk-sensitive RL has received increased research attention over the last decade in the ML community and the contributions of this paper fall under this realm.

**Broader Impact Concerns:**

None.

**Claims And Evidence:**

No

**Claims Explanation:**

See the weaknesses listed in the box above. In particular, the authors claim to address CPT saying it is more general than distortion risk measures. However, the claims are problematic owing to
1. Lipschitz assumption on the weights;
2. the fact that distortion riskmetrics are not special cases of CPT; and
3. an important related work (reference [1] in the list above) that proposes a policy gradient style algorithm for CPT, is missed.

**Requested Changes:**

1. Compare against references [1] and [2], and show why the algorithmic/analytic contributions of this work are non-trivial in comparison.
2. For CPT gradient estimation, handle the case of Holder weight functions.
3. Emprically, establish the benefits of using CPT, beyond showing the CPT returns are better for the proposed algorithm.
4. Highlight the significant deviations in proofs compared to [1], [2], and [Vijayan et al. 2021]. In addition, for the CPT policy gradient theorem justify the interchange of differentiation and integration rigorously using DCT.
5. Reduce variance in gradient estimation and establish convergence to  and SOSP, and if possible, keep the asymptotic and non-asymptotic guarantees for the same algorithm (i.e., same updates and similar hyperparameter choices). Alternately, for the case of tabular parameterization, check if gradient domination holds and if an improved convergence guarantee (say to CPT-optimal policy) can be obtained.

In my opinion, a setting that is not considered in [1], [2], and [Vijayan et al. 2021] corresponds to an MDP with return r.v. that is not necessarily bounded. CPT and its gradient estimation in this regime is challenging. If the authors address this aspect, then it is a clear contribution beyond existing results for CPT/DRM/DRs.

---

> ### Author Response · Authors · 2026-06-04
>
> We thank the reviewer for the detailed comments. We revised the manuscript to clarify prior work, regularity assumptions, and empirics.
>
> **1.** CPT-PG is not the existing DRM estimator applied to utility-transformed returns. Vijayan & L.A. use a CDF-gradient representation with a plug-in empirical CDF-gradient estimator. Theorem 5 instead gives a REINFORCE-style identity with a trajectory-level CPT marginal valuation $\phi$. Algorithm 1 estimates $\phi$ by order statistics and uses it in a score-function average, so the proof controls order-statistic and Monte Carlo score-average errors. In the risk-neutral case, it reduces to vanilla REINFORCE.
>
> **2.** Theorem 5 itself does not require Lipschitzness and now assumes only differentiability of $w_\pm$. Lipschitzness of $w_\pm'$, not $w_\pm$, is used only for sample complexity. TK/Prelec weights are typically Hölder but not Lipschitz on $[0,1]$; their derivatives are endpoint-singular and hence neither bounded, Lipschitz, nor Hölder on $[0,1]$. Thus Hölder $w_\pm$ helps CPT value estimation, not gradient estimation. As noted in Remark 11, endpoint-regularized TK/Prelec weights satisfy our assumption. $\varepsilon$ is a probability-resolution parameter: with $n$ trajectories, probabilities below $O(1/n)$ cannot be reliably resolved. Regularization preserves the CPT shape on $[\varepsilon,1-\varepsilon]$, including resolvable rare-event overweighting, while capping infinite marginal sensitivity near 0 and 1. If $w_\pm'$ is only $\alpha$-Hölder, the same proof gives $\tilde O(\epsilon^{-2/\alpha})$ sample complexity, as stated in Remark 10.
>
> **3.** We revised related work accordingly. The settings are complementary: Pachal et al. study second-order Newton methods for distortion risk metrics, including deviation-type functionals; we study first-order PG for CPT objectives with reference dependence, gain/loss utilities, and asymmetric weighting. Our PG estimator is not a plug-in CDF-gradient estimator.
>
> **4.** Pachal et al.’s SOSP guarantee is stronger, but for a different objective class and second-order updates. Our contribution is a first-order CPT PG theorem and algorithm, so FOSP convergence is the natural guarantee.
>
> **5.** Bounded iterates are standard in stochastic approximation; one can also use projected updates on a compact set, as in L.A. et al. (2016) as noted after Theorem 14. Diminishing steps give exact asymptotic convergence by vanishing noise. The finite-time theorem targets approximate FOSP guarantees, where constant steps give sharper bounds. With decreasing Robbins--Monro steps, the bound scales as $1/\sum_{t<T}\alpha_t$, worse than the constant-step $O(1/T)$ rate. We added Remark 16.
>
> **6.** Under Lipschitzness of $w_\pm'$, our $\tilde O(\varepsilon^{-2})$ CPT PG sample complexity is consistent with L.A. et al. (2016). Under only $\alpha$-Hölder $w_\pm'$, the reviewer is right: the rate degrades to $\tilde O(\varepsilon^{-2/\alpha})$. We added Remark 10.
>
> **7.** We added Markowitz et al. and clarified the distinction. Their CPT-inspired objective differs from the standard gain/loss CPT value used here and in L.A. et al. (2016): they weight ranks/CDFs of returns, whereas CPT first transforms returns into gain/loss utilities and then distorts survival functions of the transformed variables. Their rank-based estimator differs from our marginal-valuation estimator in a REINFORCE-style weighted score average. They focus on practical PPO/KL-regularized optimization; we give a PG theorem, tailored estimator, sample-complexity bounds, and convergence guarantees.
>
> **8.** The cross-trajectory terms removed in Markowitz et al. arise from their CDF-gradient approximation. Our auxiliary trajectories only estimate the scalar marginal weight $\phi$ and do not contribute likelihood-ratio scores, so cross-term removal does not apply directly.
>
> **Empirical concern.** The empirical section has three messages. First, with fixed parameters, CPT-PG reproduces the reflection effect: risk aversion over gains and risk seeking over losses. Second, comparison with CPT-SPSA-G shows likelihood-ratio information improves robustness over zeroth-order SPSA as state space grows. Third, the finance and other experiments illustrate interpretable control through reference points, utility curvature, and weighting parameters.
>
> Interchanging integral and differentiation does not require DCT because the CPT integral reduces to a finite trajectory sum; see Remark 21.
>
> Unbounded returns require tail/integrability assumptions beyond our bounded-reward finite-horizon setting, and are left for future work. We also do not expect gradient domination generally, since probability distortion breaks the hidden convexity used in standard or convex RL.
>
> Compared to DRM work, we also prove reward-augmented Markov policy sufficiency (Prop. 3), PG estimator consistency, asymptotic convergence of the algorithm, and high-probability rather than in-expectation sample-complexity guarantees.

---

> > ### Comment · Reviewer_zzN8 · 2026-06-12
> >
> > ## Response to authors
> >
> > 1. Thanks for clarifying the difference between your CPT gradient estimator and the ones in [Vijayan et al.]. My main question now is as follows: What is the utility of such an estimator, when it is straightforward to obtain a DRM-style one with utility transformed returns? One can easily adapt the [Vijayan et al.]/[Pachal et al.] results to obtain order statistics based CPT gradient estimates and remove certain cross terms to even reduce variance. A similar observation applies to [Markowtiz et al.]. At the very least, the benefits of the new CPT gradient estimator has to be established in comparison to the ones in the aforementioned references.
> >
> > 2. On Lipschitz weights requirement: This is restrictive and the authors are overselling their contribution as going beyond DRM to cover probability distortions underlying CPT. Connecting to the point above, covering Lipschitz weights is fairly straightforward given policy gradient works done for DRMs and the contribution in this context for end-point regularized CPT is marginal, while hanlding vanilla Holder weights from CPT would lead to non-trivial theoretical advance.
> >
> > 3. In comparison to [Pachal et al.], I do not understand why the contributions here are complementary. As mentioned above, it is straightforward to adapt the results in [Pachal et al.] to get improved theoretical results for CPT with Lipschitz weights, e.g., convergence to an SOSP. This submission establishes convergence to an FOSP, which include saddle points.
> >
> > 4. FOSP is not necessarily the natural guarantee in a non-convex landscape, while it is preferabble to have algorithms that escape saddle points.
> >
> > 5. Ok.
> >
> > 6. Where is the proof for the claim in Remark 10? I could not find it.
> >
> > 7. The comparison to [Markowitz et al.] is not enough to establish the superiority of the CPT gradient estimates here. Moreover, the empirical results appear stronger in the aforementioned reference.
> >
> > 8. Same observation as in points 1 and 2 above.
> >
> > The authors have not clearly responded to the following earlier comment: "For instance, in MuJoCo, vanilla REINFORCE results in better CPT returns in Figure 25, if I have read it correctly. In [1], the authors exhibit better risk profiles with a CPT-PG style algorihtm with a KL regularizer, while [2] shows better returns with risk-seeking DRs. In contrast, from the experiments,it is not clear how CPT reference point/utilities lead to better policies over vanilla REINFORCE".
> > The MuJoCo results then were misleading and the interpretability/appeal of CPT on continuous control tasks is unclear. In my opinion, it is not enough to show that the proposed algorithm results in better CPT returns. It is also necessary to show why the CPT-based converged policy is appealing over vanilla REINFORCE.

---

> ### Author Response · Authors · 2026-06-15
>
> We thank the reviewer for engaging further with the revision. We address the remaining concerns below.
>
> **1. Utility of the estimator and comparison to DRM-style alternatives.** We agree that one can consider alternative CPT estimators by adapting CDF-gradient or rank/order-statistic estimators from DRM work after transforming returns using asymmetric utility and probability weight distortion functions. We have clarified this in the related work section (end of section 6) and do not claim that our estimator is uniformly superior to such alternatives.
>
> The contribution of our estimator is different: it follows directly from our CPT policy-gradient theorem for the standard gain/loss CPT objective with reference dependence, asymmetric utilities, and potentially different gain/loss probability distortions. The resulting estimator has a REINFORCE-style form: a scalar trajectory-level CPT marginal valuation multiplied by the cumulative score function. It does not require an empirical CDF-gradient term, reduces exactly to vanilla REINFORCE in the risk-neutral case, and can therefore be incorporated into existing policy-gradient implementations with minimal structural changes. Our theoretical contribution is to provide the full analysis for this estimator: consistency, finite-sample gradient-estimation bounds, asymptotic convergence, and finite-time FOSP guarantees.
>
> We agree that a direct theoretical and empirical comparison against adapted DRM-style CPT estimators would be valuable and we now state this as future work (after our related work remark).
>
> **2. Lipschitz regularity of probability weights.**
> We agree that the Lipschitz assumption on the derivatives of the weighting functions is restrictive and excludes classical unregularized KT/Prelec weights (while it is satisfied by their endpoint-regularized versions as we mention in Remark 11). We have revised the manuscript to highlight this limitation in the conclusion as an opportunity for future work to address endpoint-singular weighting functions. The CPT policy-gradient identity itself does not require this Lipschitz condition; it is used in the sample-complexity analysis to control the order-statistic approximation error.
> Thus, our contribution should be read as covering CPT objectives with sufficiently regular or endpoint-regularized probability distortions, not as resolving the harder vanilla Hölder/singular KT-Prelec case. We agree that treating the classical endpoint-singular weights directly would be a more substantial theoretical advance.
>
> **3. Relation to Pachal et al.** Compared with [Pachal et al.], (a) we deal with a different objective, we propose a different estimator which does not build on their approach (their PG estimator does not reduce to ours when dropping second-order policy Hessian information); (b) We prove a novel policy gradient theorem, we show consistency of our PG estimator and establish asymptotic convergence to FOSP which are not established in [Pachal et al.], in addition to our sample complexity FOSP results.
> Hence, our contributions are complementary in the sense that we propose a different REINFORCE-style CPT policy-gradient representation and analyze a distinct first-order estimator, rather than a second-order Newton-type extension of DRM.
> Extending our results to establish SOSP guarantees under additional second-order information and policy smoothness assumptions, and comparing to an extension of their distinct approach to CPT would be valuable directions for future work, and we have now highlighted this in the conclusion.
>
> **4. FOSP vs SOSP.** We agree that SOSP guarantees are stronger than FOSP guarantees when available in the same setting. In our distinct CPT setting, the CPT-PG algorithm is a first-order PG method and the standard nonconvex guarantee is convergence to FOSP.
> In particular, we do not use second-order policy Hessian information and we do not make stronger second-order smoothness assumptions required by prior work establishing SOSP guarantees *for DRM* (such as A3 and A4 in Pachal et al. including Hessian score smoothness). In this setting, convergence to FOSP is the standard default guarantee in nonconvex smooth optimization. Note that hidden convexity is not immediately available in general to target global optimality. Extending our guarantees to obtain SOSP guarantees in our CPT setting under adequate additional assumptions is left for future work.  We have now revised the conclusion to add this.
>
> **6.** We have added the proof at the end of Appendix E.2 (p. 35-36) and a pointer to the proof in Remark 10. The proof follows the same argument as the Lipschitz case after replacing the Lipschitz control by $\alpha$-Hölder control, which yields the stated degraded dependence.

---

> > ### Author Response · Authors · 2026-06-15
> >
> > **7-8.** As we mentioned, [Markowitz et al.] address a different setting which is not the standard CPT objective form and follow a different CDF-gradient plug-in approach to develop a policy gradient estimator. We do not claim superiority of our PG estimator which is different and addresses a different setting. While [Markowitz et al.] provide more experimental evidence for their setting, our focus is different as our main methodological and theoretical contributions are not covered by [Markowitz et al.]. These include a CPT policy-gradient theorem, a corresponding first-order estimator, consistency, finite-sample gradient-estimation bounds, and asymptotic as well as non-asymptotic convergence guarantees.
> >
> > **Simulations in appendix J.9.** We agree that the MuJoCo appendix should not be read as showing that CPT-PG learns policies preferable to vanilla REINFORCE. The purpose of that appendix is only to show that the proposed algorithm can be implemented in a continuous-control benchmark and that different CPT profiles induce different CPT-return curves. Thus, the returns are not comparable across different profiles, and we do not seek to show that we achieve higher returns. The plotted quantities are CPT returns under different CPT profiles and are not meant to establish better mean return or better risk profile than REINFORCE, [Markowitz et al.], or [Pachal et al.]. As we mention, 'we test our algorithm on the InvertedPendulum-v5 environment to demonstrate that our PG algorithm is also applicable to other control benchmarks with continuous state and action spaces.' Note that Figure 25 displays *CPT* returns. [Markowitz et al.] consider a different setting and display different quantities (e.g. mean episode reward). [Pachal et al.] also consider a different setting (with risk-seeking DRs) and display the average episodic returns rather than CPT return.
> >
> > The main qualitative evidence for interpretable CPT-induced policy behavior is instead in the discrete examples where learned policies can be directly visualized, e.g. Figures 19 and 22. Our work is focused on the discrete state-action setting. A complete study for the continuous state-action space setting with further experiments beyond the preliminary experiments in the appendix is an interesting direction for future work. We have now highlighted this in the conclusion.